# Coordinated hippocampal-thalamic-cortical communication crucial for engram dynamics underneath systems consolidation

Douglas Feitosa Tomé [1], Sadra Sadeh [1] & Claudia Clopath [1]✉

Systems consolidation refers to the time-dependent reorganization of memory representations or engrams across brain regions. Despite recent advancements in unravelling this process, the exact mechanisms behind engram dynamics and the role of associated pathways remain largely unknown. Here we propose a biologically-plausible computational model to address this knowledge gap. By coordinating synaptic plasticity timescales and incorporating a hippocampus-thalamus-cortex circuit, our model is able to couple engram reactivations across these regions and thereby reproduce key dynamics of cortical and hippocampal engram cells along with their interdependencies. Decoupling hippocampal-thalamic-cortical activity disrupts systems consolidation. Critically, our model yields testable predictions regarding hippocampal and thalamic engram cells, inhibitory engrams, thalamic inhibitory input, and the effect of thalamocortical synaptic coupling on retrograde amnesia induced by hippocampal lesions. Overall, our results suggest that systems consolidation emerges from coupled reactivations of engram cells in distributed brain regions enabled by coordinated synaptic plasticity timescales in multisynaptic subcortical-cortical circuits.

[1] Department of Bioengineering, Imperial College London, London, UK. ✉email: c.clopath@imperial.ac.uk

Pioneering hippocampal lesion studies[1–3] have motivated an ever-growing body of lesion experiments[4,5] with a common goal of understanding the role of hippocampus (HPC) and neocortex in systems consolidation of memory. In turn, this spawned many theories of this process but with widely different views concerning its underlying mechanisms and properties. These discrepancies can be mainly attributed to seemingly conflicting reports in the retrograde amnesia literature[4,5]. Specifically, retrograde amnesia induced by hippocampal damage has been reported as temporally-graded (i.e., recent memories are lost but remote memories are spared following hippocampal lesion), flat (i.e., recent and remote memories are disrupted by hippocampal lesion), or absent (i.e., recent and remote memories are preserved post hippocampal lesion). In light of these experimental findings, some systems consolidation theories posited that HPC is essential for recent but not for remote memory recall[6–16] while others have proposed that HPC is either always necessary for recall[4,17–19] or required for recall depending on the circumstances of encoding and retrieval[20–29]. Despite their differences, these theories share the view that systems consolidation relies on interactions between HPC and neocortex. Surprisingly, it has been recently demonstrated that thalamic spindles have a causal role in systems consolidation by coupling hippocampal, thalamic, and cortical oscillations[30]. Therefore, current theories of systems consolidation fail to provide a unifying framework that reconciles the available experimental data.

Recent advances in experimental technologies have the potential to clarify the nature and dynamics of systems consolidation by enabling the identification and manipulation of engrams – more specifically of engram cells[31]. These cells are defined as a set of neurons that become active in response to learning, undergo enduring changes as a result of learning, and are able to be reactivated when presented part of the original stimuli resulting in memory recall[32]. Adopting this definition, a landmark contextual fear conditioning (CFC) study found that engram cells in medial prefrontal cortex (CTX) are initially generated in a silent state (i.e., cannot be reactivated from a partial cue) but over time gradually become active (i.e., can be reactivated from a partial cue)[33]. In contrast, engram cells in HPC are active following learning but eventually turn silent. The silent-to-active transition in CTX engram cells was named maturation and the active-to-silent switch in HPC engrams was termed de-maturation. Both engram dynamics are associated with systems consolidation of memory. Moreover, the output of HPC engram cells after learning was found to be crucial for the subsequent maturation of CTX engrams. It has been proposed that the observed dynamics of engrams in CTX and HPC[33] are "mirrored" in different types of episodic memory[5]. Nevertheless, the exact neural mechanisms underlying these engram dynamics and the role of associated circuits remain unknown and, consequently, the ability of recent engram findings to advance our knowledge towards a consistent view of systems consolidation is hindered. This is at least in part due to existing theoretical and computational models lagging behind the groundbreaking advancements in engram cell research enabled by new technologies developed in the past decade[31,32,34,35]. In particular, previous computational studies have employed abstract neuronal models that are intended to capture high-level properties of systems consolidation (e.g., recent memory recall relies on HPC) but are unable to reproduce engram cell-level data produced by recent experiments[9,11,25,36–40].

Here, our goal is to provide insights into engram cell dynamics and associated pathways using computational modeling. To that end, we simulate systems consolidation in an episodic memory task using a multi-region spiking recurrent neural network model subject to biologically-plausible plasticity mechanisms acting on different timescales in distinct brain regions. Contrary to current theories[4,6–29], our results show that direct, monosynaptic HPC → CTX projections cannot reproduce the known inter-dependencies between engrams in these regions[33]. However, a network with hippocampal-thalamic-cortical communication is able to overcome this limitation. Specifically, after verifying that our model with three-region communication displays engram cell maturation in CTX and de-maturation in HPC, we then show that HPC engram cells as well as coupled engram reactivations across brain regions are essential for proper engram dynamics in line with previous experiments[30,33]. Our modeling results also yield the following experimentally-testable predictions: engram cells in mediodorsal thalamus (THL) are active in recent and remote recall and are crucial for the maturation of engram cells in CTX; engram cells in HPC and THL are crucial for coupling engram reactivations across HPC, THL, and CTX in consolidation periods; inhibitory engram cells have distinct region-specific dynamics with coupled reactivations; inhibitory input to THL is critical for CTX engram maturation; and THL → CTX synaptic coupling is predictive of CTX engram dynamics and the retrograde amnesia pattern induced by HPC damage—thus providing a unifying mechanistic account for reconciliation of HPC lesion studies. Altogether, our results suggest that coordinated hippocampal-thalamic-cortical communication underlies engram dynamics subserving systems consolidation.

## Results

**Synaptic plasticity timescales drive engram cell dynamics.** To understand the mechanisms underlying engram cell dynamics, we start by examining the effects of synaptic plasticity timescales on the initial state and subsequent evolution of engram cells. We use spiking neural network models that consist of a stimulus population (STIM) that projects to both HPC and CTX (Fig. 1a). Feedforward and recurrent synapses are initialized at random with excitatory synapses onto excitatory neurons displaying long-term plasticity and inhibitory synapses onto excitatory neurons exhibiting inhibitory plasticity. Long-term excitatory plasticity is composed of a combination of Hebbian and non-Hebbian forms of plasticity[41]. The Hebbian term takes the form of triplet spike-timing-dependent plasticity (STDP)[42] while the non-Hebbian terms include heterosynaptic plasticity[43] and transmitter-induced plasticity[44]. Importantly, the heterosynaptic plasticity term incorporates synaptic consolidation dynamics[41]. Inhibitory synaptic plasticity consists of a network activity-based STDP term[41] whose primary goal is to regulate firing rate levels[45] (for a detailed description of the model, see Methods). One of four non-overlapping random stimuli is presented to the network at a time either for training or testing (Fig. 1a), and the network is subject to an episodic memory task to investigate engram dynamics. Following a brief burn-in period to stabilize network activity, the network simulation consists of three consecutive phases: training, consolidation, and testing (Fig. 1b). In the training phase, the complete stimuli (i.e., full patterns) are randomly presented to the network. Next, no stimulus is presented to the network during the consolidation phase and, consequently, the network is allowed to evolve spontaneously. At different points throughout the consolidation phase, the network proceeds to the final testing phase where partial cues of the original stimuli are presented and the ability of HPC and CTX to recall the encoded memory is evaluated.

Engram cells are formed in both HPC and CTX at the end of training (Fig. 1c). These cells are identified via the average stimulus-evoked firing rate of neurons (see Methods). We then compute the mean recurrent excitatory weights between engram cells encoding the same stimulus (i.e., within-ensemble) and between engram cells representing different stimuli (i.e., inter-ensemble). We plot the computed weights in a matrix format

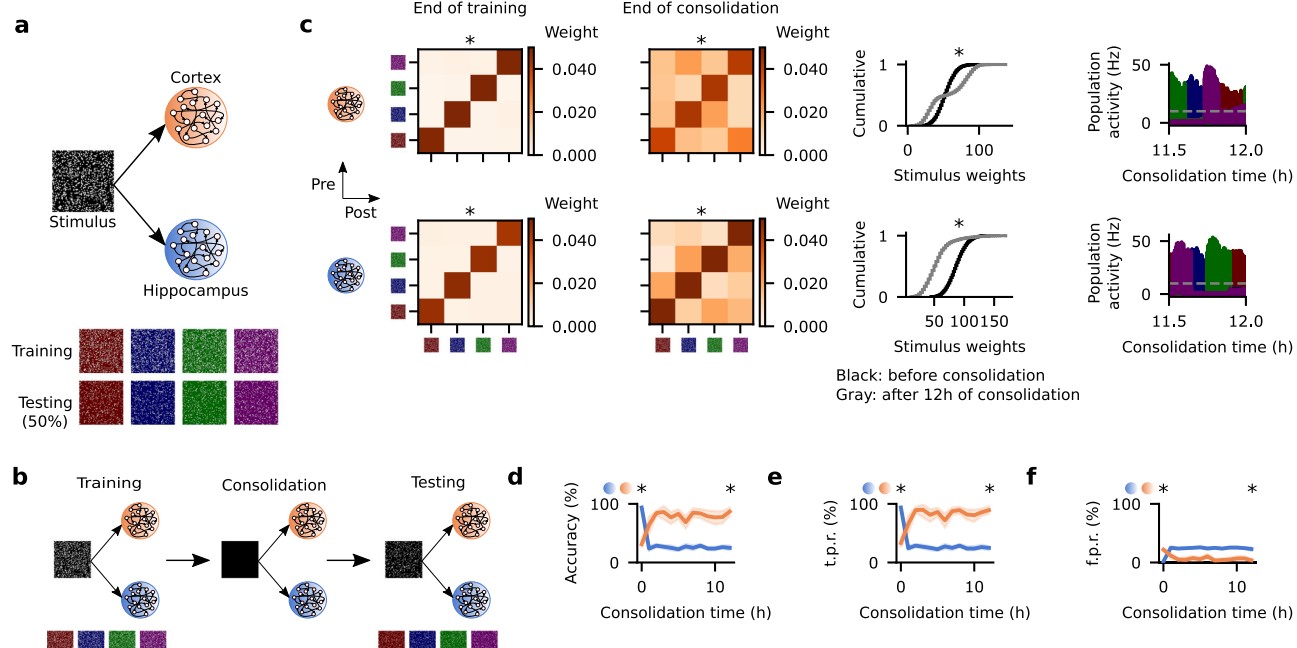

**Fig. 1 Divergent synaptic plasticity timescales lead to opposite engram cell dynamics. a** Schematic of network model with Stimulus (STIM), Hippocampus (HPC), and Cortex (CTX) (top) and stimuli presented in the training phase with their respective partial cues used in the testing phase (bottom). In STIM, training stimuli, and testing stimuli, light gray indicates active neurons firing above baseline whereas the remaining background color (i.e., black in STIM and red/blue/green/purple in training and testing stimuli) indicates neurons at baseline firing rate. Each background color in training and testing stimuli denotes a distinct non-overlapping stimulus and its respective partial cue. **b** Schematic of simulation protocol. **c** From left to right: mean weight strength of recurrent excitatory synapses onto excitatory neurons at the end of the training phase clustered according to engram cell preference (i.e., only mean weights between engram cells), mean weight strength of recurrent excitatory synapses onto excitatory neurons after 12 h of consolidation clustered according to engram cell preference (i.e., only mean weights between engram cells), cumulative distribution of the total feedforward synaptic weights from STIM onto individual engram cells, and population activity of engram cells in the consolidation phase (each color designates the engram cells encoding the respective stimulus in (**a**); dashed line indicates threshold $\zeta^{thr} = 10$ Hz for engram cell activation). Top: CTX. Bottom: HPC. Two-sided Kolmogorov–Smirnov test between the distribution of recurrent excitatory weights among engram cells encoding the same stimulus (i.e., diagonal) and that of recurrent excitatory weights among engram cells encoding different stimuli (i.e., off-diagonal) at the end of training (CTX: $p$-value = 0; HPC: $p$-value = 0) and at the end of consolidation (CTX: $p$-value = 0; HPC: $p$-value = 0). Two-sided Kolmogorov–Smirnov test between the distribution of feedforward stimulus weights onto excitatory engram cells at consolidation time = 0 and 12 h (CTX: $p$-value = $3.387302 \times 10^{-182}$; HPC: $p$-value = 0). **d–f** Memory recall in the testing phase as a function of consolidation time. $n = 10$ trials. Mean values and 90% confidence intervals shown. **d** Recall accuracy. Two-sided Mann–Whitney $U$ test between accuracy in HPC and CTX at consolidation time = 0 ($p$-value = $1.004910 \times 10^{-4}$) and 12 h ($p$-value = $1.333409 \times 10^{-4}$). **e** Recall true positive rate. Two-sided Mann–Whitney $U$ test between true positive rate in HPC and CTX at consolidation time = 0 ($p$-value = $1.004910 \times 10^{-4}$) and 12 h ($p$-value = $1.277653 \times 10^{-4}$). **f** Recall false positive rate. Two-sided Mann–Whitney $U$ test between false positive rate in HPC and CTX at consolidation time = 0 ($p$-value = $1.407747 \times 10^{-4}$) and 12 h ($p$-value = $2.194140 \times 10^{-4}$). *$p$-value < 0.05 (see Methods). **c–f** Color as in (**a**).

where within-ensemble mean weights are located in the diagonal while inter-ensemble mean weights are positioned off-diagonal. The resulting block diagonal structure (i.e., strong diagonal and weak off-diagonal mean weights) of the recurrent excitatory synapses in HPC and CTX shows that engram cells in these regions have encoded the four stimuli by the end of the training phase. After 12 h of consolidation, the diagonal structure is preserved in both regions. However, feedforward synapses projecting to HPC and CTX evolve in opposite ways in the consolidation phase. Specifically, while the cumulative distribution of the total feedforward synaptic weights to individual engram cells in HPC shows that there is a decrease in STIM → HPC weights, the reverse was observed in CTX. These changes in STIM feedforward weights are consistent with experimental findings which showed that changes in the dendritic spine density of engram cells over time are region-specific with cells in HPC experiencing a decrease but neurons in CTX undergoing an increase in spine density[33]. Furthermore, we plot the population activity of engram cells encoding each stimulus (i.e., average firing rate of engram cells without smoothing or

convolution, see Methods) and we measure the degree of coupling between engrams in two different regions in our network by defining lag$_{max}$ as the lag that maximizes the correlation between the population activity of engram cells in one region and the population activity of engram cells in another region (see Methods). Although engrams are spontaneously reactivated during the consolidation period in the two regions of the model (Fig. 1c), engram reactivations are not coupled as evidenced by engrams encoding the same stimulus in HPC and CTX having |lag$_{max}$| between 132.56 and 940.17 s. (Supplementary Fig. 1). This was expected given that there are no connections between HPC and CTX (Fig. 1a) and, hence, they behave independently in this network configuration. In addition, the timescale and firing rate of engram reactivations in the model are set by the rate of transmitter-induced plasticity and the time constants of spike-triggered adaptation (compare Fig. 1c to Supplementary Fig. 2). Critically, the differences in engram dynamics in HPC and CTX are a direct result of their diverging synaptic plasticity timescales: learning rate ($\eta$) and synaptic consolidation time constant ($\tau^{cons}$) are higher in HPC relative to CTX.

We next evaluate the ability of the network to retrieve memories from partial cues by computing three memory recall metrics. First, we compute recall true positive rate (t.p.r.) as the fraction of cue presentations in the testing phase that elicit population responses of the corresponding engram cells above the threshold $\zeta^{thr} = 10\,Hz$. Second, we measure recall false positive rate (f.p.r.) as the average fraction of engram cell ensembles encoding stimuli different than the one corresponding to a partial cue whose population responses were nonetheless above the threshold $\zeta^{thr} = 10\,Hz$. Third, we define recall accuracy as the fraction of cue presentations that only elicited a population response above the threshold $\zeta^{thr} = 10\,Hz$ for the engram cells encoding the respective stimulus (for further details on the previous recall metrics, see Methods). The resulting memory recall curves show that the model exhibits de-maturation and maturation of engram cells in HPC and CTX, respectively (Fig. 1d–f), in line with reported experiments[33]. At the end of training (i.e., 0 h of consolidation), t.p.r. is nearly 100% and f.p.r. is virtually 0% in HPC, leading to a corresponding recall accuracy of almost 100%. In CTX, however, t.p.r. is approximately 40% and f.p.r. is around 20% with a resulting accuracy of ~40% at the end of encoding. Over the course of the consolidation phase, though, the recall metrics reverse: t.p.r. and accuracy decrease in HPC but they increase in CTX. Importantly, the changes in recall accuracy are reflected in the changes in t.p.r. in both regions. This means that engram cells in HPC are initially reactivated in response to partial cues but over time become unable to do so while those in CTX cannot be reactivated by cues immediately after training but acquire this ability over the course of consolidation. Consequently, memory recall switches from HPC to CTX with systems consolidation. These engram dynamics are a direct result of region-specific changes in STIM feedforward weights: depression of STIM → HPC synapses and potentiation of STIM → CTX projections (Fig. 1c). Note that (I) HPC engram cells are able to retain their recurrent excitatory structure despite turning silent because of engram reactivations during consolidation (Fig. 1c) and (II) CTX engram cells already have structured recurrent excitatory connectivity at the end of encoding when they are still silent and this enables engram reactivations throughout consolidation (Fig. 1c). Given the block diagonal structure of the recurrent excitatory weights of silent engram cells in both HPC and CTX, our model suggests that optogenetically stimulating silent engrams in either region triggers memory recall. Interestingly, previous experiments have demonstrated that this is the case[33]. Altogether, our modeling results suggest that variability in synaptic plasticity timescales underlies the observed divergence in engram dynamics across distinct regions of the brain.

We also perform additional analyses to gain further mechanistic insights into the engram dynamics in our model. First, we train our network on overlapping stimuli and find that it exhibits engram dynamics analogous to our original results with non-overlapping stimuli (Supplementary Fig. 3). Specifically, we train the network on a set of random stimuli where each stimulus is represented by a random 25% of the neurons in the stimulus population resulting in an average overlap of 25% between stimulus pairs (Supplementary Fig. 3a). We test memory recall with partial cues consisting of a random 50% of the corresponding full stimulus (Supplementary Fig. 3a) following a consolidation phase (Supplementary Fig. 3b). Our results show that CTX engrams undergo maturation and HPC engrams are subject to de-maturation (Supplementary Fig. 3c) as evidenced by their respective t.p.r. curves (Supplementary Fig. 3d) in a manner analogous to the network trained on random non-overlapping stimuli (Fig. 1d, e). Notably, the f.p.r. of CTX engrams with overlapping stimuli does not settle at near-zero as it does with non-overlapping stimuli (compare Supplementary Fig. 3e and

Fig. 1f). This leads to differences in the CTX recall accuracy curve between the two stimulus conditions (compare Supplementary Fig. 3c and Fig. 1d). However, engram cells in CTX are initially silent and become active with consolidation in both cases (compare Supplementary Fig. 3d and Fig. 1e). Therefore, our model predicts that CTX engram maturation and HPC engram de-maturation also underlie systems consolidation with overlapping training stimuli but that in this case remote memory recall has a higher f.p.r. compared to training with non-overlapping stimuli. This prediction could be tested in future experiments investigating the effects of task similarity on the consolidation dynamics of multiple engrams encoding different tasks.

In addition, we perform ablation simulations and demonstrate that each Hebbian and non-Hebbian form of synaptic plasticity in our model is essential to reproduce engram dynamics observed in experiments (Supplementary Fig. 4). Specifically, we train our network on the original non-overlapping stimuli (Fig. 1a) following the same simulation protocol (Fig. 1b) but blocking the triplet STDP, heterosynaptic, and transmitter-induced forms of plasticity one at a time and verify that this disrupts engram dynamics in the network in each case. First, blocking triplet STDP leads to silent engram cells in both CTX and HPC at the end of training and after 12 h of consolidation (Supplementary Fig. 4a) because this prevents learning recurrent excitatory weights with a block diagonal structure (Supplementary Fig. 4b). As a result, although engram cells in CTX and HPC initially become active with consolidation, they eventually turn silent again due to indiscriminate potentiation of inhibitory synapses onto engram cells as evidenced by the t.p.r. in these regions (Supplementary Fig. 4a). This behavior was expected since in the absence of triplet STDP there is no Hebbian learning in the network and, hence, potentiation of excitatory synapses is non-specific (see Eq. (10a) in Methods). Second, blocking heterosynaptic plasticity prevents accurate memory recall in both CTX and HPC (Supplementary Fig. 4c). In this case, engram cells are active in both regions at the end of training and throughout consolidation but presenting partial cues of one stimulus leads to reactivation of the engrams encoding all stimuli as evidenced by the 100% t.p.r. and f.p.r. in these regions. This is a result of large and indiscriminate potentiation of recurrent excitatory weights in both CTX and HPC (Supplementary Fig. 4d). In turn, this effect is a consequence of potentiation due to triplet STDP being left unchecked in the absence of heterosynaptic plasticity (see Eq. (10b) in Methods). Third, blocking transmitter-induced plasticity prevents engram maturation in CTX because this disrupts engram reactivations in this region and as a result suppresses the potentiation of STIM → CTX synapses that would normally drive the maturation of CTX engram cells (Supplementary Fig. 4e). This is consistent with previous results showing that blocking transmitter-induced plasticity impairs engram reactivations[41]. Note that in this case some engram cell ensembles in CTX have strong inter-ensemble synaptic weights (Supplementary Fig. 4f). Importantly, the previous work that proposed combining the Hebbian and non-Hebbian forms of plasticity used in our model also showed that each is essential for stable memory formation and recall in a single-region spiking recurrent network[41]. Furthermore, a mean-field analysis incorporating this combination of plasticity mechanisms also supports their essential role for stable memory[41]. Thus, our ablation simulations show that each Hebbian and non-Hebbian form of plasticity in our model is crucial to reproduce experimentally-observed engram dynamics by impacting the evolution of engram cells in specific ways. Lastly, we conduct a sensitivity analysis that shows that the engram dynamics in our network are robust to changes in E/I ratio in CTX and HPC (Supplementary Fig. 5).

**Subcortical engram cells are essential for cortical engram maturation.** Despite replicating major engram cell dynamics in HPC and CTX, the network model in Fig. 1a cannot capture any interdependence between these regions. Given that it has been shown that the output of HPC engram cells after training is crucial for the maturation of CTX engram cells[33], we evaluated whether monosynaptic projections from HPC to CTX could reproduce this experimental finding. However, neither plastic (Supplementary Fig. 6) nor static (Supplementary Fig. 7) HPC → CTX synapses can replicate both cortical engram maturation and its reliance on hippocampal engram cells. Bidirectional HPC ↔ CTX synapses cannot reproduce these experimental findings either (Supplementary Fig. 8). To find a solution to this dilemma, we re-examined the brain regions that provide monosynaptic input to engram cells in CTX. Specifically, it has been reported that in CFC the ventral HPC (vHPC) is the only hippocampal area that has direct projections to CTX engram cells, but this amounts to only ~5% of their total monosynaptic input[33]. This led us to hypothesize that HPC engram cells use a multisynaptic pathway to CTX to support the maturation of its engrams.

In order to test this hypothesis, we include THL in our model since it simultaneously (I) receives input from HPC (via the medial temporal lobes: entorhinal and perirhinal cortices[46,47]), (II) has a large share of the monosynaptic projections to CTX engram cells in CFC (~20%)[33], (III) is essential for remote memory recall in CFC[48], and (IV) has increased activity around

hippocampal ripples coupled to spindles[49]—noting that the latter have a causal role in the systems consolidation of an episodic memory[30]. As a result, we expand the network with THL and set plastic and static circular receptive fields in STIM → HPC and STIM → THL, respectively (Fig. 2a). We then use a different set of stimuli for training (i.e., four non-overlapping horizontal bars) and testing (i.e., the central 50% of each bar) (Fig. 2a, see also Methods). In this network configuration, HPC and CTX are readout populations but not THL on its own. This means that memory recall from a partial cue is only considered successful if it can be retrieved in either HPC or CTX in a manner consistent with previous experiments[33]. We then set the learning rates ($\eta_{hpc \to thl}^{exc} = \eta_{thl}^{exc} = \eta_{hpc}^{exc}$ and $\eta_{thl \to ctx}^{exc} = \eta_{ctx}^{exc}$ with $\eta_{hpc}^{exc} > \eta_{ctx}^{exc}$) reflecting that subcortical synapses tend to change at a faster rate than their cortical counterparts. In line with our previous results (Fig. 1), STIM → HPC synapses have a longer synaptic consolidation time constant $\tau_{stim \to hpc}^{cons}$ relative to the other excitatory projections in the network (for further details, see Methods). Altogether, this network configuration allows us to evaluate whether the HPC → THL → CTX multisynaptic circuit can provide a pathway for HPC engram cells to support the maturation of CTX engrams.

We then subject the three-region network (Fig. 2a) to training, consolidation, and testing (Fig. 2b) and verify that it also exhibits de-maturation and maturation of engram cells in HPC and CTX, respectively (Fig. 2c). Hence, memory recall switches from HPC to CTX with consolidation (Supplementary Figs. 9 and 10) due to

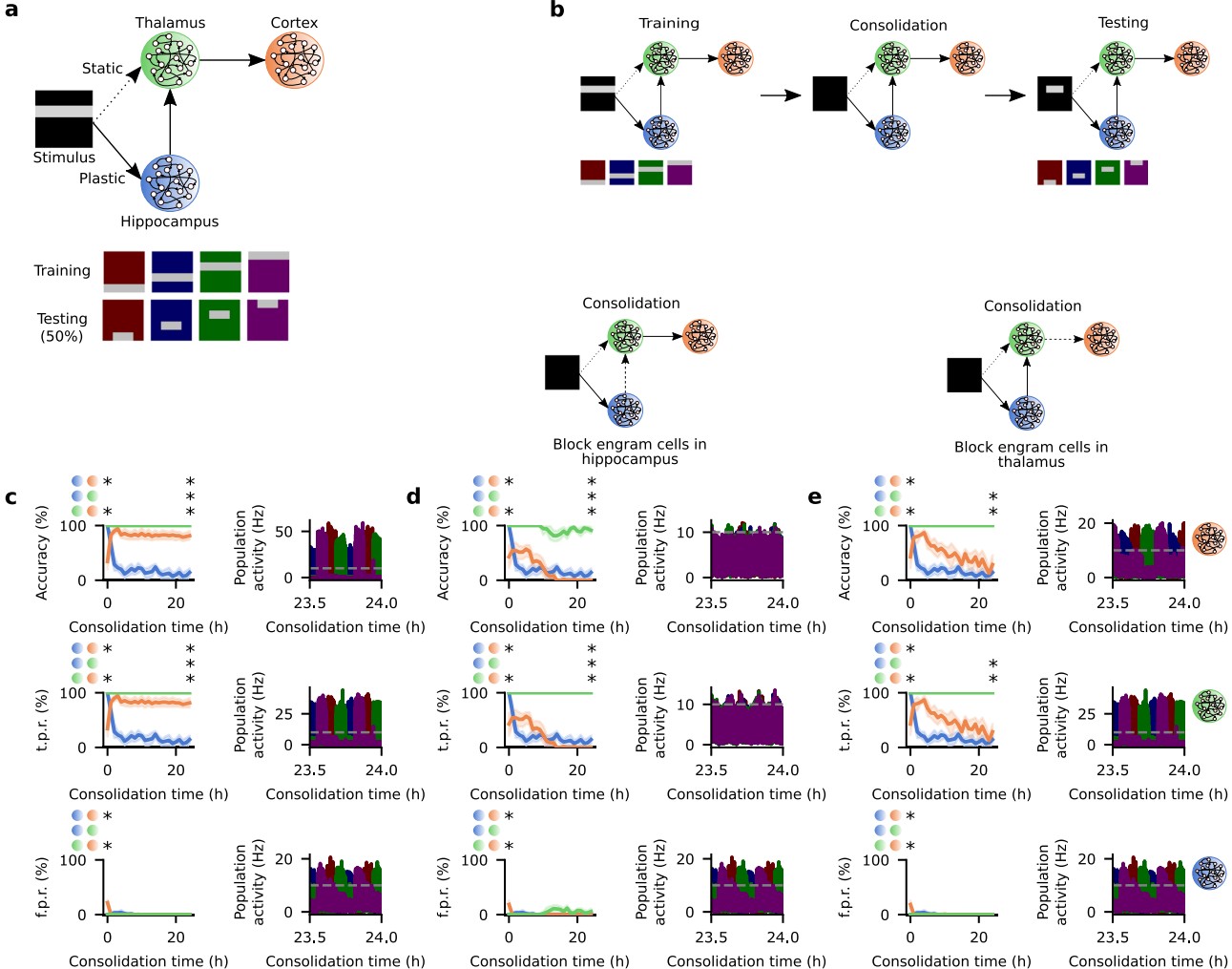

**Fig. 2 Hippocampal and thalamic engram cells are crucial for the maturation of cortical engrams. a** Schematic of network model with Stimulus (STIM), Hippocampus (HPC), Thalamus (THL), and Cortex (CTX) (top) and stimuli presented in the training phase with their respective partial cues used in the testing phase (bottom). STIM → THL synapses are static but the remaining feedforward projections are plastic. In STIM, training stimuli, and testing stimuli, light gray indicates active neurons firing above baseline whereas the remaining background color (i.e., black in STIM and red/blue/green/purple in training and testing stimuli) indicates neurons at baseline firing rate. Each background color in training and testing stimuli denotes a distinct non-overlapping stimulus and its respective partial cue. **b** Schematic of simulation protocol. **c** Memory recall in the testing phase as a function of consolidation time (left) and population activity of engram cells in the consolidation phase (right). Recall curves (top to bottom): accuracy, true positive rate, and false positive rate ($n = 5$ trials, mean values and 90% confidence intervals shown). For each recall metric, two-sided Mann–Whitney $U$ test. Accuracy: consolidation time $= 0$ (HPC vs. CTX $p$-value $= 7.494958 \times 10^{-3}$; HPC vs. THL $p$-value $= 0.920340$; THL vs. CTX $p$-value $= 7.494958 \times 10^{-3}$) and 24 h (HPC vs. CTX $p$-value $= 1.192523 \times 10^{-2}$; HPC vs. THL $p$-value $= 7.494958 \times 10^{-3}$; THL vs. CTX $p$-value $= 7.290358 \times 10^{-3}$). True positive rate: consolidation time $= 0$ (HPC vs. CTX $p$-value $= 7.494958 \times 10^{-3}$; HPC vs. THL $p$-value $= 0.920340$; THL vs. CTX $p$-value $= 7.494958 \times 10^{-3}$) and 24 h (HPC vs. CTX $p$-value $= 1.192523 \times 10^{-2}$; HPC vs. THL $p$-value $= 7.494958 \times 10^{-3}$; THL vs. CTX $p$-value $= 7.290358 \times 10^{-3}$). False positive rate: consolidation time $= 0$ (HPC vs. CTX $p$-value $= 7.290358 \times 10^{-3}$; HPC vs. THL $p$-value $= 0.920340$; THL vs. CTX $p$-value $= 7.290358 \times 10^{-3}$) and 24 h (HPC vs. CTX $p$-value $= 0.920340$; HPC vs. THL $p$-value $= 0.920340$; THL vs. CTX $p$-value $= 0.920340$). Population activity of engram cells (top to bottom): CTX, THL, and HPC (each color designates the engram cells encoding the respective stimulus in (**a**); dashed line indicates threshold $\zeta^{\mathrm{thr}} = 10$ Hz for engram cell activation). **d** Same as (**c**) but with the output of engram cells in HPC blocked during consolidation (recall curves: $n = 5$ trials, mean values and 90% confidence intervals shown). For each recall metric, two-sided Mann–Whitney $U$ test. Accuracy: consolidation time $= 0$ (HPC vs. CTX $p$-value $= 7.494958 \times 10^{-3}$; HPC vs. THL $p$-value $= 0.920340$; THL vs. CTX $p$-value $= 7.494958 \times 10^{-3}$) and 24 h (HPC vs. CTX $p$-value $= 1.192523 \times 10^{-2}$; HPC vs. THL $p$-value $= 7.494958 \times 10^{-3}$; THL vs. CTX $p$-value $= 7.290358 \times 10^{-3}$). True positive rate: consolidation time $= 0$ (HPC vs. CTX $p$-value $= 7.494958 \times 10^{-3}$; HPC vs. THL $p$-value $= 0.920340$; THL vs. CTX $p$-value $= 7.494958 \times 10^{-3}$) and 24 h (HPC vs. CTX $p$-value $= 1.192523 \times 10^{-2}$; HPC vs. THL $p$-value $= 7.494958 \times 10^{-3}$; THL vs. CTX $p$-value $= 7.290358 \times 10^{-3}$). False positive rate: consolidation time $= 0$ (HPC vs. CTX $p$-value $= 7.290358 \times 10^{-3}$; HPC vs. THL $p$-value $= 0.920340$; THL vs. CTX $p$-value $= 7.290358 \times 10^{-3}$) and 24 h (HPC vs. CTX $p$-value $= 0.920340$; HPC vs. THL $p$-value $= 0.920340$; THL vs. CTX $p$-value $= 0.920340$). **e** Same as (**c**) but with the output of engram cells in THL blocked during consolidation (recall curves: $n = 5$ trials, mean values and 90% confidence intervals shown). For each recall metric, two-sided Mann–Whitney $U$ test. Accuracy: consolidation time $= 0$ (HPC vs. CTX $p$-value $= 7.494958 \times 10^{-3}$; HPC vs. THL $p$-value $= 0.920340$; THL vs. CTX $p$-value $= 7.494958 \times 10^{-3}$) and 24 h (HPC vs. CTX $p$-value $= 0.342782$; HPC vs. THL $p$-value $= 7.494958 \times 10^{-3}$; THL vs. CTX $p$-value $= 7.494958 \cdot 10^{-3}$). True positive rate: consolidation time $= 0$ (HPC vs. CTX $p$-value $= 7.494958 \cdot 10^{-3}$; HPC vs. THL $p$-value $= 0.920340$; THL vs. CTX $p$-value $= 7.494958 \times 10^{-3}$) and 24 h (HPC vs. CTX $p$-value $= 0.342782$; HPC vs. THL $p$-value $= 7.494958 \times 10^{-3}$; THL vs. CTX $p$-value $= 7.494958 \times 10^{-3}$). False positive rate: consolidation time $= 0$ (HPC vs. CTX $p$-value $= 7.290358 \times 10^{-3}$; HPC vs. THL $p$-value $= 0.920340$; THL vs. CTX $p$-value $= 7.290358 \times 10^{-3}$) and 24 h (HPC vs. CTX $p$-value $= 0.920340$; HPC vs. THL $p$-value $= 0.920340$; THL vs. CTX $p$-value $= 0.920340$). *$p$-value $< 0.05$ (see Methods). **b–e** Color as in (**a**).

changes in engram cell state that are reflected in changes in t.p.r. This is consistent with previous findings[33] and is a result of region-specific plastic changes in feedforward afferent synapses: depression of STIM → HPC projections and potentiation of THL → CTX synapses (Supplementary Fig. 11) in a manner analogous to the two-region network (Fig. 1c). Additionally, engram cells in THL are initially active and remain so throughout the consolidation period in our simulations. Importantly, excitatory and inhibitory plasticity are required for proper engram dynamics (Supplementary Fig. 12).

We can also directly compare neural activation rates reported in previous experiments to those in our model (Supplementary Fig. 13). Specifically, previous experiments have subjected mice to CFC in Context A (training) and subsequently either placed them back in Context A or in a novel Context B (testing) after a delay period had elapsed[33] (Supplementary Fig. 13a, c). When mice were placed back in Context A, the fraction of activated engram and non-engram cells in medial prefrontal cortex did not differ in recent recall (i.e., delay of 1 day) but cortical engram cells had a higher activation rate in remote recall (i.e., delay of 12 days) (Supplementary Fig. 13a) whereas hippocampal engram cells had higher activation rates than non-engram cells in recent recall but not in remote recall (Supplementary Fig. 13c). In our model, engram and non-engram cells in CTX and HPC displayed analogous trends (compare Supplementary Fig. 13b–a and Supplementary Fig. 13d–c) but in an accelerated timescale (24 h in our model vs. 12 days in experiments, noting that this timescale in our model is subjective as it depends on the learning rate used in simulations). This observation indicates that cortical engram cells maturate while hippocampal engram cells de-maturate in our model in a manner consistent with the reported experimental findings[33]. We also note that there are a few differences between our model and the reported experimental data. First, hippocampal engram and non-engram

cells in our model exhibit comparable activation rates in remote recall while in experiments hippocampal engram cells have a lower activation rate than non-engram cells (compare Supplementary Fig. 13c, d). However, hippocampal engram cells are not robustly activated by partial cues in remote recall neither in our model nor in the reported experiments and, hence, they become silent in both cases. Second, engram and non-engram cells have comparable activation rates in experiments when testing is conducted with recall cues not present during training but in our model this is not always the case (compare Supplementary Figs. 13a–d). This is a result of strongly-potentiated inhibitory synapses onto engram and non-engram cells in our model and, consequently, it reflects the near-zero f.p.r. observed in our network (Fig. 2c). Lastly, activation rates of active engram cells in the reported experiments range roughly between 10 and 40% while in our model they vary from approximately 60–80% (compare Supplementary Fig. 13a–d). The low activation rates of active engram cells in experiments have been hypothesized to either be a consequence of imprecise tagging of active neurons or to reflect that engrams are dynamic with neurons "dropping into" or "dropping out of" the engram[31]. In our model, the relatively high activation rates of active engram cells reflect pattern completion. Taken together, the previous analysis shows that our model has engram cell dynamics consistent with previous experiments.

In addition, we find that the engram dynamics observed in HPC, THL, and CTX are accompanied by coupled engram reactivations across these three regions in our model (Fig. 2c and Supplementary Fig. 14a). Specifically, engrams encoding the same stimulus in two different regions have $|\mathrm{lag}_{\mathrm{max}}|$ between 0 and 140 ms whereas engrams encoding distinct stimuli exhibit $|\mathrm{lag}_{\mathrm{max}}|$ of at least 450.34 s. Therefore, engram reactivations in consolidation periods are coordinated throughout the network in our model. In the brain, the precise coupling of hippocampal sharp-wave ripples, thalamic

spindles, and cortical slow oscillations taking place during non-rapid-eye-movement (NREM) sleep has been shown to have a causal role in memory consolidation[30]. These cardinal rhythms are marked by characteristic oscillations in local field potential (LFP) that are thought to facilitate the coupling of engram reactivations across cortical and subcortical structures[50,51]. Importantly, it has been shown that memory engrams in HPC and CTX are reactivated mostly during NREM sleep[50] but the neural mechanisms through which engram reactivations can lead to systems consolidation remain elusive. Although our model is not designed to reproduce the stereotypical LFP oscillations associated with NREM sleep as we use point leaky integrate-and-fire neurons (see Methods) from which LFPs cannot be computed directly[52], our previous analysis shows that our network exhibits coupled engram reactivations throughout the HPC → THL → CTX circuit in line with the purported role of coupled LFP oscillations across these regions[50,51]. Critically, coupled engram reactivations in our model drive the de-maturation of HPC engram cells and the maturation of CTX engram cells by promoting the depression of feedforward synapses onto HPC engrams and the potentiation of feedforward projections onto CTX engrams over the course of consolidation (Supplementary Fig. 11). These opposing effects of engram reactivations on the dynamics of engram cells in HPC and CTX are a consequence of the distinct synaptic plasticity timescales in our network as previously discussed. Furthermore, in our model the presence of both coupled engram reactivations during consolidation and opposite engram cell responses in HPC and CTX during recall requires a multisynaptic circuit between HPC and CTX via THL (compare Supplementary Figs. 14a and 1). Thus, our model proposes neural mechanisms through which coupled engram reactivations can subserve systems consolidation of memory.

We also examine the behavior of extensions of the HPC → THL → CTX circuit in Fig. 2a to gain further insights into the functional roles of coupled engram reactivations in our model. First, a network with two cortical regions and reentrant connectivity exhibits analogous engram dynamics and reactivation patterns but with bidirectional multisynaptic coupling between hippocampal and cortical regions (Supplementary Fig. 15). It has been reported that a number of brain regions are engaged in the encoding, consolidation, and recall of memories[33,53] and our expanded network model suggests that coupled engram reactivations may be a generic mechanism used to coordinate activity among multiple brain regions to support these brain-wide mnemonic processes. Second, we probe the ability of our two- (Fig. 1a) and three-region (Fig. 2a) networks to support stable recall in a shared downstream readout region (RDT). We find that the extended two-region model has decoupled engram reactivations that disrupt recall in RDT (Supplementary Fig. 16) while the extended three-region circuit has coupled engram reactivations that enable stable recall in RDT throughout consolidation (Supplementary Fig. 17). This suggests that while the inclusion of THL as an intermediary region between HPC and CTX increases the complexity of the circuit underlying systems consolidation, it expands the computational functions supported by engrams in these regions through coordination of engram reactivations across the network. Notably, RDT in our extended three-region model (Supplementary Fig. 17) has parallels with the basolateral amygdala (BLA) in CFC: both receive projections from HPC and CTX and maintain active engram cells throughout consolidation[33].

We next probe the role of HPC engram cells in the maturation of CTX engrams. To that end, we block the output of engram cells in HPC during consolidation and subsequently test memory recall (Fig. 2d). Although recall accuracy in CTX initially shows a modest increase, it goes on to suffer a sharp decline and eventually settles at nearly zero. This is reflected in the CTX t.p.r.

curve which displays the same pattern. Hence, engram cells in HPC are crucial for the maturation of CTX engram cells in the three-region network and are consistent with previous findings[33]. Plotting the population activity of engram cells in the network reveals that the blockage of HPC engram cells disrupts the coupling of engram reactivations in the consolidation phase (Fig. 2d and Supplementary Fig. 18a). Recent experiments that tampered with the coupling of oscillations in HPC, THL, and CTX have demonstrated that this has an adverse effect on the systems consolidation of episodic memory[30,54]. Thus, our model suggests that engram cells in HPC are essential for the maturation of CTX engrams because they support coupling engram reactivations across brain regions throughout consolidation periods.

We then evaluate whether THL engram cells are also critical for CTX engram maturation. In a manner analogous to our previous probe of HPC engram cells, we block the output of engram cells in THL in the consolidation phase. The resulting memory recall curves show that THL engram cells are also crucial for the maturation of engram cells in CTX (Fig. 2e). The CTX recall accuracy curve in this case has a different shape though: at first accuracy increases substantially and reaches a level similar to the control network (Fig. 2c) before gradually declining and finally reaching a low point after more than 20 hours of consolidation (compare to Fig. 2d). This trend in recall accuracy is reflected in the CTX t.p.r. and suggests that THL engram cells are essential for stabilizing active engram cells in CTX. As expected, blocking the output of THL engram cells does not affect the coupling of reactivations in HPC and THL but it decouples reactivations in HPC and CTX and in THL and CTX (Fig. 2e and Supplementary Fig. 18b). Despite the differences in recall profile between blocking HPC and THL engram cells, both lead to decoupled engram reactivations in the network (Fig. 2d, e and Supplementary Fig. 18) due to disruption of the synaptic coupling in the HPC → THL → CTX circuit (Supplementary Fig. 19). Taken together, our modeling results predict that THL engram cells are essential for CTX engram cell maturation as they also support the coupling of subcortical–cortical engram reactivations in a manner similar to HPC engram cells.

**Inhibitory engram cells have distinct dynamics.** Engram cell experiments have focused on excitatory neurons given that expression of immediate early genes (IEGs) used for activity-dependent labeling occurs predominantly in these cells[35,55,56]. However, growing evidence suggests that inhibitory engram cells co-exist with excitatory engram cells[57,58]. We therefore also investigate the behavior of inhibitory neurons in our model. We start by comparing the recall profile of inhibitory and excitatory engram cells (Figs. 2c and 3a, respectively). The CTX recall accuracy of both sets of engram cells increases over the consolidation period but this rise is reflected in a decrease in f.p.r. in the case of inhibitory engrams while it is reflected in an increase in t.p.r. for excitatory engrams. Effectively, CTX inhibitory engram cells have a high t.p.r. post-training with only minor subsequent oscillations but CTX excitatory engrams have a flat near-zero f.p.r. throughout consolidation. Therefore, inhibitory engram cells in CTX become stimulus-specific with consolidation whereas excitatory engrams become active. The sharpening of the response of CTX inhibitory engrams can be attributed to potentiation of their inhibitory synapses onto excitatory engram cells in the consolidation period (Fig. 3b). In addition, the recall accuracy of THL inhibitory engram cells immediately after training is at 100% and then quickly decays to zero in association with a sharp uptake in f.p.r. while t.p.r. remains at 100%. This means that THL inhibitory engrams are continuously active after

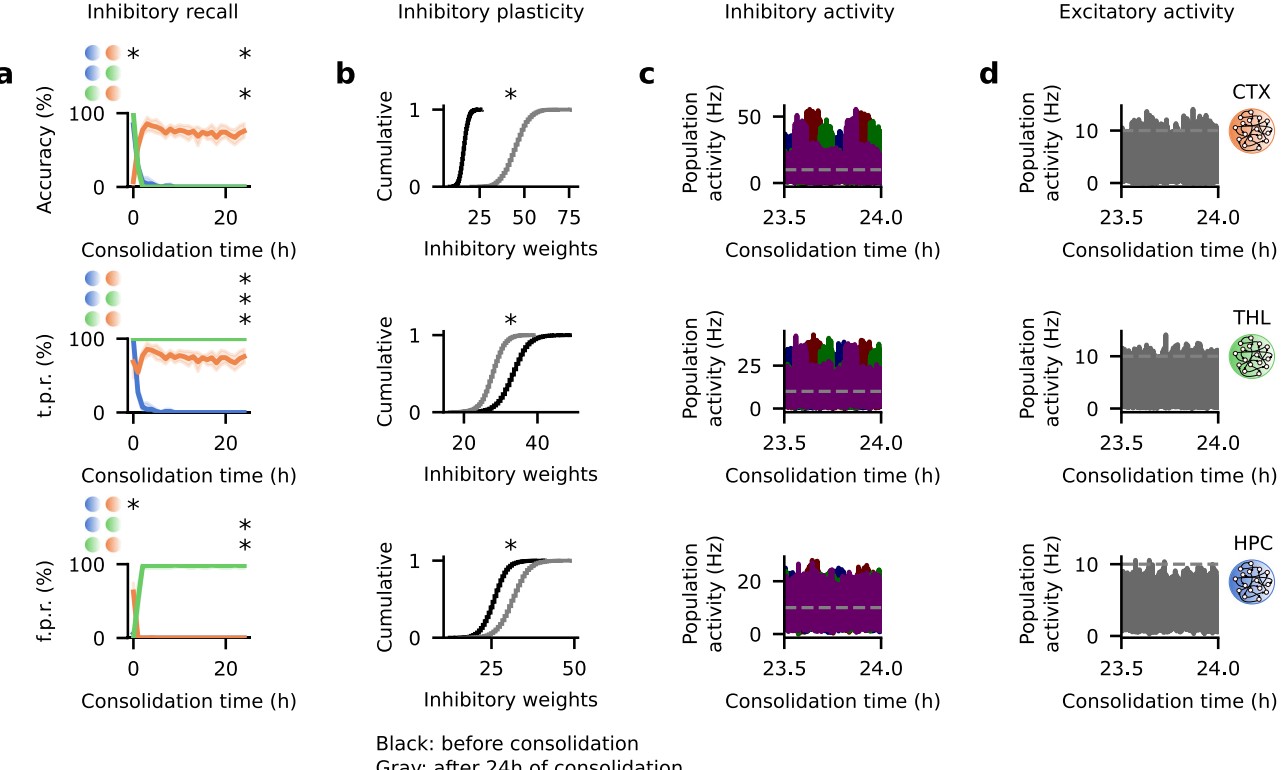

Black: before consolidation
Gray: after 24h of consolidation

**Fig. 3 Dynamics of inhibitory engram cells are region-specific. a–d** Analysis of engram dynamics in Fig. 2b. **a** Recall of inhibitory engram cells in HPC, THL, and CTX during the testing phase as a function of consolidation time. Top to bottom: accuracy, true positive rate, and false positive rate. Color as in Fig. 2a. $n = 5$ trials. Mean values and 90% confidence intervals shown. For each recall metric, two-sided Mann–Whitney $U$ test. Accuracy: consolidation time = 0 (HPC vs. CTX $p$-value = $1.166731 \times 10^{-2}$; HPC vs. THL $p$-value = $7.491290 \times 10^{-2}$; THL vs. CTX $p$-value = $8.968602 \times 10^{-2}$) and 24 h (HPC vs. CTX $p$-value = $7.290358 \times 10^{-3}$; HPC vs. THL $p$-value = 0.920340; THL vs. CTX $p$-value = $7.290358 \times 10^{-3}$). True positive rate: consolidation time = 0 (HPC vs. CTX $p$-value = $7.200566 \times 10^{-2}$; HPC vs. THL $p$-value = 0.920340; THL vs. CTX $p$-value = $7.200566 \times 10^{-2}$) and 24 h (HPC vs. CTX $p$-value = $7.290358 \times 10^{-3}$; HPC vs. THL $p$-value = $3.976752 \times 10^{-3}$; THL vs. CTX $p$-value = $7.290358 \times 10^{-3}$). False positive rate: consolidation time = 0 (HPC vs. CTX $p$-value = $1.218578 \times 10^{-2}$; HPC vs. THL $p$-value = $7.491290 \times 10^{-2}$; THL vs. CTX $p$-value = $9.469294 \times 10^{-2}$) and 24 h (HPC vs. CTX $p$-value = 0.920340; HPC vs. THL $p$-value = $3.976752 \times 10^{-3}$; THL vs. CTX $p$-value = $3.976752 \times 10^{-3}$). **b** Cumulative distribution of the total inhibitory synaptic weights onto individual excitatory engram cells at the end of training and after 24 h of consolidation. Top to bottom: CTX, THL, and HPC. Two-sided Kolmogorov–Smirnov test between the distribution of inhibitory weights onto excitatory engram cells at consolidation time = 0 and 24 h (CTX: $p$-value = 0; THL: $p$-value = 0; HPC: $p$-value = 0). **c** Population activity of inhibitory engram cells in the consolidation phase. Top to bottom: CTX, THL, and HPC. Each color designates the inhibitory engram cells encoding the respective stimulus in Fig. 2a. **d** Population activity of excitatory neurons in the consolidation phase. Top to bottom: CTX, THL, and HPC. *$p$-value < 0.05 (see Methods). **c, d** Dashed line indicates threshold $\zeta^{\text{thr}} = 10$ Hz for engram cell activation.

encoding but become progressively more unspecific to stimuli with consolidation as a result of depression of their inhibitory synapses projecting to excitatory engrams (Fig. 3b). This is in stark contrast to THL excitatory engrams which remain active and stimulus-specific after training. Furthermore, excitatory and inhibitory engram cells in HPC undergo de-maturation in a cascading manner: excitatory engram cells encoding a given stimulus become silent with consolidation and, consequently, the corresponding inhibitory engram cells are no longer activated by partial cues either. Note that de-maturation of HPC inhibitory engrams is accompanied by potentiation of their inhibitory synapses onto excitatory engram cells (Fig. 3b). Lastly, excitatory and inhibitory engram cells have different composition profiles (Supplementary Fig. 20). Overall, the dynamics of inhibitory engram cells in our model vary by brain region.

Inhibitory engram cells in HPC, THL, and CTX also have coupled reactivations in the consolidation phase (Fig. 3c and Supplementary Fig. 14b). Note that reactivations of inhibitory engrams in the three regions remain coupled despite the fact that the activity of HPC inhibitory engrams becomes gradually lower

as consolidation progresses. Comparing the activity of excitatory and inhibitory engram cells (Figs. 2c and 3c, respectively), we can see that excitatory and inhibitory engram reactivations are coupled across regions during consolidation (Supplementary Fig. 21). The activity of inhibitory engram cells combined with inhibitory synaptic plasticity (Fig. 3b, c) is able to tame the activity of excitatory neurons in each individual area (Fig. 3d and Supplementary Fig. 22) while still allowing coupled reactivations of excitatory engram cells. Taken together, our results predict that inhibitory engram cells have region-specific dynamics and that reactivations of inhibitory and excitatory engrams are coupled in consolidation periods.

**Inhibitory input is crucial for cortical engram dynamics.** We probe the role of region-specific inhibitory input in the engram dynamics throughout the network by blocking the output of inhibitory neurons during consolidation. Specifically, we first block the output of HPC inhibitory neurons in the consolidation phase and this disrupts CTX engram maturation and the coupling of engram reactivations in the HPC → THL → CTX circuit

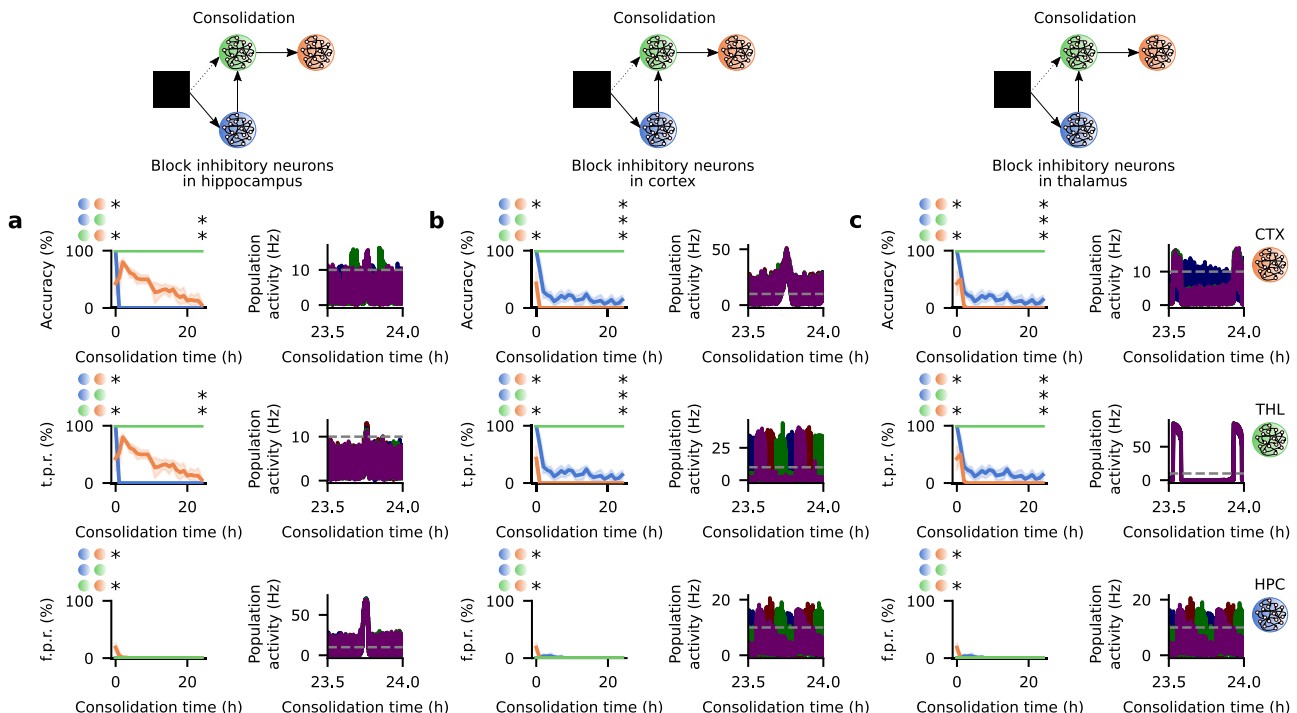

**Fig. 4 Inhibitory input is critical for cortical engram maturation. a** Memory recall as a function of consolidation time (left) and population activity of excitatory engram cells in the consolidation phase (right) when inhibitory neurons in HPC are blocked in the consolidation phase of the protocol in Fig. 2b. Recall curves (top to bottom): accuracy, true positive rate, and false positive rate ($n = 5$ trials, mean values and 90% confidence intervals shown). For each recall metric, two-sided Mann–Whitney $U$ test. Accuracy: consolidation time $= 0$ (HPC vs. CTX $p$-value $= 7.494958 \times 10^{-3}$; HPC vs. THL $p$-value $= 0.920340$; THL vs. CTX $p$-value $= 7.494958 \times 10^{-3}$) and 24 h (HPC vs. CTX $p$-value $= 0.177016$; HPC vs. THL $p$-value $= 3.976752 \times 10^{-3}$; THL vs. CTX $p$-value $= 6.501702 \times 10^{-3}$). True positive rate: consolidation time $= 0$ (HPC vs. CTX $p$-value $= 7.494958 \times 10^{-3}$; HPC vs. THL $p$-value $= 0.920340$; THL vs. CTX $p$-value $= 7.494958 \times 10^{-3}$) and 24 h (HPC vs. CTX $p$-value $= 0.177016$; HPC vs. THL $p$-value $= 3.976752 \times 10^{-3}$; THL vs. CTX $p$-value $= 6.501702 \times 10^{-3}$). False positive rate: consolidation time $= 0$ (HPC vs. CTX $p$-value $= 7.290358 \times 10^{-3}$; HPC vs. THL $p$-value $= 0.920340$; THL vs. CTX $p$-value $= 7.290358 \times 10^{-3}$) and 24 h (HPC vs. CTX $p$-value $= 0.920340$; HPC vs. THL $p$-value $= 0.920340$; THL vs. CTX $p$-value $= 0.920340$). Population activity of excitatory engram cells (top to bottom): CTX, THL, and HPC (each color designates the engram cells encoding the respective stimulus in Fig. 2a; dashed line indicates threshold $\zeta^{thr} = 10$ Hz for engram cell activation). **b** Same as (**a**) but with inhibitory neurons in CTX blocked during consolidation (recall curves: $n = 5$ trials, mean values and 90% confidence intervals shown). For each recall metric, two-sided Mann–Whitney $U$ test. Accuracy: consolidation time $= 0$ (HPC vs. CTX $p$-value $= 7.494958 \times 10^{-3}$; HPC vs. THL $p$-value $= 0.920340$; THL vs. CTX $p$-value $= 7.494958 \times 10^{-3}$) and 24 h (HPC vs. CTX $p$-value $= 2.536986 \times 10^{-2}$; HPC vs. THL $p$-value $= 7.494958 \times 10^{-3}$; THL vs. CTX $p$-value $= 3.976752 \times 10^{-3}$). True positive rate: consolidation time $= 0$ (HPC vs. CTX $p$-value $= 0.494958 \times 10^{-3}$; HPC vs. THL $p$-value $= 0.920340$; THL vs. CTX $p$-value $= 7.494958 \times 10^{-3}$) and 24 h (HPC vs. CTX $p$-value $= 2.536986 \times 10^{-2}$; HPC vs. THL $p$-value $= 7.494958 \times 10^{-3}$; THL vs. CTX $p$-value $= 3.976752 \times 10^{-3}$). False positive rate: consolidation time $= 0$ (HPC vs. CTX $p$-value $= 7.290358 \times 10^{-3}$; HPC vs. THL $p$-value $= 0.920340$; THL vs. CTX $p$-value $= 7.290358 \times 10^{-3}$) and 24 h (HPC vs. CTX $p$-value $= 0.920340$; HPC vs. THL $p$-value $= 0.920340$; THL vs. CTX $p$-value $= 0.920340$). **c** Same as (**a**) but with inhibitory neurons in THL blocked during consolidation (recall curves: $n = 5$ trials, mean values and 90% confidence intervals shown). For each recall metric, two-sided Mann–Whitney $U$ test. Accuracy: consolidation time $= 0$ (HPC vs. CTX $p$-value $= 7.494958 \times 10^{-3}$; HPC vs. THL $p$-value $= 0.920340$; THL vs. CTX $p$-value $= 7.494958 \times 10^{-3}$) and 24 h (HPC vs. CTX $p$-value $= 2.536986 \times 10^{-2}$; HPC vs. THL $p$-value $= 7.494958 \times 10^{-3}$; THL vs. CTX $p$-value $= 3.976752 \times 10^{-3}$). True positive rate: consolidation time $= 0$ (HPC vs. CTX $p$-value $= 7.494958 \times 10^{-3}$; HPC vs. THL $p$-value $= 0.920340$; THL vs. CTX $p$-value $= 7.494958 \times 10^{-3}$) and 24 h (HPC vs. CTX $p$-value $= 2.536986 \times 10^{-2}$; HPC vs. THL $p$-value $= 7.494958 \times 10^{-3}$; THL vs. CTX $p$-value $= 3.976752 \times 10^{-3}$). False positive rate: consolidation time $= 0$ (HPC vs. CTX $p$-value $= 7.290358 \times 10^{-3}$; HPC vs. THL $p$-value $= 0.920340$; THL vs. CTX $p$-value $= 7.290358 \times 10^{-3}$) and 24 h (HPC vs. CTX $p$-value $= 0.920340$; HPC vs. THL $p$-value $= 0.920340$; THL vs. CTX $p$-value $= 0.920340$). The white space above the horizontal axis in the activity plots in (**a**–**c**) indicates that the population activity of engrams is continuously above 0 for a short period even when it is computed every 10 ms without smoothing or convolution (see Methods). This is a consequence of blocking inhibitory neurons: excitatory engram cells exhibit periods of very high activity in the absence of inhibition due to recurrent excitatory interactions. *$p$-value $< 0.05$ (see Methods). **a**–**c** Network and simulation parameters are the same as in Fig. 2a except for blocked inhibitory neurons during consolidation. Color as in Fig. 2a.

(Fig. 4a and Supplementary Fig. 23a). Blocking inhibitory neurons in CTX also tampers with CTX engram dynamics and the subcortical-cortical coupling of engram reactivations (Fig. 4b and Supplementary Fig. 23b). These results are aligned with previous findings that showed that blocking parvalbumin-positive interneurons either in HPC or CTX decoupled oscillations in these regions in consolidation periods and disrupted systems consolidation[54]. We then block inhibitory neurons in THL and this also prevents engram cell maturation in CTX and the coupling of engram reactivations in the network (Fig. 4c and

Supplementary Fig. 23c). In each of the previous simulations, blocking inhibitory input to one region significantly alters the dynamics of engrams in that region and in any downstream areas (Fig. 4a–c) as inhibitory drive is essential for the consolidation of subcortical-cortical synaptic coupling (Supplementary Fig. 24). Our model then predicts that inhibitory input to HPC, CTX, and THL is essential for CTX engram maturation by coupling engram reactivations in consolidation periods. Note that our simulations with blockage of inhibitory neurons are motivated by previous experiments that blocked parvalbumin-positive interneurons

irrespective of how they responded to the conditioning stimulus[54]. We also examine the effects of blocking exclusively inhibitory engram cells in each region of our model. Interestingly, we find that this does not prevent CTX engram maturation because the remaining unblocked inhibitory neurons are still able to support coupled engram reactivations due to strong potentiation of their inhibitory synapses onto excitatory engram cells (Supplementary Fig. 25). This suggests that inhibitory engram cells may be replaced by inhibitory neurons that were originally unresponsive to the training stimuli as long as their inhibitory synapses can be sufficiently potentiated. Given that blocking excitatory engram cells in HPC and THL (Fig. 2d, e) and blocking inhibitory input to HPC, CTX, and THL (Fig. 4a–c) both decouple engram reactivations across these regions (Supplementary Figs. 18 and 23) and disrupt CTX engram maturation, our results suggest that coupled engram reactivations across the HPC → THL → CTX circuit underlie engram dynamics that mediate systems consolidation.

**Thalamocortical coupling underlies retrograde amnesia profiles**. We then investigate to what extent memory recall relies on HPC over time by examining retrograde amnesia patterns induced by HPC ablation. Ablation of HPC in the testing phase (Fig. 5a) leads to significant impairment in recent recall (Fig. 5b) since after the HPC lesion recall relies exclusively on CTX. This was expected given that recent recall originally relied on HPC (Fig. 2c). However, remote recall is virtually not affected by HPC ablation (Fig. 5b). Therefore, memory recall reliance on HPC is time-dependent and the model exhibits a temporally-graded retrograde amnesia curve[4].

We next probe the effect of THL → CTX synaptic coupling on HPC reliance by varying the plasticity rate of these synapses. Specifically, we explore how heterosynaptic plasticity strength $\beta_{THL \to CTX}$ can increase or decrease coupling between THL and CTX at the end of encoding and the resulting effect on memory recall. First, we increase $\beta_{THL \to CTX}$ substantially (Fig. 5c–f) and this severely impairs the ability of THL → CTX synapses to potentiate (see Methods). As a result, THL and CTX are only weakly coupled at the end of the training phase (Fig. 5c, compare to Supplementary Fig. 19a) and remain so despite subsequent consolidation (Fig. 5d). Accordingly, remote recall with the intact control network (Fig. 2a) is lost since the weak coupling of THL → CTX synapses prevents CTX engram maturation and HPC engram cells still become silent (Fig. 5e). Naturally, HPC ablation does not improve remote recall and it also prevents recent memory retrieval (Fig. 5f). Thus, the network with weakly coupled THL → CTX synapses relies exclusively on HPC for memory recall and displays a flat retrograde amnesia pattern[4].

Subsequently, we reduce $\beta_{THL \to CTX}$ to effectively enable faster synaptic potentiation (see Methods) and, consequently, increase THL → CTX synaptic coupling at the end of encoding (Fig. 5g, compare to Supplementary Fig. 19a). Coupling between these regions is reinforced with consolidation (Fig. 5h) and CTX recall accuracy is therefore extremely high both immediately following training and throughout consolidation (Fig. 5i). Ablating HPC has a negligible effect on CTX recall and, hence, memories can be recalled independently of HPC (Fig. 5j). This network configuration then exhibits an absent retrograde amnesia curve[4]. We also examine the effect of ablating HPC either at the very beginning of the consolidation phase or only after 12 hours of consolidation and we observe the same relationship between THL → CTX synaptic coupling and the pattern of retrograde amnesia exhibited by the network (Supplementary Fig. 26). Altogether, our model predicts that the degree of THL → CTX synaptic coupling at the end of encoding is a major driver of the

ensuing CTX engram cell dynamics and the associated retrograde amnesia profile induced by HPC ablation.

**Discussion**
Our model is able to reproduce key experimental findings associated with systems consolidation. Specifically, it captures engram cell maturation and de-maturation in CTX and HPC, respectively, and the crucial role that HPC engram cells have in the maturation of CTX engrams[33]. The model also reflects the causal role of coupled oscillations across HPC, THL, and CTX in the systems consolidation of episodic memory[30] and connects it to the associated engram dynamics observed in experiments[33]. We have demonstrated that these experimental findings can be reproduced in a computational model of the HPC → THL → CTX multisynaptic pathway with region-specific synaptic plasticity rates. Our results suggest that the timescale of synaptic plasticity is precisely conducted across brain regions to enable coordinated HPC–THL–CTX communication and that these concerted subcortical–cortical interactions are vital for engram dynamics behind systems consolidation of memory.

The timescales of the various forms of synaptic plasticity in our model need to be coordinated to reproduce specific engram cell state transitions taking place in parallel. The learning rate of the triplet STDP is higher in subcortical regions (i.e., HPC and THL) relative to CTX consistent with the view that subcortical synapses tend to be more plastic than cortical ones. However, synaptic consolidation is slower in HPC compared to THL and CTX in line with the observation that HPC engram cells are less stable and, hence, more prone to becoming silent. Interestingly, it has been previously suggested that synaptic consolidation has an active role in systems consolidation[59]. Transmitter-induced plasticity rates are scaled linearly to the learning rate of each individual region to prevent long-term depression (LTD) from making the network silent while the timescales of heterosynaptic plasticity are set to avoid excessive network activity while still allowing long-term potentiation (LTP) to take place. This combination of Hebbian (triplet STDP) and non-Hebbian (heterosynaptic and transmitter-induced) plasticity has been shown to enable stable memory formation and recall in a single-region spiking neural network model[41] and here we build on those results to show that coordinated synaptic plasticity timescales across brain regions can extend the mnemonic functions supported by these forms of plasticity.

There are multiple circuits that can potentially be used by HPC to support the maturation of CTX engram cells but we include the HPC → THL → CTX pathway in our model. As noted earlier, the inclusion of THL is motivated by its afferent and efferent projections (i.e., HPC and CTX, respectively[46,47,60]) and the observation that in CFC this region is both responsible for ~20% of the monosynaptic input to CTX engram cells[33] and crucial for remote recall[48]. Here, we assume that THL also has an essential role in the remote recall of other types of episodic memories in a similar way as it has been proposed that engram cell dynamics observed in CFC are present in generic episodic memories[5] — a view that is consistent with numerous reports of memory impairments in a wide range of tasks following lesions to THL[46,47]. Furthermore, the increased THL activity around hippocampal ripples coupled to spindles[49] suggests that this region may play a part in the essential role that spindles have in coupling cortical, thalamic, and hippocampal oscillations in systems consolidation[30]. Thus, the HPC → THL → CTX circuit seems to be a prime candidate for having a crucial role in the maturation of CTX engram cells and our modeling results support this view. Nevertheless, we cannot exclude the possibility that alternative intermediary regions may also be recruited by HPC to mediate

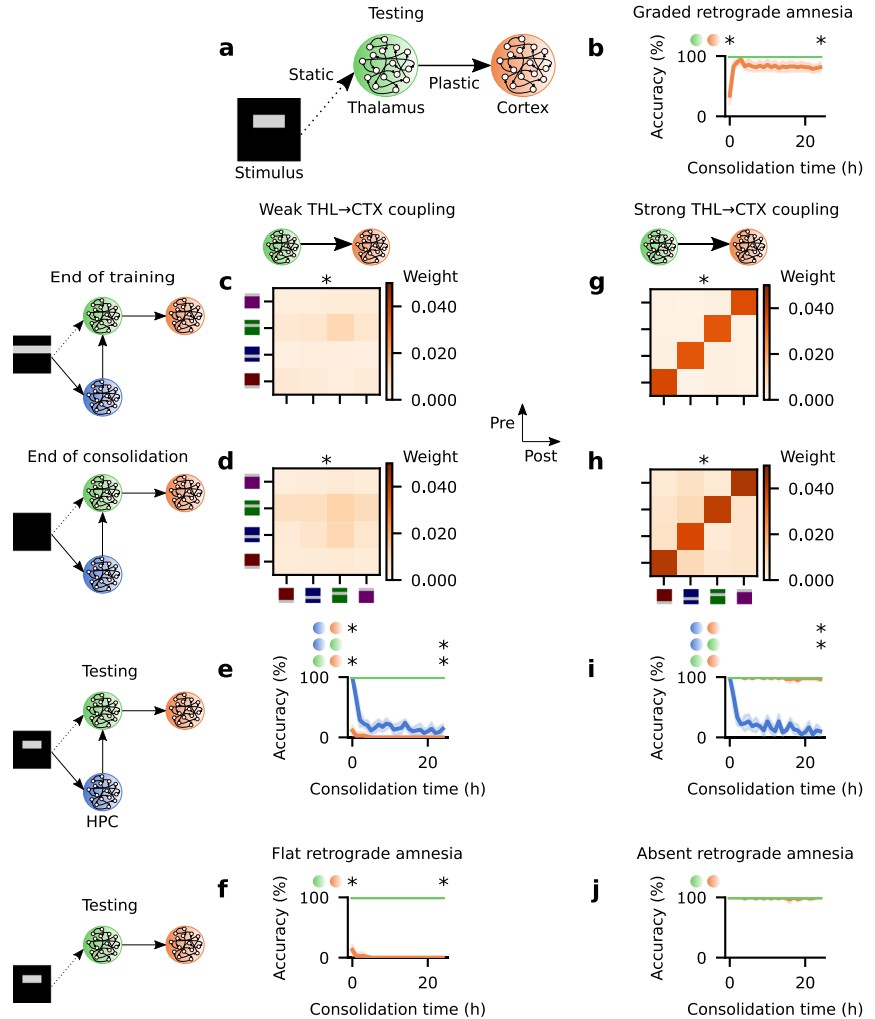

**Fig. 5 Thalamocortical coupling at the end of encoding underlies retrograde amnesia pattern. a** Schematic of network model with ablation of HPC at testing time. **b** Memory recall accuracy as a function of consolidation time with HPC ablation in the testing phase (as in **a**) of the protocol shown in Fig. 2b. Two-sided Mann–Whitney $U$ test between accuracy in THL and CTX at consolidation time $= 0$ ($p$-value $= 7.088721 \times 10^{-3}$) and 24 h ($p$-value $= 2.480842 \times 10^{-2}$). **c–f** Simulation with weakly coupled THL $\rightarrow$ CTX. **c** Mean weight strength of THL $\rightarrow$ CTX synapses clustered according to engram cell preference at the end of training. Two-sided Kolmogorov–Smirnov test between the distribution of THL $\rightarrow$ CTX weights among engram cells encoding the same stimulus (i.e., diagonal) and that of THL $\rightarrow$ CTX weights among engram cells encoding different stimuli (i.e., off-diagonal): $p$-value $= 1.659955 \times 10^{-151}$. **d** Mean weight strength of THL $\rightarrow$ CTX synapses clustered according to engram cell preference after 24 h of consolidation. Two-sided Kolmogorov–Smirnov test between the distribution of THL $\rightarrow$ CTX weights among engram cells encoding the same stimulus (i.e., diagonal) and that of THL $\rightarrow$ CTX weights among engram cells encoding different stimuli (i.e., off-diagonal): $p$-value $= 3.599719 \times 10^{-10}$. **e** Memory recall accuracy as a function of consolidation time with the intact network in the protocol depicted in Fig. 2b. Two-sided Mann–Whitney $U$ test at consolidation time $= 0$ (HPC vs. CTX $p$-value $= 5.583617 \times 10^{-3}$; HPC vs. THL $p$-value $= 0.920340$; THL vs. CTX $p$-value $= 5.583617 \times 10^{-3}$) and 24 h (HPC vs. CTX $p$-value $= 0.193816$; HPC vs. THL $p$-value $= 7.290358 \times 10^{-3}$; THL vs. CTX $p$-value $= 5.583617 \times 10^{-3}$). **f** Memory recall accuracy as a function of consolidation time with HPC ablation in the testing phase (as in **a**) of the protocol shown in Fig. 2b. Two-sided Mann–Whitney $U$ test at consolidation time $= 0$ (THL vs. CTX $p$-value $= 5.583617 \times 10^{-3}$) and 24 h (THL vs. CTX $p$-value $= 5.583617 \times 10^{-3}$). **g–j** Simulation with strongly coupled THL $\rightarrow$ CTX. **g** Mean weight strength of THL $\rightarrow$ CTX synapses clustered according to engram cell preference at the end of training. Two-sided Kolmogorov–Smirnov test between the distribution of THL $\rightarrow$ CTX weights among engram cells encoding the same stimulus (i.e., diagonal) and that of THL $\rightarrow$ CTX weights among engram cells encoding different stimuli (i.e., off-diagonal): $p$-value $= 0$. **h** Mean weight strength of THL $\rightarrow$ CTX synapses clustered according to engram cell preference after 24 h of consolidation. Two-sided Kolmogorov–Smirnov test between the distribution of THL $\rightarrow$ CTX weights among engram cells encoding the same stimulus (i.e., diagonal) and that of THL $\rightarrow$ CTX weights among engram cells encoding different stimuli (i.e., off-diagonal): $p$-value $= 0$. **i** Memory recall accuracy as a function of consolidation time with the intact network in the protocol depicted in Fig. 2b. Two-sided Mann–Whitney $U$ test at consolidation time $= 0$ (HPC vs. CTX $p$-value $= 0.920340$; HPC vs. THL $p$-value $= 0.920340$; THL vs. CTX $p$-value $= 0.920340$) and 24 h (HPC vs. CTX $p$-value $= 1.066227 \times 10^{-2}$; HPC vs. THL $p$-value $= 7.290358 \times 10^{-3}$; THL vs. CTX $p$-value $= 0.177809$). **j** Memory recall accuracy as a function of consolidation time with HPC ablation in the testing phase (as in **a**) of the protocol shown in Fig. 2b. Two-sided Mann–Whitney $U$ test at consolidation time $= 0$ (THL vs. CTX $p$-value $= 0.920340$) and 24 h (THL vs. CTX $p$-value $= 0.920340$). *$p$-value $< 0.05$ (see Methods). **b–j** Color as in Fig. 2a. **b**, **e–f**, **i**, **j** $n = 5$ trials. Mean values and 90% confidence intervals shown.

systems consolidation in CTX. In fact, three other brain regions have monosynaptic projections to CTX engram cells to a similar extent as THL in CFC: anterodorsal thalamus (ADT), medial entorhinal cortex layer Va (MEC-Va), and BLA[33]. Note, however, that (I) ADT is only essential for recent but not for remote CFC memory recall[48], (II) MEC-Va → CTX is not required for neither recent nor remote recall in CFC[33], and (III) BLA → MEC stimulation improved retention of the contextual but not foot shock components of memory in CFC[61]. In addition, a multisynaptic pathway involving the dorsoventral axis of HPC may also be used by HPC engram cells to support engram dynamics in CTX given that the dorsal HPC (dHPC) has a critical role in CFC[62,63]. Hence, dHPC → vHPC → CTX may be recruited by HPC engrams but as noted earlier in CFC only ~5% of the total monosynaptic input to CTX engram cells originates in vHPC[33]. Further, although another possible circuit may involve dHPC and retrosplenial cortex (RSC) (i.e., dHPC → RSC → CTX), RSC projections only account for less than 10% of the monosynaptic input to CTX engram cells in CFC[33]. Altogether, these findings pose HPC → THL → CTX as a plausible minimal circuit for the encoding, consolidation, and recall of episodic memory and our simulation results support this viewpoint. Moreover, we assign the roles of CTX, HPC, and THL to the three recurrent neural networks (RNNs) in our model in an attempt to match engram dynamics in each region to experimental reports. However, our modeling results are effectively agnostic to the identity of each RNN and, therefore, can be interpreted in the context of a generic circuit that exhibits multiple, region-specific engram dynamics.

The mechanisms through which HPC engram cells support the maturation of CTX engrams remain unknown but our modeling results suggest that HPC engrams are essential for coupling hippocampal, thalamic, and cortical engram reactivations and thereby are crucial for CTX engram cell maturation. The causal role of coupled HPC–THL–CTX oscillations in the systems consolidation of episodic memories has been previously demonstrated[30,54] and our model predicts that HPC engram cells have themselves a causal role in this coupling. Although engram cells have been found in various thalamic nuclei[53], the potential role that thalamic engram cells have in systems consolidation is not known either. Our simulations suggest that engram cells in THL are also crucial for the maturation of CTX engram cells by coupling engram reactivations across HPC, THL, and CTX. Importantly, two lines of evidence support the view that engrams are present in THL: (I) a large body of THL lesion studies that showed post-lesion memory deficits in a diverse array of tasks[46,47,64]; and (II) a recent experiment that found THL to be one of the regions with a high probability of holding an engram in a brain-wide mapping of CFC memory[53]. Furthermore, our model predicts that THL engram cells are active in both recent and remote recall in a similar way as BLA engram cells in CFC[33].

Our model also aims to shed light on the dynamics of inhibitory engram cells. "Inhibitory replicas" of learning-induced excitatory connectivity patterns have been found[58,65], but the behavior of inhibitory engram cells has not been probed yet. Our model predicts that the dynamics of inhibitory engrams are region-specific: CTX inhibitory engram cells are active in recent and remote recall but become more selective to stimuli over time, THL inhibitory engrams also maintain an active status after training but rather become unspecific to stimuli, and HPC inhibitory engram cells undergo de-maturation similarly to their excitatory counterparts. Inhibitory engrams are formed in our model via the potentiation of inhibitory synapses onto excitatory cells that display learning-induced activity increase in line with other computational models[57,65]. Furthermore, our results also predict coupled reactivations of excitatory and inhibitory engram cells in HPC, THL, and CTX. Previously, inhibitory neurons were shown to control the size

of excitatory engram cell ensembles[66,67] and to mediate memory discrimination[68]. Here, we suggest that interneurons also undergo learning-induced changes akin to excitatory neurons (i.e., inhibitory engrams). Moreover, blocking inhibitory neurons in HPC and CTX disrupts systems consolidation in our simulations by preventing coupled reactivations between these regions. This is consistent with the crucial role of inhibitory neurons in coupling CTX spindles and HPC ripples[54]. Our model also predicts that inhibitory input to THL has a similar critical role in coupling engram reactivations across subcortical and cortical regions. Importantly, local THL inhibitory neurons are present in primates but not in lower species such as rodents[47]. However, the thalamic reticular nucleus (TRN) provides robust inhibitory input to THL across species via GABAergic projections[47,69]. For those species that rely exclusively on TRN for inhibitory control of network activity, TRN inhibitory neurons may play an analogous role to that of local THL interneurons in higher species given that (I) TRN was shown to have a high probability of holding engram cells in a brain-wide mapping of CFC in rodents[53] and (II) TRN has an active role in the generation of thalamocortical oscillatory rhythms[69,70]. Altogether, our results suggest that inhibitory neurons in distributed brain regions have a crucial role in the coordination of HPC–THL–CTX communication mediated by engram reactivations.

We also investigate how recent and remote recall rely on HPC by reproducing the different patterns of retrograde amnesia induced by HPC damage: temporally-graded, flat, and absent[4]. Our model predicts that the degree of THL → CTX synaptic coupling at the end of encoding is predictive of the subsequent CTX engram cell dynamics (i.e., silent or active at recent and remote recall) and the corresponding retrograde amnesia profile caused by HPC lesions. Our model then also predicts that silent CTX engram cells are the basis of retrograde amnesia induced by HPC damage. This is consistent with protein synthesis inhibitor-induced retrograde amnesia studies that showed that silent HPC engram cells underlie this form of amnesia and that their afferent synapses from upstream engram cells exhibit reduced potentiation relative to active engram cells in healthy mice[71,72]. Furthermore, the discovery of a rapidly-encoded engram in human posterior parietal cortex[73] suggests the existence of cortical engram cells that are active in recent and remote recall as predicted by our model. Taken together, our modeling results predict that distinct engram cell dynamics underlie specific patterns of retrograde amnesia induced by HPC damage and, thus, provide a mechanistic account to reconcile seemingly conflicting reports in the HPC lesion literature[4,5].

Our model makes several testable predictions. First, our results predict that engram cells in THL are active in recent and remote recall and are crucial for the maturation of engram cells in CTX. This could be tested by labeling THL engram cells during encoding and subsequently blocking their output in a manner analogous to previous protocols[33]. Second, our model predicts that engram cells in HPC and THL are essential for coupling engram reactivations in HPC, THL, and CTX in consolidation periods. Blocking separately HPC and THL engram cells after training[33] and measuring the degree of coupling of reactivations in HPC–THL–CTX during sleep[30] could test this prediction. Third, our results suggest that inhibitory engram cells with region-specific dynamics and coupled reactivations co-exist with excitatory engrams in subcortical and cortical regions. Although engram cell studies have focused on excitatory neurons due to their increased IEG expression[35,55,56] as discussed previously, inhibitory neurons can also up-regulate IEGs (e.g., c-fos and Arc) under strong stimulation[35,74,75]. Therefore, our predictions regarding inhibitory engram cells could also be tested by modifying parameters of existing experiments[33] to induce reliable IEG expression in inhibitory neurons and, hence, enable labeling of

inhibitory engram cells. Fourth, our model predicts that inhibitory input to THL is critical for CTX engram maturation by coupling engram reactivations in subcortical-cortical circuits. This prediction could be tested by extending current engram cell protocols[33] with chemogenetic techniques already used to block interneurons in HPC and CTX[54] but applying them to inhibitory neurons projecting to THL. Fifth, our model suggests that the degree of synaptic coupling in THL → CTX at the end of encoding is predictive of CTX engram dynamics and the resulting pattern of retrograde amnesia induced by HPC damage. These predictions could also be tested by incorporating activity-dependent cell labeling—combined with strategies for circuit-specific manipulations and in vivo calcium imaging—to existing HPC lesion protocols. THL → CTX synaptic coupling could potentially be either decreased by applying protein synthesis inhibitors to CTX[71,72] or increased by extending total training time and/or stimulus exposure.

Despite capturing engram cell dynamics in HPC and CTX and the coupling of engram reactivations in the HPC → THL → CTX circuit, our model has several limitations. First, systems consolidation takes place over days, months, or even years after memory acquisition[5] but our simulations extend for only 24 h after training. Although the synaptic plasticity rates in our model could conceivably be reduced to match more realistic timescales of systems consolidation, this would immensely increase the computational cost of simulations and, consequently, it would be impractical to simulate multi-region large-scale networks like ours for long periods. Note, however, that our simulation results are consistent with experimental evidence regarding engram maturation and de-maturation with systems consolidation[33] independent of the exact timing of engram cell state transitions. Second, our model is not specifically designed to reproduce hippocampal sharp-wave ripples, thalamic spindles, and cortical slow oscillations as discussed previously. However, these LFP oscillations have been thought to facilitate the coupling of engram reactivations across regions in the brain[50,51] and recent evidence has supported this view[76]. Our model then reproduces coupled engram reactivations and proposes neural mechanisms through which they mediate systems consolidation of memory. Therefore, coupled engram reactivations in our model serve the same purported functional role as coupled engram reactivations in the brain orchestrated by LFP oscillations. Third, we do not attempt to model a gradual shift in memory from episodic (i.e., specific and detail-rich) to semantic (i.e., abstract and gist-like) over systems consolidation. The extent of such change is still an open question[5] and is beyond the scope of the present study. Fourth, we have assumed that STIM → THL synapses only change over developmental timescales that are much longer than those captured by our model and, hence, these synapses are static in our simulations. We then use non-random stimuli in the three-region model and initialize STIM → THL synapses with circular receptive fields in order to facilitate learning in THL (see Methods). Although in theory the engram dynamics exhibited by our current model could be reproduced by a network where STIM → THL synapses are still static but random and the training and testing stimuli are random, this would require a larger separation of timescales between the subcortical and cortical regions in the network such that at the end of training THL engrams are accurate while CTX engram cells are still silent. Fifth, our model does not include behavioral outputs but instead is limited to stimulus imprinting. As a result, memory recall is determined on the basis of reactivation of engram cells by partial cues and we do not examine how artificial reactivation of engram cells could lead to memory recall in the absence of stimulus cues. However, our extended three-region model with RDT could be interpreted as a model of stable behavioral output. In this configuration, artificial reactivation of engram cells in any region should elicit the same

response in RDT as the presentation of corresponding partial cues. Lastly, we do not explore the potential role of systems consolidation in avoiding catastrophic forgetting of older memories when new ones are acquired as proposed by previous theories[11]. Instead, our work focuses on gaining mechanistic insights into experimentally-observed engram dynamics associated with systems consolidation[33].

In the long history of the field, many computational models and theories of systems consolidation have been proposed[4,5,9,11,36–40]. While early computational studies relied on networks with highly abstract, simplified neuron models[9,11,36,37], recent computational models have become increasingly more complex to incorporate a wider range of experimental findings: a three-stage Bayesian Confidence Propagation Neural Network was used to bridge the gap between working and long-term memory[38], a spiking network was developed to explore the role of anatomical properties of the cortex-hippocampus loop in systems consolidation[39], and a rate-coded multi-layer network with a form of Hebbian learning was employed to investigate the effect of preexisting knowledge on memory consolidation[40]. Nevertheless, previous models have not addressed recent findings regarding engram cells and their role in systems consolidation. Our work, however, reproduces engram cell dynamics in HPC and CTX in a computational model—specifically, a multi-region spiking RNN with biologically-plausible synaptic plasticity. In addition, our model reflects the role of coupled oscillations across HPC, THL, and CTX in systems consolidation and connects it to engram cell reactivations. Importantly, our findings also offer a different perspective when examining previous theories of systems consolidation. For example, multiple trace theory argues that a number of representations of the same episodic memory co-exist in the hippocampus and that each acts as an index to neocortical regions encoding features associated with the memory[18]. In this account, memory recall always requires the hippocampus since it binds the different neocortical representations to form a cohesive memory. Trace transformation theory built on multiple trace theory to propose that feature-rich episodic memory always requires engagement of the hippocampus but transformed, gist-like versions of the same memory are developed in neocortical areas such that memory recall may be possible without the hippocampus depending on the conditions of memory retrieval[20,28]. In contrast to these theories, our model suggests that memory recall reliance on the hippocampus is related to the degree of thalamocortical synaptic coupling. Thus, our findings motivate the inclusion of hippocampal-thalamic-cortical interactions in existing theories of systems consolidation and suggest that this process encompasses a vast, complex network of brain regions throughout the lifetime of a memory.

In conclusion, our model of systems consolidation exhibits known region-specific engram cell dynamics and captures the active role of both HPC engram cells and coupled HPC–THL–CTX engram reactivations in this process. We also make several testable predictions regarding HPC and THL engram cells, inhibitory engram cells, inhibitory input to THL, and the relationship between THL → CTX synaptic coupling and retrograde amnesia induced by HPC lesions. Overall, our results suggest that coordinated communication across subcortical-cortical circuits—enabled by coupled engram reactivations—is essential for engram dynamics that ultimately culminate in systems consolidation. Engram cell dynamics in other brain regions, engram interactions in multi-task settings, and the link between engram cells and neurodegenerative diseases will each warrant future experimental and computational studies.

## Methods

**Neuron model**. Our model makes use of leaky integrate-and-fire neurons with spike frequency adaption. The membrane voltage $U_i$ of neuron $i$ evolves according to[41]

$$\tau^m \frac{dU_i}{dt} = (U^{\text{rest}} - U_i) + g_i^{\text{exc}}(t)(U^{\text{exc}} - U_i) + \left(g_i^{\text{gaba}}(t) + g_i^a(t)\right)(U^{\text{inh}} - U_i) \quad (1)$$

where $\tau^m$ is the membrane time constant, $U^{rest}$ is the membrane resting potential, $U^{exc}$ is the excitatory reversal potential, and $U^{inh}$ is the inhibitory reversal potential. The evolution of the synaptic conductance terms $g_i^{exc}(t)$, $g_i^{gaba}(t)$, and $g_i^a(t)$ is discussed in the next section.

A neuron $i$ fires a spike when its membrane voltage exceeds a threshold $\vartheta_i$. At this point, its membrane voltage is set to $U_i^{rest}$ and its firing threshold is temporarily increased to $\vartheta^{spike}$. Without further spikes, the firing threshold decays to its resting value $\vartheta^{rest}$ with time constant $\tau^{thr}$ following:

$$\tau^{thr}\frac{d\vartheta_i}{dt} = \vartheta^{rest} - \vartheta_i \tag{2}$$

**Synapse model.** We adopted a conductance-based synaptic input model. The dynamics of inhibitory synaptic input $g_i^{gaba}$ and spike-triggered adaption $g_i^a$ follow[41]:

$$\frac{dg_i^{gaba}}{dt} = -\frac{g_i^{gaba}}{\tau^{gaba}} + \sum_{j \in inh} w_{ij} S_j(t) \tag{3}$$

$$\frac{dg_i^a}{dt} = -\frac{g_i^a}{\tau^a} + \Delta^a S_i(t) \tag{4}$$

where $S_j(t) = \sum_k \delta(t - t_j^k)$ is the presynaptic spike train and $S_i(t) = \sum_k \delta(t - t_i^k)$ is the postsynaptic spike train. In both cases, $\delta$ denotes the Dirac delta function and $t_x^k (k = 1, 2, ...)$ are the firing times of neuron $x$. $w_{ij}$ is the weight from neuron $j$ to neuron $i$. $\Delta^a$ is a fixed adaptation strength. $\tau^{gaba}$ is the GABA decay time constant and $\tau^a$ is the adaptation time constant.

Excitatory synaptic input is determined by a combination of a fast AMPA-like conductance $g_i^{ampa}(t)$ and a slow NMDA-like conductance $g_i^{nmda}(t)$

$$g_i^{exc}(t) = \alpha g_i^{ampa}(t) + (1 - \alpha)g_i^{nmda}(t) \tag{5}$$

$$\frac{dg_i^{ampa}}{dt} = -\frac{g_i^{ampa}}{\tau^{ampa}} + \sum_{j \in exc} w_{ij} \underbrace{u_j(t)x_j(t)}_{Short-TermPlasticity} S_j(t) \tag{6}$$

$$\tau^{nmda}\frac{dg_i^{nmda}}{dt} = -g_i^{nmda} + g_i^{ampa} \tag{7}$$

where $\alpha$ is a constant that determines the relative contribution of $g_i^{ampa}(t)$ and $g_i^{nmda}(t)$ while $\tau^{ampa}$ and $\tau^{nmda}$ are their respective time constants. $u_j(t)$ and $x_j(t)$ are variables that determine the state of short-term plasticity as described in the following section.

**Synaptic plasticity model.** Our synaptic plasticity model was designed after previous work that showed that a combination of Hebbian (i.e., triplet STDP) and non-Hebbian (i.e., heterosynaptic and transmitter-induced) forms of plasticity can yield stable memory formation and recall in a single-region spiking RNN[41].

*Short-term plasticity.* The state variables $u_j(t)$ and $x_j(t)$ associated with short-term plasticity evolve according to[41]

$$\frac{d}{dt}x_j(t) = \frac{1 - x_j(t)}{\tau^d} - u_j(t)x_j(t)S_j(t) \tag{8}$$

$$\frac{d}{dt}u_j(t) = \frac{U - u_j(t)}{\tau^f} + U\left(1 - u_j(t)\right)S_j(t) \tag{9}$$

where $\tau^d$ and $\tau^f$ are the depression and facilitation time constants, respectively. The parameter $U$ is the initial release probability.

*Long-term excitatory synaptic plasticity.* Long-term excitatory synaptic plasticity takes the form of combined triplet STDP[42], heterosynaptic plasticity[43], and transmitter-induced plasticity[44] with a synaptic weight $w_{ij}$ from neuron $j$ to neuron $i$ following[41]:

$$\frac{d}{dt}w_{ij}(t) = \eta^{exc}\left(Az_j^+(t)z_i^{slow}(t - \epsilon)S_i(t) - B_i(t)z_i^-(t)S_j(t)\right) \quad triplet \tag{10a}$$

$$-\beta\left(w_{ij} - \tilde{w}_{ij}(t)\right)\left(z_i^-(t - \epsilon)\right)^3 S_i(t) \quad heterosynaptic \tag{10b}$$

$$+\delta S_j(t) \quad transmitter - induced \tag{10c}$$

where $\eta^{exc}$ (excitatory learning rate), $A$ (LTP rate), $\beta$ (heterosynaptic plasticity strength), and $\delta$ (transmitter-induced plasticity strength) are fixed parameters. $\epsilon$ is an infinitesimal offset used to ensure that the current action potential is not considered in the trace. State variables $z_{j/i}^x$ denote either pre- or postsynaptic traces and each has an independent temporal evolution with time constant $\tau^x$ given by

$$\frac{dz_{j/i}^x}{dt} = -\frac{z_{j/i}^x}{\tau^x} + S_{j/i}(t) \tag{11}$$

The reference weights $\tilde{w}_{ij}(t)$ also have their own independent synaptic consolidation dynamics following the negative gradient of a double-well potential[41]

$$\tau^{cons}\frac{d}{dt}\tilde{w}_{ij}(t) = w_{ij}(t) - \tilde{w}_{ij}(t) - P\tilde{w}_{ij}(t)\left(\frac{w^P}{2} - \tilde{w}_{ij}(t)\right)\left(w^P - \tilde{w}_{ij}(t)\right) \tag{12}$$

where $P$ and $w^P$ are fixed parameters and $\tau^{cons}$ is the synaptic consolidation time constant. $P$ controls the magnitude of the double-well potential. For $w^P = 0.5$, an upper stable fixed point is reached when $w_{ij}(t) = \tilde{w}_{ij}(t)$ while a lower stable fixed point is set when $\tilde{w}_{ij}(t) = 0$. If $w_{ij}(t) > \tilde{w}_{ij}(t)$ by a small margin, then both stable fixed points of $\tilde{w}_{ij}(t)$ experience an increase. If $w_{ij}(t) \gg \tilde{w}_{ij}(t)$, then $\tilde{w}_{ij}(t)$ only retains a single fixed point with a high value. This synaptic consolidation model is consistent with previous work[77] and with synaptic tagging experiments that found that the persistence of LTP depends on events occurring both during and prior to its initial induction[78]. Furthermore, the model assumes that there are molecular mechanisms in place such that synapses can retain a stable efficacy (i.e., weight) despite intermittent fluctuations such as those associated with molecular turnover[79,80]. Importantly, the LTD rate $B_i(t)$ is subject to homeostatic regulation and evolves according to:

$$B_i(t) = \begin{cases} AC_i(t) & for\ C_i(t) \le 1. \\ A & otherwise \end{cases} \tag{13}$$

$$\frac{d}{dt}C_i(t) = -\frac{C_i(t)}{\tau^{hom}} + \left(z_i^{ht}(t)\right)^2 \tag{14}$$

where $\tau^{hom}$ is a time constant and $z_i^{ht}(t)$ is a synaptic trace that follows Eq. (11) with its own time constant $\tau^{ht}$. Lastly, plastic excitatory weights are constrained to lower and upper bounds $w_{exc}^{min}$ and $w_{exc}^{max}$, respectively. However, excitatory weights never reach their upper bound with the exception of some simulations with blockage of neurons.

*Inhibitory synaptic plasticity.* Inhibitory synaptic plasticity follows a network activity-based STDP rule[41]

$$\frac{d}{dt}w_{ij}(t) = \eta^{inh}G(t)\left[\left(z_i(t) + 1\right)S_j(t) + z_j(t)S_i(t)\right] \tag{15}$$

$$G(t) = H(t) - \gamma \tag{16}$$

$$\frac{d}{dt}H(t) = -\frac{H(t)}{\tau^H} + \sum_{i \in exc} S_i(t) \tag{17}$$

where $\eta^{inh}$ is a constant inhibitory learning rate and $z_{j/i}$ denotes either pre- or postsynaptic traces that follow Eq. (11) with a common time constant $\tau^{iSTDP}$. $G(t)$ is a linear function of the difference between a hypothetical global secreted factor $H(t)$ and the target local network activity level $\gamma$. $H(t)$ is itself a low-pass-filtered version of the spikes fired by all excitatory neurons in the local network (i.e., either HPC, THL, or CTX) with time constant $\tau^H$. Note that inhibitory synaptic plasticity primarily aims to control network activity. Finally, inhibitory weights are constrained to the interval between $w_{inh}^{min}$ and $w_{inh}^{max}$ but they never reach their upper limit except in some simulations with blockage of neurons.

**Network model.** In each network configuration considered (i.e., Fig. 1a, Supplementary Figs. 6a, 7a and 8a, and Fig. 2a), the model consists of a stimulus population of $N_{stim} = 4096$ Poisson neurons (STIM) and two or three RNNs each corresponding to a different brain region (i.e., HPC, CTX, or THL). Anatomical evidence has motivated the use of RNNs to model these regions. First, the following evidence supports an RNN model of HPC: recurrent excitatory synapses among pyramidal cells in CA3[81] and to a less extent in CA1[82]; feedforward projections from dentate gyrus to CA3 and from CA3 to CA1[83]; "back" projections from CA1 to CA3 and dentate gyrus[84] and from CA3 to dentate gyrus[85]; and local recurrent inhibitory synapses in CA3[81], CA1[84], and dentate gyrus[86]. Second, the presence of recurrent excitatory and inhibitory synapses in CTX[87] substantiates the use of an RNN model of this region. Third, several features of thalamic connectivity motivate the use of an RNN model of THL: thalamothalamic projections[88], local recurrent inhibitory synapses in THL present in primates[47], and inhibitory projections from the TRN to THL across species[47,69]. For a discussion of how TRN inhibitory input to THL may play the role of local THL inhibitory neurons in lower species that lack the latter, see Discussion. Furthermore, each region RNN is composed of $N_{exc} = 4096$ excitatory neurons and $N_{inh} = 1024$ inhibitory neurons that are recurrently connected. Recurrent excitatory synapses onto excitatory neurons display short- and long-term excitatory synaptic plasticity while excitatory synapses projecting onto inhibitory neurons exhibit only short-term plasticity. Feedforward inter-region synapses may display both short- and long-term plasticity or only short-term plasticity depending on the network configuration and they project exclusively from excitatory neurons in one region to excitatory cells in another area. In the two-region networks (Fig. 1a, Supplementary Figs. 6a, 7a and 8a), all recurrent and feedforward synapses are initialized at random following a uniform distribution. In the three-region network (Fig. 2a), STIM → HPC synapses are plastic while STIM → THL projections are static. This choice was motivated by I) the different levels of engagement of HPC and THL in memory recall at recent versus remote

time points, and II) our results in Fig. 1 that demonstrate that changes in feed-forward synapses over the course of systems consolidation underlie changes in engram cell state (i.e., silent-to-active or active-to-silent). Specifically, previous experiments showed that while HPC engram cells switch from active to silent with systems consolidation (i.e., de-maturation)[33], THL has a high probability of holding engram cells active at recent time points[53] and is essential for recall at remote time points[48]. This suggests that THL engram cells are active in both recent and remote recall. Consequently, making STIM → HPC synapses plastic enables the de-maturation of HPC engram cells while making STIM → THL projections static supports active THL engram cells in both recent and remote recall in a manner consistent with previous experiments for both regions. As expected, static STIM → THL synapses lead to stable recall in THL throughout systems consolidation in our simulations (Fig. 2c). Moreover, STIM → HPC and STIM → THL synapses in the three-region network have randomly-centered circular receptive fields (i.e., each excitatory neuron in HPC and THL receives projections from a small circular area in STIM of radius $R_{hpc}$ and $R_{thl}$, respectively, whose random center location follows a uniform distribution). This configuration, combined with the non-random spatial structure of the training and testing stimuli used for the three-region network (i.e., horizontal bars depicted in Fig. 2a), facilitates learning in THL when STIM → THL synapses are static. The remaining feedforward as well as all recurrent synapses in the three-region network are initially random following a uniform distribution. In addition, inhibitory synapses onto inhibitory neurons are static while inhibitory synapses projecting onto excitatory neurons display inhibitory synaptic plasticity. Plasticity is constantly active for the entirety of all simulations. Recurrent synapses are connected with probability $\epsilon_{rec}$ and are initialized with specific weights (i.e., $w^{EE}$, $w^{EI}$, $w^{II}$, and $w^{IE}$). Feedforward synapses have specific connection probabilities and initial weights (e.g., $\epsilon_{hpc \to ctx}$ and $w_{hpc \to ctx}$, respectively, for Supplementary Fig. 6a). For a complete list of network parameters, see Supplementary Table 1.

**Simulation of two-region networks with HPC and CTX.** Simulations with two-region networks (i.e., Fig. 1a, Supplementary Figs. 6a, 7a and 8a) follow a defined sequence: burn-in, training, consolidation, and testing. The initial brief burn-in period of duration $T_{burn}$ stabilizes activity in each RNN under STIM background firing at rate $\nu^{bg}$. Subsequently, four random stimuli (depicted in Fig. 1a) are randomly presented to the network in the training phase of duration $T_{training}$ with equal probability and with inter-stimulus interval and stimulus presentation duration drawn from exponential distributions with means $T_{Off}^{training}$ and $T_{On}^{training}$, respectively. This is accomplished by maintaining the STIM background firing at $\nu^{bg}$ but selectively increasing the firing rate of the STIM neurons that correspond to a given stimulus to $\nu^{stim}$ for the duration of its presentation. Each stimulus consists of a non-overlapping random subset of 25% of the STIM neurons. Post-training, the network evolves spontaneously in the consolidation phase of duration $T_{consolidation}$ in the absence of stimulus presentations with STIM sustaining background firing at $\nu^{cons}$. It has been shown that reactivations of past experiences can take place during awake periods in both CTX and HPC[89,90] and, hence, awake states may also be suitable for consolidation. However, our model aims to capture recent findings that coupled oscillatory hippocampal–thalamic–cortical activity during sleep is essential for systems consolidation[30] and, hence, we set separate periods for training and consolidation. After consolidation, the network advances to the final testing phase of duration $T_{testing}$. During testing, we present partial cues (depicted in Fig. 1a) to the network by keeping STIM background firing at $\nu^{bg}$ and increasing the firing rate of the cue neurons to $\nu^{stim}$. Cue-off and cue-on periods also follow exponential distributions with means $T_{Off}^{testing}$ and $T_{On}^{testing}$, respectively. Each cue consists of a random 50% of the original stimulus. In the two-region networks, feedforward synapses have short- and long-term plasticity with the exception of HPC → CTX synapses in Supplementary Fig. 7a that only exhibit short-term plasticity. When blocking the output of engram cells in a given region, the inter-region efferent synapses of those cells are blocked but the recurrent counterparts are not to avoid finite-size effects. This procedure effectively allows for probing the effect of engram cells in downstream regions without changing the local engram dynamics. Critically, we set $\eta_{hpc}^{exc/inh} > \eta_{ctx}^{exc/inh}$ and $\tau_{hpc}^{cons} > \tau_{ctx}^{cons}$. The higher learning rate in HPC relative to CTX reflects the experimental observation that engram cells in HPC are generated in an active state while those in CTX are initially in a silent state[33]. On the other hand, the longer synaptic consolidation timescale in HPC compared to CTX is intended to render newly-encoded engrams in HPC less stable than those in CTX consistent with reported engram dynamics[33]: engram cells in HPC switch from active to silent while those in CTX change from silent to active. In Supplementary Fig. 6a, we set $\eta_{hpc \to ctx}^{exc} = \eta_{ctx}^{exc}$ and $\tau_{hpc \to ctx}^{cons} = \tau_{ctx}^{cons}$. In Supplementary Fig. 7a, HPC → CTX synapses are fixed (i.e., only have short-term plasticity). For a complete list of parameters for simulations of the two-region networks, see Supplementary Table 1.

**Simulation of three-region network with HPC, THL, and CTX.** Simulations with the three-region network (Fig. 2a) follow the same sequence as those with two-region networks (i.e., burn-in, training, consolidation, and testing). However, the three-region network is trained with four horizontal bars (as opposed to random stimuli) and is tested with partial cues consisting of the central 50% of the full bars (full bars and cues depicted in Fig. 2a). Furthermore, STIM does not provide

background firing in the consolidation phase to reflect the gating of sensory processing by spindles during sleep[91–93]. Instead, random background input at rate $\nu_{ext}^{cons}$ is provided independently to HPC and CTX during consolidation via two separate external populations of $N_{ext}^{hpc} = N_{ext}^{ctx} = 4096$ Poisson neurons. This reflects previous observations that THL activity is increased around hippocampal ripples coupled to spindles but suppressed otherwise[49] and that oscillations in HPC and CTX can occur independently of each other[54]. Outside consolidation periods, the external populations projecting to HPC and CTX remain silent (i.e., $\nu_{ext}^{bg} = 0$ Hz). The procedure to block the output of engram cells in the three-region network is the same as in the two-region configuration. When blocking the output of inhibitory neurons, their efferent synapses onto both inhibitory and excitatory neurons are blocked. In simulations with HPC ablation, HPC and all its afferent and efferent synapses are removed from the network either for the entirety of the testing phase (Fig. 5) or for part of the consolidation phase and the subsequent testing phase (Supplementary Fig. 26). In the three-region network, feedforward synapses exhibit short- and long-term plasticity with the exception of those from STIM to THL (i.e., STIM → THL synapses only have short-term plasticity) as we assume that THL receptive fields have been learned during development and only change over timescales longer than those captured in our simulations. We set $\eta_{hpc \to thl}^{exc} = \eta_{thl}^{exc} = \eta_{hpc}^{exc}$ and $\eta_{thl \to ctx}^{exc} = \eta_{ctx}^{exc}$ with $\eta_{hpc}^{exc/inh} > \eta_{ctx}^{exc/inh}$. This is based on the view that the rate of change of synaptic weights tends to be higher for sub-cortical synapses compared to cortical ones. All plastic excitatory synapses share the same $\tau_*^{cons}$ except the ones from STIM to HPC for which $\tau_{stim \to hpc}^{cons} > \tau_*^{cons}$. This is in line with the results of our simulations of two-region networks (Fig. 1) which showed that having a longer $\tau_{stim \to hpc}^{cons}$ leads to a post-training decrease in feed-forward STIM weights to HPC and the resulting de-maturation of HPC engram cells. For a complete list of parameters for simulations of the three-region network, see Supplementary Table 1.

**Labeling engram cells and computing recall metrics.** Engram cells are labeled in our model by computing the average stimulus-evoked firing rate of each neuron in a given RNN (i.e., HPC, CTX, or THL). A neuron is said to be an engram cell encoding a given stimulus if its average stimulus-evoked firing rate is above a threshold $\zeta^{thr} = 10$ Hz for the last $\Delta t^{eng} = 300$ s of the training phase and, hence, a single neuron may become an engram cell encoding multiple stimuli. In addition, an engram cell ensemble encoding a given stimulus is taken as activated upon presentation of a partial cue if its population firing rate (i.e., average firing rate computed over all engram cells in a given ensemble) is above the threshold $\zeta^{thr} = 10$ Hz during cue presentation. We then define recall true positive rate as the number of instances when the corresponding engram cell ensemble was activated following cue presentation divided by the total number of cue presentations in the testing phase. Inversely, upon presentation of a partial cue, false positive rate is defined as the fraction of engram cell ensembles encoding stimuli different than the one corresponding to the cue but that were nonetheless activated during cue presentation. This rate is then averaged across all cue presentations during the testing phase and this average is reported as recall false positive rate. Successful recall is said to happen when only the corresponding engram cell ensemble is activated by a partial cue (i.e., all other engram cell ensembles must be inactive). We then define recall accuracy as the number of successful recalls divided by the number of cue presentations in the testing phase. We compute 90% confidence intervals for the mean of recall metrics using a non-parametric bootstrap to provide a measure of uncertainty and to aid in the visualization of these metrics.

**Statistics.** Memory recall metrics (i.e., accuracy, true positive rate, and false positive rate) of two different regions in a given simulation are compared at multiple time points using a Mann–Whitney $U$ test. Percentages of activated engram and non-engram cells are compared using either an unpaired $t$-test, a Welch's $t$-test, or a Mann–Whitney $U$ test. When comparing percentages of activated cells, an unpaired $t$-test is used when the data follow a normal distribution and have equal variances, a Welch's $t$-test is used when the data follow a normal distribution and have unequal variances, and a Mann–Whitney $U$ test is used when the data do not follow a normal distribution. Weight distributions are compared using a Kolmogorov–Smirnov test. For each test, the null hypothesis is rejected at the $p$-value $< 0.05$ level. Data met required assumptions of statistical tests. Sample sizes were selected on the basis of previous studies[33,54,66]. The coupling between the population activity of engram cells encoding a given stimulus in one region and the population activity of engram cells encoding the same or a different stimulus in another region is analyzed by plotting the cross-correlogram of the pair of population activities. For each cross-correlogram, we extract the lag that maximizes the correlation between the engram activity in two different regions (i.e., $\text{lag}_{max}$). We then use $\text{lag}_{max}$ to characterize the timescale of coupling between the engram activity in the corresponding pair of regions.

**Simulation and data analysis details.** We use the forward Euler method to update neuronal state variables with a time step $\Delta = 0.1$ ms (except in the case of reference weights $\tilde{w}$ for which we use a longer time step $\Delta_{long} = 1.2$ s for efficiency

reasons). Population activity is computed with a temporal resolution of 10 ms without smoothing or convolution.

**Code details**. Code used to perform all simulations was written in C++ utilizing the Auryn framework for spiking neural network simulation[94]. We conducted several preliminary simulations and found that setting the number of Message Passing Interface (MPI) ranks $N_{ranks} = 16$ minimized the runtime of our simulations with Auryn. Code used to analyze simulation results was written in Python 3 (version 3.8 or later) using the the following packages: numpy 1.20.1, pandas 1.2.2, cython 0.29.22, scikit-learn 0.24.1, scipy 1.6.1, matplotlib 3.3.4, and seaborn 0.11.1.

**Reporting summary**. Further information on research design is available in the Nature Research Reporting Summary linked to this article.

## Data availability

The data necessary to reproduce the simulations reported in this study are available in a public repository[95].

## Code availability

The code used to perform the simulations and data analyses described in this work is available in a public repository[95].

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

## Acknowledgements
We thank Dheeraj S. Roy for feedback on an earlier version of this manuscript. This work was funded by the President's PhD Scholarship from Imperial College London (D.F.T.). This work was also supported by BBSRC BB/N013956/1 (C.C.), BB/N019008/1 (C.C.), Wellcome Trust 200790/Z/16/Z (C.C.), Simons Foundation 564408 (C.C.), and EPSRC EP/R035806/1 (C.C.).

## Author contributions
D.F.T., S.S. and C.C. conceived the study and wrote the paper. D.F.T. performed the simulations and data analyses.

## Competing interests
The authors declare no competing interests.
