## [Peer Review File · Nature Communications]

Coordinated hippocampal-thalamic-cortical communication crucial for engram dynamics underneath systems consolidationREVIEWER COMMENTS

Reviewer #1 (Remarks to the Author):

Systems consolidation is both a widespread hypothesis in the neuroscience (stated as a 'theory' formulated with words or with abstract AI-like concepts) and a collection of diverse experimental observations. Network models of spiking neurons have not yet been able to shed light on the question of systems consolidation (as opposed to, say, synaptic consolidation where several models exist), and this paper presents an important step in this direction. This simulation study from the Clopath group links several, previously unexplained, experimental facts using a spiking neural network with three groups of neurons, representing Hippocampus, Thalamus, and Cortex, respectively. Most importantly, the slow transfer of memory from Hippocampus to Cortex is described by the model. Intriguingly, the model suggests that transfer is most likely indirect, i.e., from Hippocampus to Thalamus and from there to Cortex, rather than direct.

An attractive feature of the study is that the authors do not propose yet another novel plasticity mechanism tailored to the task of systems consolidation, but apply the modeling framework of Zenke et al. to their novel network architecture. A minor change in the time constants of the plasticity mechanism across the three groups of neurons is sufficient to account for systems consolidation. In addition to this exciting generic finding, the authors perform several lesion/blocking experiments in their simulations to check that all components of their model are necessary, and also study the role of inhibitory engram cells that emerge in their model.

I have three major and a short list of minor issues.

MAJOR.

1. I find the accumulation of simulation results repetitive. Not surprisingly, once one has a big simulation running, one can easily shut down 25 different components and change each of 50 parameters, but such a broad exploration is just a distraction when reading the main text. I would suggest to refocus on the core story line, shift Figures 2 and 4 to the supplementary material, and shorten the main text around the observations in Fig 2 and 4 to just two lines each.

2. I have a problem with the time scale of 'oscillation'.

(i) The population activity in Figs 1C and 3C shows a switch in the color on the time scale of, roughly, ten minutes. In the main text you repeatedly speak of 'precise coupling of oscillations' (e.g., twice on page 5). To the reader, this implies that the experimentally observed oscillations are also on the time scale of 10 minutes, but is that correct?

(ii) How would you have to change model parameters to get switching/oscillation on a desired time scale? What does the switching time scale in your simulations depend on?

(iii) In methods you mention that you have adaptive IF neurons, but then you use a single super-fast adaptation time scale of 5ms (which I would rather interpret as refractoriness). Adaptation is known to stretch across many time scales, hence why not use three time scales, say 5ms, 50ms, and 500ms? Would that make switching as discussed in point (i) faster? I would expect a switching every second or so, due to the fatigue of the neurons in a given assembly. Plus, maybe, sharper correlations on the millisecond time scale during the switch-on phase of the next assembly.

3. I also have a question regarding the time scale of maturation/de-maturation: the model study suggests that the maturation/de-maturation transition happens essentially during the first 1 or 2 hours. Is this also the time scale observed in experiments? Should we think of this time as corresponding to a specific phase of sleep (e.g., REM sleep), so that 1 h corresponds to a full night of sleep?

I am aware that you quickly mention issues 2 and 3 in the discussion, but I believe that point 2

needs simulations and point 3 needs further discussion.

MINOR/Detailed comments

False positive and False negative rates: I am confused how you calculate this. If there is one neuron that responds to two stimuli above threshold, is it then labeled as belonging to both stimuli? Or do you assign it to the 'best' stimulus and the response to the other stimulus is counted as false positive? If so, why would that be the right thing to do? May be the stimulus is encoded in cortex in a distributed code? In methods, you try to clarify this by the statement: number of stimuli minus one, but this did not help me. Is it always the same subset of neurons (i.e. those that respond to several stimuli) that go into false positive or false negative model; or is it rather that sometimes for stimulus A the whole cell assembly encoding stimulus B responds? For me the latter would be a 'real' false positive.

Fig 1A. I assume white means active neuron. Should be mentioned in the caption. Moreover, I suggest to reorder all the white neurons for stimulus one at the bottom and those for stimulus 4 at the top of the 2d layout, so that the reader sees the similarity and difference to the stimuli of Fig 3.

Fig 1C. I have difficulties understanding the cumulative distributions. Top graph, why is there a vertical line at 110, and 150, respectively for the two distributions? The vertical lines should not be there. Also, what is the horizontal axis? I did not see it clearly defined in the caption (I also searched in methods, but could not find it there, probably my mistake). It should be explained in the caption.

Fig 2. Shift to supplementary.

Fig 4. Shift to supplementary

Fig 5A. Are these inhibitory engram neurons in cortex? Please specify in the caption.

Fig 6B. Population activity: what is the white space close to/above the x-axis?

Why do you use an update step of 0.1ms? Can you re-run with an update step of 1ms and compare the results? My prediction is that the result is basically the same but with a simulation that runs 9 times faster

I appreciate that you give all the parameters of the model (many, indeed). Can you add in the column of the 'value' an indication (say, a label 1, or 2, or 3) that indicates if: (1) the value is taken from literature (e.g., the time constant of the IF neuron), ideally with a citation. (2) the value was chosen somewhat arbitrarily and set for convenience to this value (e.g., size of network). (3) The value has been optimized over several preliminary runs to achieve the results shown in the paper. Otherwise the reader may get the impression that you optimized about 100 parameters ...

In the Results, you state at multiple occasions. 'This is driven by the t.p.r.'. I cannot understand the logic of the sentence. The network simulation gets no information on what is a true and what is a false response, therefore the error cannot drive anything. Probably you simply mean: 'This is also reflected by the t.p.r.' (?)

In the Results, you state several times 'coupled reactivations' etc. It seems that the reader has to infer this from visual inspection of the activation pattern. The scientific way of doing this would be to measure covariance $\langle A1(t), A2(t+\tau) \rangle$ of activity across two areas, as a function of delay τ , which would then also indicate a time scale of correlation. May be there IS in fact also a correlation on the millisecond time scale? See also major comment above.

Reviewer #2 (Remarks to the Author):

Summary: Systems consolidation of memory (SCM) refers to the dynamic of memory storage and memory retrieval through time. There is now a large body of evidence (referenced in this manuscript) demonstrating that the set of brain regions responsible for the storage and recall of episodic memories changes as time passes. In particular, the hippocampus, which was shown to support the retrieval of recent memory, seems to progressively disengage as the neocortex takes over (remote memory recall). Recently, Kitamura et. al (see ref. 33) have linked this phenomenon to the progressive maturation of an originally silent memory engram in the prefrontal cortex, and the de-maturation of an originally active engram in the hippocampus (HPC). Additionally, they showed that HPC engram cells support the maturation of PFC engram cells. In this manuscript, Douglas Feitosa Tomé and colleagues propose a computational model aimed at explaining the mechanisms and circuit underlying SCM, with a particular focus on recapitulating Kitamura et. al 2017's engram dynamics results as well as demonstrating the need for a "third party" region (thalamus) to coordinate the process.

Using spiking recurrent neural networks to model various brain regions, as well as biologically plausible plasticity rules that describe how these networks change through experience, the authors assemble various anatomically-inspired networks featuring HPC and cortex, and test their ability to recall (pattern completion), after a phase of consolidation, specific stimuli that they encountered during training. Using this paradigm, they demonstrate the following points:

The maturation and de-maturation of cortical and hippocampal engrams can be recapitulated in a two regions network by setting different synaptic plasticity timescales for HPC and cortical networks (figure 1).

Even though plastic connections are added between HPC and cortical networks, simple two regions models cannot explain Kitamura et. al's results showing that the HPC engram is required for the maturation of the cortical engram (figure 2).

All three biological findings (maturation of cortical engram, de-maturation of HPC engram, and HPC engram necessity for cortical engram maturation) are recapitulated when a third region, which the authors identify as playing the role of the thalamus, is added to the network (figure 3 & 4).

Inhibitory neurons in the HPC and thalamus also play a critical role in the maturation of the cortical engram and the successful recall of stimuli at remote time points (post-consolidation) (figure 5 & 6).

Different cases of retrograde amnesia reported in studies where HPC was lesioned, can be recapitulated by manipulations of the three region model (figure 7).

Novelty and interest to the Nature Communication readership: Overall this manuscript is pleasantly well written. The findings are clear and do indeed provide a useful computational framework to think about SCM in terms of engram dynamics, which so far was missing from the literature. In addition, their HPC-THL-CTX model makes a series of novel predictions that can be experimentally tested, increasing the appeal of the study. For these reasons, I believe that this manuscript would be of interest to the Nature Communication readership. However, I did raise a few concerns (see list below), which will need to be addressed before the manuscript can be accepted.

Major points:

On line 569, the authors write "engram cell ensemble encoding a given stimulus is taken as activated upon presentation of a partial cue if its population firing rate is above the threshold". What do the authors mean by population firing rate? Is that an average computed across all

engram cells constituting the ensemble? If so, this might be problematic as a small subpopulation firing at rates sufficiently high might lead to the false conclusion that the entire ensemble is being reactivated, essentially giving a false illusion of pattern completion. In particular, the subpopulation aforementioned could be the engram cells directly reactivated by the partial stimuli through feedforward connections. The author should provide in a supplementary figure data showing how much of the labeled engram ensemble is being reactivated, both right after training and post-consolidation.

In figure 3C, I could not explain to myself why THL recall accuracies were not affected by the consolidation. Figure 4a,b show that, similar to HPC (figure 1C), the weight of recurrent connections in THL "deteriorate" through the consolidation process. However, the recall accuracy of HPC at test time is significantly worse than THL's, suggesting that the difference resides in the fact that the authors use static feedforward STIM -> THL projections. Please justify this choice, as well as explain its impact on THL recall accuracy at test time.

Could the authors explain why they did not silence specifically the inhibitory engram cells rather than blocking all inhibitory inputs when proving their critical role for CTX engram maturation (figure 6)? The authors should address this point and, in the case where the same results can be obtained with specific engram silencing, should replace Figure 6 with the new data. If the results are different, it should be mentioned in the text and the data should appear in a supplementary figure.

'Engram cells are a population of neurons that are activated by learning, have enduring cellular changes as a consequence of learning, and whose reactivation by a part of the original stimuli delivered during learning results in memory recall" (Tonegawa et al. 2015). In neuroscientific studies, the reactivation of the engram cells through the presentation of partial cues, which is the core of the paper under review, is never considered definitive proof for memory recall but only a correlation. Instead, the sole artificial reactivation of the engram cells in the absence of relevant cues must elicit the same behavior as natural recall would upon pattern completion. While I understand that extending the study past simple stimuli imprinting to include behavioral outputs might be too much, this shortcoming should be discussed and the authors should detail how one could go about addressing this issue.

Minor points:

The author should include references providing anatomical evidence motivating the use of recurrent neural networks to model HPC, CTX, and THL. Additionally, the authors should explain why, in their three regions model, STIM -> HPC is plastic while STIM -> THL is static.

Could the authors explain why (i) they changed stimuli (full and partial) between the two regions analysis and three regions analysis (e.g. figure 1a and figure 3a), and (ii) modified the patterns of feedforward connections (i.e. introduction of circular receptive fields) between these two experiments? Additionally, the authors should make it clear in the manuscript how these changes might affect the outcome of their experiments.

While the difference in engram dynamics observed between HPC and CTX can simply be explained by their diverging synaptic plasticity timescales (figure 1), the authors show that a third region (THL in their model) is required to recapitulate findings showing that the HPC engram is necessary for CTX engram maturation (figure 3). I think the manuscript would benefit from a discussion addressing why would the brain require such a third party if the end goal, namely maturation of a cortical engram, can be achieved without it.

In line with the previous question, I was wondering if there might be a qualitative difference between mature cortical engrams formed in the two regions system (figure 1), and the similar engram maturing with the support of THL and HPC (figure 3)? Maybe these two ensembles can be compared past their recall accuracy.

The authors could discuss how their model help understand other competing theories (e.g. multiple trace theory).

References:

Tonegawa, S. et al., 2015. Memory Engram Cells Have Come of Age. *Neuron*, 87(5), pp.918–931.

Reviewer #3 (Remarks to the Author):

In “Coordinated hippocampal-thalamic-cortical communication crucial for engram dynamics underneath systems consolidation” by Tome et al., the authors describe a model of memory consolidation. They speculate that coupling between the hippocampus, thalamus and cortex lead to long-term memories being consolidated in the cortex. They further speculate that inhibition and synaptic plasticity with varying timescales play roles in consolidation. A strength of this paper is that it makes testable predictions for experimental neuroscience.

MAJOR ISSUES

I find the claims made in the paper hard to justify for the following reasons:

1. The stimuli used for encoding episodic memory are overly simple and non-overlapping.
2. The feedforward anatomy of these 2 or 3 recurrent neural networks (RNN) does not reflect the complex anatomical interactions between the hippocampus and cortex. For example, the 1994 Alvarez and Squire model of memory consolidation had multiple cortical areas with bidirectional connections between the cortex and the MTL. Bidirectional coupling, reentrant connectivity, and associating different regions of cortex with a memory are key to episodic memory.
3. The dynamics of the model is not analyzed to show that there is oscillatory coupling between regions. It is claimed, but there should be analyses and quantitative metrics showing coupling and synchrony between areas. The current thinking is there is synchrony between sharp wave ripples in the hippocampus and spindles in the cortex. Does the present model show evidence for that?
4. The results are not directly compared with existing experimental evidence.

GENERAL COMMENTS

1. In general, I found the results section difficult to read. The figures are extremely busy with multiple panels that are difficult to navigate. The captions are hard to digest with A/G, E/K and J/L referred to as being related. The wording of the results points to these panels and also make it hard to follow the theme or idea being made.
2. I am not sure what makes these areas the cortex, hippocampus, and thalamus. The connectivity of these 3 RNNs are not necessarily close to the anatomy from these regions. THL in this case is just an intermediary between the HPC RNN and the CTX RNN. The projection to the THL is static.
3. A strength of this model is the biologically plausible synaptic plasticity. How do the “Hebbian (i.e., triplet STDP) and non-Hebbian (i.e., heterosynaptic and transmitted-induced) forms of plasticity” all contribute to the present results. Are all of these necessary? Are some more important than others?
4. The figures show that different neurons or synapses are contributing to encoding the stimuli (engrams). There are a number of ablation simulations to show the various contributions. But there should be some explanatory dynamics. Especially for the excitatory and inhibitory contributions. Could you vary the E/I ratios? Or could you lesion the different types of synaptic plasticity rules to see which ones are contributing to the learning?
5. The authors use “Coupling”, “oscillations” and “coordinated communication” in their language. Line 418 states, “our model captures the fact that oscillations in HPC, THL, and CTX need to be

coupled for effective systems consolidation by displaying coupled engram cell reactivations." But this should be backed up with an analysis of the synchrony and oscillatory dynamics. I did not see such an analysis. The simulated ablations are not enough.

6. The results are mostly qualitative. Quantitative statistics should be provided for all the results. For example, the weight matrices and memory recall panels that are in most figures need statistics to back up the claims being made.

SPECIFIC COMMENTS

Line 64. "One of four non-overlapping random stimuli is presented to the network at a time either for training or testing." What would happen if there were overlapping stimuli? This seems to be a more rigorous and realistic test of memory consolidation.

Figure 1C and 2C. There is no overlap for the weights. Give more details on how they were chosen for the figure. Is it just the recurrent weights into the Engram cells? What about the input weights from the Stimuli?

Section: "Subcortical engram cells are essential for CTX engram cell maturation." The assumption is that only vHPC has direct connections to the CTX and this amounts to only 5% of the connections. But are you assuming that 95% are then coming from the THL? Is that a correct assumption? What about connections from CTX to HPC?

Figure 3. If I am understanding this correctly, the THL has perfect knowledge of the stimuli. The accuracy is 100%, the t.p.r is 100%, and the f.p.r is 0. Is this reasonable? Why would the HPC and CTX be necessary if there is perfect memory in the THL?

Line 216. "This results in decoupled oscillations in HPC, THL and CTX (Fig. 3D)." There needs to be an analysis of coupled oscillations to make this claim.

Section: "Thalamocortical coupling underlies retrograde amnesia profiles." I don't see the retrograde amnesia in the reported results. It seems that to show this, the HPC would need to be lesioned at different stages of consolidation to see a graded amnesia depending on consolidation. In fact, Figure 3C (control) is near identical to Figure 7B (graded amnesia). Am I missing something?

Methods section. There are numerous open parameters. How were those values chosen? What is their justification?

Equation 12. The synaptic consolidation time constant is important for the present results. Give more background on how it works? How is it justified by neurobiology? What assumptions were made?

Lines 503-506. Why are their different distributions for the different versions of the model (i.e., uniform for the 2 RNN model and circular RFs for the 3 RNN model)? Couldn't these variations make a difference in the outcome?

Lines 544-546. Why are different stimuli used for the different models? The 2 RNN model is trained with random non-overlapping stimuli. The 3 RNN model is trained with four horizontal bars (as opposed to random stimuli) and is tested with partial cues consisting of the central 50% of the full bars (full bars and cues depicted in Fig. 3A). Couldn't these variations make a difference in the outcome?

Reviewer #1 (Remarks to the Author):

Systems consolidation is both a widespread hypothesis in the neuroscience (stated as a 'theory' formulated with words or with abstract AI-like concepts) and a collection of diverse experimental observations. Network models of spiking neurons have not yet been able to shed light on the question of systems consolidation (as opposed to, say, synaptic consolidation where several models exist), and this paper presents an important step in this direction. This simulation study from the Clopath group links several, previously unexplained, experimental facts using a spiking neural network with three groups of neurons, representing Hippocampus, Thalamus, and Cortex, respectively. Most importantly, the slow transfer of memory from Hippocampus to Cortex is described by the model. Intriguingly, the model suggests that transfer is most likely indirect, i.e., from Hippocampus to Thalamus and from there to Cortex, rather than direct.

An attractive feature of the study is that the authors do not propose yet another novel plasticity mechanism tailored to the task of systems consolidation, but apply the modeling framework of Zenke et al. to their novel network architecture. A minor change in the time constants of the plasticity mechanism across the three groups of neurons is sufficient to account for systems consolidation. In addition to this exciting generic finding, the authors perform several lesion/blocking experiments in their simulations to check that all components of their model are necessary, and also study the role of inhibitory engram cells that emerge in their model.

I have three major and a short list of minor issues.

MAJOR.

1. I find the accumulation of simulation results repetitive. Not surprisingly, once one has a big simulation running, one can easily shut down 25 different components and change each of 50 parameters, but such a broad exploration is just a distraction when reading the main text. I would suggest to refocus on the core story line, shift Figures 2 and 4 to the supplementary material, and shorten the main text around the observations in Fig 2 and 4 to just two lines each.

2. I have a problem with the time scale of 'oscillation'.

(i) The population activity in Figs 1C and 3C shows a switch in the color on the time scale of, roughly, ten minutes. In the main text you repeatedly speak of 'precise coupling of oscillations' (e.g., twice on page 5). To the reader, this implies that the experimentally observed oscillations are also on the time scale of 10 minutes, but is that correct?

(ii) How would you have to change model parameters to get switching/oscillation on a desired time scale? What does the switching time scale in your simulations depend on?

(iii) In methods you mention that you have adaptive IF neurons, but then you use a single super-fast adaptation time scale of 5ms (which I would rather interpret as refractoriness). Adaptation is known to stretch across many time scales, hence why not use three time scales, say 5ms, 50ms, and 500ms? Would that makes switching as discussed in point (i) faster? I would expect a switching

every second or so, due to the fatigue of the neurons in a given assembly. Plus, may be, sharper correlations on the millisecond time scale during the switch-on phase of the next assembly.

3. I also have a question regarding the time scale of maturation/de-maturation: the model study suggests that the maturation/de-maturation transition happens essentially during the first 1 or 2 hours. Is this also the time scale observed in experiments? Should we think of this time as corresponding to a specific phase of sleep (e.g., REM sleep), so that 1 h corresponds to a full night of sleep?

I am aware that you quickly mention issues 2 and 3 in the discussion, but I believe that point 2 needs simulations and point 3 needs further discussion.

MINOR/Detailed comments

False positive and False negative rates: I am confused how you calculate this. If there is one neuron that responds to two stimuli above threshold, is it then labeled as belonging to both stimuli? Or do you assign it to the 'best' stimulus and the response to the other stimulus is counted as false positive? If so, why would that be the right thing to do? May be the stimulus is encoded in cortex in a distributed code? In methods, you try to clarify this by the statement: number of stimuli minus one, but this did not help me. Is it always the same subset of neurons (i.e. those that respond to several stimuli) that go into false positive or false negative model; or is it rather that sometimes for stimulus A the whole cell assembly encoding stimulus B responds? For me the latter would be a 'real' false positive.

Fig 1A. I assume white means active neuron. Should be mentioned in the caption. Moreover, I suggest to reorder all the white neurons for stimulus one at the bottom and those for stimulus 4 at the top of the 2d layout, so that the reader sees the similarity and difference to the stimuli of Fig 3.

Fig 1C. I have difficulties understanding the cumulative distributions. Top graph, why is there a vertical line at 110, and 150, respectively for the two distributions? The vertical lines should not be there. Also, what is the horizontal axis? I did not see it clearly defined in the caption (I also searched in methods, but could not find it there, probably my mistake). It should be explained in the caption.

Fig 2. Shift to supplementary.

Fig 4. Shift to supplementary

Fig 5A. Are these inhibitory engram neurons in cortex? Please specify in the caption.

Fig 6B. Population activity: what is the white space close to/above the x-axis?

Why do you use an update step of 0.1ms? Can you re-run with an update step of 1ms and compare the results? My prediction is that the result is basically the same but with a simulation that runs 9 times faster

I appreciate that you give all the parameters of the model (many, indeed). Can you add in the column of the 'value' an indication (say, a label 1, or 2, or 3) that indicates if: (1) the value is taken from literature (e.g., the time constant of the IF neuron), ideally with a citation. (2) the value was chosen somewhat arbitrarily and set for convenience to this value (e.g., size of network). (3) The value has been optimized over several preliminary runs to achieve the results shown in the paper. Otherwise the reader may get the impression that you optimized about 100 parameters ...

In the Results, you state at multiple occasions. 'This is driven by the t.p.r.'. I cannot understand the logic of the sentence. The network simulation gets no information on what is a true and what is a false response, therefore the error cannot drive anything. Probably you simply mean: 'This is also reflected by the t.p.r.'(?)

In the Results, you state several times 'coupled reactivations' etc. It seems that the reader has to infer this from visual inspection of the activation pattern. The scientific way of doing this would be to measure covariance $\langle A1(t), A2(t+\tau) \rangle$ of activity across two areas, as a function of delay τ , which would then also indicate a time scale of correlation. May be there IS in fact also a correlation on the millisecond time scale? See also major comment above.

Reviewer #2 (Remarks to the Author):

Summary: Systems consolidation of memory (SCM) refers to the dynamic of memory storage and memory retrieval through time. There is now a large body of evidence (referenced in this manuscript) demonstrating that the set of brain regions responsible for the storage and recall of episodic memories changes as time passes. In particular, the hippocampus, which was shown to support the retrieval of recent memory, seems to progressively disengage as the neocortex takes over (remote memory recall). Recently, Kitamura et. al (see ref. 33) have linked this phenomenon to the progressive maturation of an originally silent memory engram in the prefrontal cortex, and the de-maturation of an originally active engram in the hippocampus (HPC). Additionally, they showed that HPC engram cells support the maturation of PFC engram cells. In this manuscript, Douglas Feitosa Tomé and colleagues propose a computational model aimed at explaining the mechanisms and circuit underlying SCM, with a particular focus on recapitulating Kitamura et. al 2017's engram dynamics results as well as demonstrating the need for a "third party" region (thalamus) to coordinate the process.

Using spiking recurrent neural networks to model various brain regions, as well as biologically plausible plasticity rules that describe how these networks change through experience, the authors assemble various anatomically-inspired networks featuring HPC and cortex, and test their ability to recall (pattern completion), after a phase of consolidation, specific stimuli that they encountered

during training. Using this paradigm, they demonstrate the following points:

The maturation and de-maturation of cortical and hippocampal engrams can be recapitulated in a two regions network by setting different synaptic plasticity timescales for HPC and cortical networks (figure 1).

Even though plastic connections are added between HPC and cortical networks, simple two regions models cannot explain Kitamura et. al's results showing that the HPC engram is required for the maturation of the cortical engram (figure 2).

All three biological findings (maturation of cortical engram, de-maturation of HPC engram, and HPC engram necessity for cortical engram maturation) are recapitulated when a third region, which the authors identify as playing the role of the thalamus, is added to the network (figure 3 & 4).

Inhibitory neurons in the HPC and thalamus also play a critical role in the maturation of the cortical engram and the successful recall of stimuli at remote time points (post-consolidation) (figure 5 & 6).

Different cases of retrograde amnesia reported in studies where HPC was lesioned, can be recapitulated by manipulations of the three region model (figure 7).

Novelty and interest to the Nature Communication readership: Overall this manuscript is pleasantly well written. The findings are clear and do indeed provide a useful computational framework to think about SCM in terms of engram dynamics, which so far was missing from the literature. In addition, their HPC-THL-CTX model makes a series of novel predictions that can be experimentally tested, increasing the appeal of the study. For these reasons, I believe that this manuscript would be of interest to the Nature Communication readership. However, I did raise a few concerns (see list below), which will need to be addressed before the manuscript can be accepted.

Major points:

On line 569, the authors write “engram cell ensemble encoding a given stimulus is taken as activated upon presentation of a partial cue if its population firing rate is above the threshold”. What do the authors mean by population firing rate? Is that an average computed across all engram cells constituting the ensemble? If so, this might be problematic as a small subpopulation firing at rates sufficiently high might lead to the false conclusion that the entire ensemble is being reactivated, essentially giving a false illusion of pattern completion. In particular, the subpopulation aforementioned could be the engram cells directly reactivated by the partial stimuli through feedforward connections. The author should provide in a supplementary figure data showing how much of the labeled engram ensemble is being reactivated, both right after training and post-consolidation.

In figure 3C, I could not explain to myself why THL recall accuracies were not affected by the consolidation. Figure 4a,b show that, similar to HPC (figure 1C), the weight of recurrent connections in THL “deteriorate” through the consolidation process. However, the recall accuracy of HPC at test time is significantly worse than THL's, suggesting that the difference resides in the fact that the authors use static feedforward STIM → THL projections. Please justify this choice, as well as

explain its impact on THL recall accuracy at test time.

Could the authors explain why they did not silence specifically the inhibitory engram cells rather than blocking all inhibitory inputs when proving their critical role for CTX engram maturation (figure 6)? The authors should address this point and, in the case where the same results can be obtained with specific engram silencing, should replace Figure 6 with the new data. If the results are different, it should be mentioned in the text and the data should appear in a supplementary figure.

‘Engram cells are a population of neurons that are activated by learning, have enduring cellular changes as a consequence of learning, and whose reactivation by a part of the original stimuli delivered during learning results in memory recall’ (Tonegawa et al. 2015). In neuroscientific studies, the reactivation of the engram cells through the presentation of partial cues, which is the core of the paper under review, is never considered definitive proof for memory recall but only a correlation. Instead, the sole artificial reactivation of the engram cells in the absence of relevant cues must elicit the same behavior as natural recall would upon pattern completion. While I understand that extending the study past simple stimuli imprinting to include behavioral outputs might be too much, this shortcoming should be discussed and the authors should detail how one could go about addressing this issue.

Minor points:

The author should include references providing anatomical evidence motivating the use of recurrent neural networks to model HPC, CTX, and THL. Additionally, the authors should explain why, in their three regions model, $STIM \rightarrow HPC$ is plastic while $STIM \rightarrow THL$ is static.

Could the authors explain why (i) they changed stimuli (full and partial) between the two regions analysis and three regions analysis (e.g. figure 1a and figure 3a), and (ii) modified the patterns of feedforward connections (i.e. introduction of circular receptive fields) between these two experiments? Additionally, the authors should make it clear in the manuscript how these changes might affect the outcome of their experiments.

While the difference in engram dynamics observed between HPC and CTX can simply be explained by their diverging synaptic plasticity timescales (figure 1), the authors show that a third region (THL in their model) is required to recapitulate findings showing that the HPC engram is necessary for CTX engram maturation (figure 3). I think the manuscript would benefit from a discussion addressing why would the brain require such a third party if the end goal, namely maturation of a cortical engram, can be achieved without it.

In line with the previous question, I was wondering if there might be a qualitative difference between mature cortical engrams formed in the two regions system (figure 1), and the similar engram maturing with the support of THL and HPC (figure 3)? Maybe these two ensembles can be compared past their recall accuracy.

The authors could discuss how their model help understand other competing theories (e.g. multiple

trace theory).

References:

Tonegawa, S. et al., 2015. Memory Engram Cells Have Come of Age. *Neuron*, 87(5), pp.918–931.

Reviewer #3 (Remarks to the Author):

In “Coordinated hippocampal-thalamic-cortical communication crucial for engram dynamics underneath systems consolidation” by Tome et al., the authors describe a model of memory consolidation. They speculate that coupling between the hippocampus, thalamus and cortex lead to long-term memories being consolidated in the cortex. They further speculate that inhibition and synaptic plasticity with varying timescales play roles in consolidation. A strength of this paper is that it makes testable predictions for experimental neuroscience.

MAJOR ISSUES

I find the claims made in the paper hard to justify for the following reasons:

1. The stimuli used for encoding episodic memory are overly simple and non-overlapping.
2. The feedforward anatomy of these 2 or 3 recurrent neural networks (RNN) does not reflect the complex anatomical interactions between the hippocampus and cortex. For example, the 1994 Alvarez and Squire model of memory consolidation had multiple cortical areas with bidirectional connections between the cortex and the MTL. Bidirectional coupling, reentrant connectivity, and associating different regions of cortex with a memory are key to episodic memory.
3. The dynamics of the model is not analyzed to show that there is oscillatory coupling between regions. It is claimed, but there should be analyses and quantitative metrics showing coupling and synchrony between areas. The current thinking is there is synchrony between sharp wave ripples in the hippocampus and spindles in the cortex. Does the present model show evidence for that?
4. The results are not directly compared with existing experimental evidence.

GENERAL COMMENTS

1. In general, I found the results section difficult to read. The figures are extremely busy with multiple panels that are difficult to navigate. The captions are hard to digest with A/G, E/K and J/L referred to as being related. The wording of the results points to these panels and also make it hard to follow the theme or idea being made.
2. I am not sure what makes these areas the cortex, hippocampus, and thalamus. The connectivity of these 3 RNNs are not necessarily close to the anatomy from these regions. THL in this case is just an intermediary between the HPC RNN and the CTX RNN. The projection to the THL is

static.

3. A strength of this model is the biologically plausible synaptic plasticity. How do the “Hebbian (i.e., triplet STDP) and non-Hebbian (i.e., heterosynaptic and transmitted-induced) forms of plasticity” all contribute to the present results. Are all of these necessary? Are some more important than others?

4. The figures show that different neurons or synapses are contributing to encoding the stimuli (engrams). There are a number of ablation simulations to show the various contributions. But there should be some explanatory dynamics. Especially for the excitatory and inhibitory contributions. Could you vary the E/I ratios? Or could you lesion the different types of synaptic plasticity rules to see which ones are contributing to the learning?

5. The authors use “Coupling”, “oscillations” and “coordinated communication” in their language. Line 418 states, “our model captures the fact that oscillations in HPC, THL, and CTX need to be coupled for effective systems consolidation by displaying coupled engram cell reactivations.” But this should be backed up with an analysis of the synchrony and oscillatory dynamics. I did not see such an analysis. The simulated ablations are not enough.

6. The results are mostly qualitative. Quantitative statistics should be provided for all the results. For example, the weight matrices and memory recall panels that are in most figures need statistics to back up the claims being made.

SPECIFIC COMMENTS

Line 64. “One of four non-overlapping random stimuli is presented to the network at a time either for training or testing.” What would happen if there were overlapping stimuli? This seems to be a more rigorous and realistic test of memory consolidation.

Figure 1C and 2C. There is no overlap for the weights. Give more details on how they were chosen for the figure. Is it just the recurrent weights into the Engram cells? What about the input weights from the Stimuli?

Section: “Subcortical engram cells are essential for CTX engram cell maturation.” The assumption is that only vHPC has direct connections to the CTX and this amounts to only 5% of the connections. But are you assuming that 95% are then coming from the THL? Is that a correct assumption? What about connections from CTX to HPC?

Figure 3. If I am understanding this correctly, the THL has perfect knowledge of the stimuli. The accuracy is 100%, the t.p.r is 100%, and the f.p.r is 0. Is this reasonable? Why would the HPC and CTX be necessary if there is perfect memory in the THL?

Line 216. “This results in decoupled oscillations in HPC, THL and CTX (Fig. 3D).” There needs to be an analysis of coupled oscillations to make this claim.

Section: “Thalamocortical coupling underlies retrograde amnesia profiles.” I don’t see the retro-

grade amnesia in the reported results. It seems that to show this, the HPC would need to be lesioned at different stages of consolidation to see a graded amnesia depending on consolidation. In fact, Figure 3C (control) is near identical to Figure 7B (graded amnesia). Am I missing something?

Methods section. There are numerous open parameters. How were those values chosen? What is their justification?

Equation 12. The synaptic consolidation time constant is important for the present results. Give more background on how it works? How is it justified by neurobiology? What assumptions were made?

Lines 503-506. Why are their different distributions for the different versions of the model (i.e., uniform for the 2 RNN model and circular RFs for the 3 RNN model)? Couldn't these variations make a difference in the outcome?

Lines 544-546. Why are different stimuli used for the different models? The 2 RNN model is trained with random non-overlapping stimuli. The 3 RNN model is trained with four horizontal bars (as opposed to random stimuli) and is tested with partial cues consisting of the central 50% of the full bars (full bars and cues depicted in Fig. 3A). Couldn't these variations make a difference in the outcome?

Response to the reviewers

Douglas Feitosa Tomé, Sadra Sadeh, Claudia Clopath

We thank the reviewers for their thorough critical evaluation of our work. Here, we address each of their concerns point by point.

Reviewer 1

Major Points

Reviewer Point P 1.1 — I find the accumulation of simulation results repetitive. Not surprisingly, once one has a big simulation running, one can easily shut down 25 different components and change each of 50 parameters, but such a broad exploration is just a distraction when reading the main text. I would suggest to refocus on the core story line, shift Figures 2 and 4 to the supplementary material, and shorten the main text around the observations in Figs 2 and 4 to just two lines each.

Reply: We have moved Figures 2 and 4 in the original manuscript to the Supplementary Material and shortened the related text. Specifically, Figure 2 in the original manuscript is now split into two in the revised manuscript (Figure S6 and Figure S7) and their accompanying text is on page 4, line 129. We have also merged Figure S1A-C in the original manuscript to Figure S6 in the revised manuscript to combine the analysis of plastic monosynaptic HPC→CTX coupling in a single supplementary figure. In addition, Figure 4 in the original manuscript was moved to Figure S18 in the revised manuscript and its accompanying text is on page 6, line 246. We have also merged Figure S6 in the original manuscript to Figure S18 in the revised manuscript to combine the analysis of all excitatory weights in the associated HPC→THL→CTX circuit in a single supplementary figure. The reorganization of figures discussed here (i.e., splitting and merging) was done to facilitate navigating each figure individually and to highlight the major limitations of each alternative model configuration separately in an effort to help readers understand their shortcomings without discussing them at length in the main text. Taken together, the changes above have refocused the main text on the key message of the paper while providing references to the Supplementary Material for additional analyses.

Reviewer Point P 1.2 — I have a problem with the time scale of “oscillation”.

Reviewer Point P 1.2.1 — (i) The population activity in Figs 1C and 3C shows a switch in the color on the time scale of, roughly, ten minutes. In the main text you repeatedly speak of “precise coupling of oscillations” (e.g., twice on page 5). To the reader, this implies that the experimentally observed oscillations are also on the time scale of 10 minutes, but is that correct?

Reply: We have modified the text to clarify that the engram reactivation patterns exhibited by our model do not capture the stereotypical oscillatory patterns associated with systems consolidation. Specifically, we have included a discussion of how our work does not explicitly model hippocampal sharp-wave ripples,

thalamic spindles, and cortical slow oscillations but instead captures the fact that the coupling of these oscillations has been linked to engram reactivation in the brain (see page 11, line 470 in the revised manuscript). Accordingly, we have altered the text in the Results section to clarify how our model compares to experimentally-observed oscillations associated with systems consolidation (see page 5, line 187 in the revised manuscript).

Reviewer Point P 1.2.2 — (ii) How would you have to change model parameters to get switching/oscillation on a desired time scale? What does the switching time scale in your simulations depend on?

Reply: We have performed new simulations that demonstrated that the timescale of engram switching in the model depends on the transmitter-induced plasticity strength and on the timescales of neuronal adaptation (see page 3, line 85 and Fig. S2 in the revised manuscript). Specifically, doubling the transmitter-induced plasticity strength δ while keeping the learning rate η constant makes engram reactivations switch approximately twice as fast (compare Fig. 1C and Fig. S2A in the revised manuscript). Furthermore, adding a second spike-triggered adaptation time constant of 50 ms besides the adaptation time constant of 100 ms already present in our original simulations led to fast engram reactivations in the timescale of seconds (compare Fig. 1C and Fig. S2B in the revised manuscript) — but note that even these accelerated timescales in our model are still longer than the ones observed in slow oscillations and sharp-wave ripples and, hence, our model does not capture these experimentally-observed oscillatory patterns as discussed on page 11, line 470 in the revised manuscript.

Reviewer Point P 1.2.3 — (iii) In methods you mention that you have adaptive IF neurons, but then you use a single super-fast adaptation time scale of 5ms (which I would rather interpret as refractoriness). Adaptation is known to stretch across many time scales, hence why not use three time scales, say 5ms, 50ms, and 500ms? Would that makes switching as discussed in point (i) faster? I would expect a switching every second or so, due to the fatigue of the neurons in a given assembly. Plus, may be, sharper correlations on the millisecond time scale during the switch-on phase of the next assembly.

Reply: We have performed new simulations that demonstrated that including additional adaptation time constants shortens the timescale of engram reactivation switching in the model (see page 3, line 85 and Fig. S2 in the revised manuscript). As discussed in the previous related point P 1.2.2, our original simulations included a single adaptation time constant $\tau^a = 100$ ms (see Table 1 in the revised manuscript) and we have verified that adding a second adaptation time constant of 50 ms makes engram switching happen in a much shorter timescale of seconds (compare Fig. 1C and Fig. S2B in the revised manuscript). Regarding the timescale of correlation between engram reactivations, we have computed the lag between engram reactivations in different regions, lag_{\max} , (see page 17, line 701 in the revised manuscript and related point P 1.14) and verified that engram reactivations in HPC and CTX are decoupled in the two-region network (Fig. S1 in the revised manuscript: lag_{\max} between 132.56 s and 940.17 s for pairs of engrams encoding the same stimulus in HPC and CTX). However, engram reactivations are coupled across regions in the HPC→THL→CTX circuit on the millisecond timescale (Fig. S13 in the revised manuscript: lag_{\max} between 0 and 140 ms for pairs of engrams encoding the same stimulus in two different regions).

Reviewer Point P 1.3 — I also have a question regarding the time scale of maturation/de-maturation: the model study suggests that the maturation/de-maturation transition happens es-

entially during the first 1 or 2 hours. Is this also the time scale observed in experiments? Should we think of this time as corresponding to a specific phase of sleep (e.g., REM sleep), so that 1 h corresponds to a full night of sleep?

Reply: We have included a discussion of how the timescale of systems consolidation in our model compares to the one observed in experiments (see page 11, line 463 in the revised manuscript). Specifically, engram maturation and de-maturation have been reported in the timescale of weeks [1] but engrams turn active or silent in our simulations in the first few hours of consolidation. Although we could in principle easily reduce learning rates in our model to match experimentally-observed timescales for engram dynamics associated with systems consolidation, it is impractical due to the increased computational cost when simulating large-scale networks like ours (e.g., our three-region network has 15,360 neurons in total). Hence, the timescale of systems consolidation in our model does not directly match experimentally-observed ones. However, it is thought that active systems consolidation takes place in periods of quiet wakefulness and non-rapid eye movement (NREM) sleep when engram reactivations subserving this process are facilitated [2]. Thus, consolidation time in our model could be interpreted as periods of quiet wakefulness and NREM sleep and, consequently, this would extend the overall effective time over which systems consolidation takes place in our simulations — bringing it closer to the timescales observed in experiments.

Minor Points

Reviewer Point P 1.4 — False positive and False negative rates: I am confused how you calculate this. If there is one neuron that responds to two stimuli above threshold, is it then labeled as belonging to both stimuli? Or do you assign it to the “best” stimulus and the response to the other stimulus is counted as false positive? If so, why would that be the right thing to do? Maybe the stimulus is encoded in cortex in a distributed code? In methods, you try to clarify this by the statement: number of stimuli minus one, but this did not help me. Is it always the same subset of neurons (i.e. those that respond to several stimuli) that go into false positive or false negative model; or is it rather that sometimes for stimulus A the whole cell assembly encoding stimulus B responds? For me the latter would be a “real” false positive.

Reply: We have clarified the definition of false positive rate in the Methods section and it is aligned with the reviewer’s definition of “real” false positive (see page 16, line 686 in the revised manuscript). Specifically, a neuron that responds to two or more stimuli is assigned to the engram cell ensembles of all stimuli to which it responds. Hence, a single neuron may be part of multiple engram ensembles encoding different stimuli. Upon presentation of a partial cue, an engram cell ensemble is regarded as activated if the average firing rate of the entire ensemble is above the threshold of 10 Hz. Then, false positive rate is defined as the fraction of engram cell ensembles that fired above the threshold but actually encode a stimulus different than the one corresponding to the presented cue (e.g., presenting cue for stimulus A and activating the engram ensembles encoding stimulus B and C but not the one encoding stimulus D would yield a false positive rate of 2/3). Finally, we average the false positive rate for individual cues over all cue presentations in the testing phase and report this average as recall false positive rate. We have modified the Methods section to clarify the steps outlined above (see page 16, line 678 in the revised manuscript).

Reviewer Point P 1.5 — Fig 1A. I assume white means active neuron. Should be mentioned in

the caption. Moreover, I suggest to reorder all the white neurons for stimulus one at the bottom and those for stimulus 4 at the top of the 2d layout, so that the reader sees the similarity and difference to the stimuli of Fig 3.

Reply: We have included a statement in the caption of Fig. 1A and Fig. 2A in the revised manuscript to indicate that light gray in stimulus plots means active neurons. Regarding reordering the active neurons in stimulus plots, we believe that this may give a false impression that the stimuli in Fig. 1A in the revised manuscript are not spatially random when in fact this is the key difference between these stimuli and the ones in Fig. 2A in the revised manuscript. Specifically, each active neuron in a particular stimulus has a specific position in the 2D-grid representation of the stimulus population and, hence, each stimulus has a fixed mapping to this 2D grid. In Fig. 1A, the positions of the active neurons in each stimulus are random but in Fig. 2A they form a spatially-structured pattern (i.e., horizontal bars). However, if the reviewer still thinks that the re-ordering would help clarity, we would be happy to change the figure accordingly.

Reviewer Point P 1.6 — Fig 1C. I have difficulties understanding the cumulative distributions. Top graph, why is there a vertical line at 110, and 150, respectively for the two distributions? The vertical lines should not be there. Also, what is the horizontal axis? I did not see it clearly defined in the caption (I also searched in methods, but could not find it there, probably my mistake). It should be explained in the caption.

Reply: We have removed the vertical lines from all cumulative distribution plots and we have modified their respective captions as well as the labels of their horizontal axes in order to clarify their meaning (see Fig. 1C, Fig. 3B, and Fig. S11 in the revised manuscript).

Reviewer Point P 1.7 — Fig 2. Shift to supplementary.

Reply: We have split Fig. 2 in the original manuscript into two and moved both to the Supplementary Material: Fig. S6 and Fig. S7 in the revised manuscript (see also related point P 1.1).

Reviewer Point P 1.8 — Fig 4. Shift to supplementary.

Reply: We have moved Fig. 4 in the original manuscript to the Supplementary Material: Fig. S18 in the revised manuscript (see also related point P 1.1).

Reviewer Point P 1.9 — Fig 5A. Are these inhibitory engram neurons in cortex? Please specify in the caption.

Reply: We have specified in the caption of Fig. 3A in the revised manuscript that this panel shows the recall of inhibitory engram cells in HPC, THL, and CTX.

Reviewer Point P 1.10 — Fig 6B. Population activity: what is the white space close to/above the x-axis?

Reply: The white space in the CTX population activity plot of Fig. 4B in the revised manuscript indicates that the population activity of CTX engrams is continuously above 0 for a short period even when it is computed every 10 ms without smoothing or convolution (see page 17, line 709 in the revised manuscript). This is a consequence of blocking the inhibitory neurons in CTX: excitatory engram cells

exhibit periods of very high activity in the absence of inhibition due to recurrent excitatory interactions. We have added this explanation in the caption of Fig. 4 in the revised manuscript.

Reviewer Point P 1.11 — Why do you use an update step of 0.1ms? Can you re-run with an update step of 1ms and compare the results? My prediction is that the result is basically the same but with a simulation that runs 9 times faster.

Reply: We used an update time step $\Delta = 0.1$ ms because this is the default integration time step in the C++ library we used to simulate large-scale spiking neural networks (i.e., Auryn [3]) and previous computational studies employing Auryn have also used the default time step [4]. Even though we have not optimized Δ in our simulations, we performed many preliminary runs to optimize the number of processes over which we split each individual simulation, N_{ranks} , taking advantage of the fact that Auryn uses Message Passing Interface (MPI) for parallel computing [3]. We experimented with $N_{ranks} = 2, 4, 8, 16,$ and 32 and found that $N_{ranks} = 16$ led to the shortest simulation times. Accordingly, all our simulations use $N_{ranks} = 16$ and we have included this in the Methods section (see page 17, line 712 and Table 1 in the revised manuscript). Furthermore, we have tried re-running our simulations with the two-region network (Fig. 1 in the revised manuscript) with $\Delta = 1$ ms but the simulations froze due to a known issue in Auryn associated with communicating large messages with MPI (https://fzenke.net/auryn/doku.php?id=manual:known_issues). Precisely, using $\Delta = 1$ ms increases the number of spikes in each time step by a factor of 10 relative to the default value and, hence, larger messages need to be communicated among MPI processes. This seems to cause a buffer overflow from which the MPI processes cannot recover and there doesn't seem to be any solution to this issue yet.

Reviewer Point P 1.12 — I appreciate that you give all the parameters of the model (many, indeed). Can you add in the column of the “value” an indication (say, a label 1, or 2, or 3) that indicates if: (1) the value is taken from literature (e.g., the time constant of the IF neuron), ideally with a citation. (2) the value was chosen somewhat arbitrarily and set for convenience to this value (e.g., size of network). (3) The value has been optimized over several preliminary runs to achieve the results shown in the paper. Otherwise the reader may get the impression that you optimized about 100 parameters...

Reply: We have included a label indicating the source of each parameter value in Table 1 in the revised manuscript. Specifically, we have used the following labels in line with the reviewer's suggestions: (a) for values taken from [4]; (b) for values chosen somewhat arbitrarily without targeted optimization; and (c) for values optimized over several preliminary simulations.

Reviewer Point P 1.13 — In the Results, you state at multiple occasions: “This is driven by the t.p.r.”. I cannot understand the logic of the sentence. The network simulation gets no information on what is a true and what is a false response, therefore the error cannot drive anything. Probably you simply mean: “This is also reflected by the t.p.r.” (?)

Reply: We agree with the reviewer and we have replaced the phrase “driven by” in every instance of “driven by the t.p.r.” or “driven by the f.p.r.” with “reflected in” (see page 3, line 96; page 5, line 182; page 6, line 210; page 6, line 223; page 7, line 262; and page 7, line 262 in the revised manuscript).

Reviewer Point P 1.14 — In the Results, you state several times “coupled reactivations” etc. It seems that the reader has to infer this from visual inspection of the activation pattern.

The scientific way of doing this would be to measure covariance of activity across two areas, as a function of delay τ , which would then also indicate a time scale of correlation. May be there IS in fact also a correlation on the millisecond time scale? See also major comment above.

Reply: We have included an analysis of the degree of coupling between engrams in different regions as suggested and verified that engrams in the HPC→THL→CTX circuit display coupled reactivations on the millisecond timescale. Specifically, we have I) plotted cross-correlograms by computing the correlation between the population activity of engram cells encoding a stimulus in one region and the population activity of engram cells encoding either the same or another stimulus in a different region as a function of the displacement of one relative to the other, and II) computed the lag between the activity of engrams in two different regions, lag_{max} , as the displacement of the engram activity in one region relative to the other such that the correlation between them is maximized (see page 17, line 701 in the revised manuscript). As a result, we have verified that engrams in the two-region network are decoupled (Fig. S1 in the revised manuscript) while engrams in the three-region network are coupled (Fig. S13 and Fig. S20 in the revised manuscript). We have also verified that engram reactivations become decoupled across the HPC→THL→CTX circuit when excitatory engram cells in either HPC or THL are blocked during consolidation (Fig. S17 in the revised manuscript) and when inhibitory neurons in either HPC, CTX, or THL are blocked during consolidation (Fig. S22 in the revised manuscript).

Reviewer 2

Major Points

Reviewer Point P 2.1 — On line 569, the authors write “engram cell ensemble encoding a given stimulus is taken as activated upon presentation of a partial cue if its population firing rate is above the threshold”. What do the authors mean by population firing rate? Is that an average computed across all engram cells constituting the ensemble? If so, this might be problematic as a small subpopulation firing at rates sufficiently high might lead to the false conclusion that the entire ensemble is being reactivated, essentially giving a false illusion of pattern completion. In particular, the subpopulation aforementioned could be the engram cells directly reactivated by the partial stimuli through feedforward connections. The authors should provide in a supplementary figure data showing how much of the labeled engram ensemble is being reactivated, both right after training and post-consolidation.

Reply: We have clarified the definition of population firing rate of an engram cell ensemble and we have included figures quantifying the extent to which engram cells encoding a given stimulus are reactivated by partial cues both at the end of training and after systems consolidation. First, we have specified that by “population firing rate” of an engram cell ensemble we refer to the “average firing rate computed over all engram cells in a given ensemble” (see page 16, line 682 in the revised manuscript). Second, we have plotted the fraction of engram cells in each region that was reactivated by a partial cue immediately after training (Figure S9 in the revised manuscript) and after 24 hours of consolidation (Figure S10 in the revised manuscript). We saw that presenting a partial cue right after training reactivates the majority of engram cells encoding the corresponding stimulus in both HPC and THL but not in CTX since CTX engram cells are still silent at the end of training (see Fig. 2C in the revised manuscript). However, presenting the same partial cue after 24 hours of consolidation reactivates the majority of engram cells

encoding the corresponding stimulus in both CTX and THL but not in HPC because HPC engram cells are silent post-consolidation (see Fig. 2C in the revised manuscript).

Reviewer Point P 2.2 — In Figure 3C, I could not explain to myself why THL recall accuracies were not affected by the consolidation. Figure 4a,b show that, similar to HPC (Figure 1C), the weight of recurrent connections in THL “deteriorate” through the consolidation process. However, the recall accuracy of HPC at test time is significantly worse than THL’s, suggesting that the difference resides in the fact that the authors use static feedforward STIM→THL projections. Please justify this choice, as well as explain its impact on THL recall accuracy at test time.

Reply: We have included a detailed discussion of the reasoning behind making STIM→THL synapses static as well as its impact on THL recall in the Methods section (see page 15, line 602 in the revised manuscript). Here, we summarize and elaborate on that discussion. First, our results with the two-region network showed that changes in feedforward but not in recurrent synapses underlie changes in engram cell state (i.e., silent-to-active or active-to-silent, see Fig. 1 in the revised manuscript). Second, previous experiments have suggested that THL engram cells are active in both recent and remote recall (see page 15, line 602 in the revised manuscript). Consequently, setting STIM→THL projections to be static supports continuously active THL engram cells. As expected, our simulations showed active THL engrams in both recent and remote recall in a manner consistent with previous experimental reports (Fig. 2 in the revised manuscript.) Thus, the high level of recall accuracy in THL is maintained throughout systems consolidation due to the stability of the STIM→THL synapses and despite changes in THL recurrent synapses (Fig. S18 in the revised manuscript).

Reviewer Point P 2.3 — Could the authors explain why they did not silence specifically the inhibitory engram cells rather than blocking all inhibitory inputs when proving their critical role for CTX engram maturation (Figure 6)? The authors should address this point and, in the case where the same results can be obtained with specific engram silencing, should replace Figure 6 with the new data. If the results are different, it should be mentioned in the text and the data should appear in a supplementary figure.

Reply: We originally blocked all inhibitory neurons in a given region to align our simulations with previous experiments that blocked parvalbumin-positive interneurons in either HPC or CTX irrespective of their response to the conditioning stimulus (see page 7, line 302 in the revised manuscript). However, we have now also blocked specifically inhibitory engram cells in either HPC, CTX, or THL and found that this did not prevent CTX engram maturation since the remaining unblocked inhibitory neurons in each case were still able to control and couple engram reactivations via strong potentiation of their inhibitory synapses onto excitatory engram cells (Fig. S24 in the revised manuscript). Note that blocking inhibitory engram cells leads to a more pronounced potentiation of inhibitory synapses onto excitatory engram cells over the course of consolidation than in the control case when no inhibitory neurons are blocked (compare Fig. S24A/C/E to Fig. 3B in the revised manuscript). This is expected since a smaller number of inhibitory neurons are active after blocking inhibitory engram cells and, hence, the inhibitory synapses of the inhibitory neurons that remain active have to be stronger to control the activity of the excitatory neurons. In the case of THL, this effect is even more apparent since inhibitory synapses onto excitatory engram cells are actually depressed in the control case with no blocking but are strongly potentiated when blocking inhibitory engram cells. These results suggest that inhibitory engram cells may be replaced by originally unresponsive inhibitory neurons as long as the latter can potentiate their

inhibitory synapses to an extent that engram reactivations can still be controlled and coupled across the network (see page 8, line 307 in the revised manuscript). These new simulations also highlight the compensatory and stabilizing nature of inhibitory synaptic plasticity in the model as it promotes a balance of excitation and inhibition that is resilient to perturbations like blocking inhibitory engram cells.

Reviewer Point P 2.4 — “Engram cells are a population of neurons that are activated by learning, have enduring cellular changes as a consequence of learning, and whose reactivation by a part of the original stimuli delivered during learning results in memory recall” (Tonogawa et al. 2015). In neuroscientific studies, the reactivation of the engram cells through the presentation of partial cues, which is the core of the paper under review, is never considered definitive proof for memory recall but only a correlation. Instead, the sole artificial reactivation of the engram cells in the absence of relevant cues must elicit the same behavior as natural recall would upon pattern completion. While I understand that extending the study past simple stimuli imprinting to include behavioral outputs might be too much, this shortcoming should be discussed and the authors should detail how one could go about addressing this issue.

Reply: We have performed new simulations to explore how behavioral output can be incorporated to our model and we have included a discussion of this issue (see page 11, line 482 in the revised manuscript). Specifically, while our original model does not include behavioral outputs and, hence, cannot capture the fact that artificial reactivation of engram cells should elicit the same behavior as would the presentation of stimulus partial cues, we have extended our three-region network by adding a shared downstream readout region (RDT) that can be interpreted as a behavioral output (see Fig. S16 in the revised manuscript). In particular, artificial reactivation of engram cells in any stimulus-encoding region (i.e., HPC, THL, and CTX) can elicit the same response in RDT as does the presentation of matching stimulus partial cues due to strong synaptic coupling between each stimulus-encoding region and RDT. Furthermore, the extended three-region network with RDT supports stable reactivation of engram cells in RDT upon presentation of partial cues throughout consolidation (see Fig. S16 in the revised manuscript). Consequently, RDT in this network configuration could be interpreted as a very simplified model of stable behavioral output.

Minor Points

Reviewer Point P 2.5 — The author should include references providing anatomical evidence motivating the use of recurrent neural networks to model HPC, CTX, and THL. Additionally, the authors should explain why, in their three regions model, $STIM \rightarrow HPC$ is plastic while $STIM \rightarrow THL$ is static.

Reply: We have included a detailed discussion of both I) the anatomical evidence motivating the use of recurrent neural networks (RNNs) to model the brain regions in our network, and II) the reasoning behind the presence or absence of plasticity in $STIM \rightarrow HPC$ and $STIM \rightarrow THL$ synapses. Specifically, we have listed a number of experimental reports that found recurrent synapses in HPC, CTX, and THL and, hence, motivated our use of RNNs to model these regions (see page 14, line 586 in the revised manuscript). In addition, we have discussed the reasons why we have plastic $STIM \rightarrow HPC$ and static $STIM \rightarrow THL$ projections in our model (see page 15, line 602 in the revised manuscript). In particular, our simulations with the two-region network revealed that changes in feedforward but not in recurrent

synapses underlie changes in engram cell state (i.e., silent-to-active or active-to-silent, see Fig. 1 in the revised manuscript). Furthermore, while it has been shown that HPC engram cells undergo de-maturation (i.e., switch from active to silent) [1], experiments probing the role of THL in recent [5] and remote [6] recall suggest that THL engram cells are active in recall at both time points (see page 15, line 602 in the revised manuscript). Therefore, we set STIM→HPC synapses to be plastic to enable HPC engram de-maturation and set STIM→THL projections to be static to support active THL engram cells in recent and remote recall in a manner consistent with previous experimental reports. Our results showed that engram cells in HPC and THL behave as expected in our three-region network model with plastic STIM→HPC and static STIM→THL synapses (see Fig. 2C in the revised manuscript, and related point P 2.2).

Reviewer Point P 2.6 — Could the authors explain why (i) they changed stimuli (full and partial) between the two regions analysis and the three regions analysis (e.g. Figure 1a and Figure 3a), and (ii) modified the patterns of feedforward connections (i.e. introduction of circular receptive fields) between these two experiments? Additionally, the authors should make it clear in the manuscript how these changes might affect the outcome of their experiments.

Reply: We have changed stimuli and the pattern of feedforward STIM projections between the two- and the three-region models in order to facilitate learning in THL when STIM→THL synapses are static. Specifically, we have set static STIM→THL projections to support active THL engram cells in both recent and remote recall in a manner consistent with previous experiments (see page 15, line 602 in the revised manuscript, and related point P 2.2). Given this choice to have static STIM→THL synapses, we used training and testing stimuli with a spatial structure (i.e., horizontal bars in Fig. 2A in the revised manuscript) that could be explored by circular receptive fields in the STIM→THL projections with the goal of facilitating learning in THL (see page 15, line 612). Conversely, learning accurate THL representations of random stimuli (as in Fig. 1A of the revised manuscript) with randomly-connected static STIM→THL projections is challenging. In this scenario, the separation of timescales between subcortical regions in our model (i.e., HPC and THL) and the cortical region (i.e., CTX) would have to be significantly increased to allow the emergence of accurate THL engrams at the end of training while CTX engrams are still silent. Therefore, we changed stimuli and the configuration of STIM feedforward projections in the three-region network to facilitate learning in THL in the presence of static STIM→THL synapses and, consequently, reduce the required gap between plasticity timescales in subcortical vs. cortical regions in our model. Importantly, these changes were motivated by the assumption that STIM→THL synapses only change over developmental timescales much longer than the ones covered by our model and, hence, these synapses are static in our simulations. We have included a summary of the discussion above on page 11, line 476 in the revised manuscript to highlight the impact of this set of changes between the two- and three-region networks (i.e., static STIM→THL, non-random stimuli, and STIM feedforward projections with circular receptive fields) on our results.

Reviewer Point P 2.7 — While the difference in engram dynamics observed between HPC and CTX can simply be explained by their diverging synaptic plasticity timescales (Figure 1), the authors show that a third region (THL in their model) is required to recapitulate findings showing that the HPC engram is necessary for CTX engram maturation (Figure 3). I think the manuscript would benefit from a discussion addressing why would the brain require such a third party if the end goal, namely maturation of a cortical engram, can be achieved without it.

Reply: We agree with the reviewer and we have performed new simulations that suggest that the brain may recruit an intermediary region between HPC and CTX to support stable recall in a shared downstream readout region. Specifically, we have extended our two- and three-region models by adding a shared downstream readout region (RDT) (see Fig. S15 and Fig. S16 in the revised manuscript). Our simulations revealed that the decoupled engram reactivations in the extended two-region network could not sustain recall in RDT but the coupled engram reactivations in the extended three-region network could maintain stable recall in RDT throughout consolidation. This suggests that the role of an intermediary region (THL in our model) between HPC and CTX may be to enable coupled engram reactivations across these regions to support stable recall in a shared readout region. Interestingly, the basolateral amygdala has parallels with RDT in our model: both receive projections from HPC and CTX and maintain active engram cells throughout consolidation (see page 5, line 197 in the revised manuscript). Thus, our new results suggest a computational role for an intermediary region between HPC and CTX that could justify why the brain would recruit such a third party if cortical engram maturation can be achieved in its absence.

Reviewer Point P 2.8 — In line with the previous question, I was wondering if there might be a qualitative difference between mature cortical engrams formed in the two regions system (Figure 1), and the similar engram maturing with the support of THL and HPC (Figure 3)? Maybe these two ensembles can be compared past their recall accuracy.

Reply: We have performed new simulations that have identified a functional difference between the cortical engrams in the two- and three-region models. As discussed in the previous point P 2.7, we have added a shared downstream readout region (RDT) to the two- and to the three-region networks and probed their ability to sustain stable recall in RDT (see Fig. S15 and Fig. S16 in the revised manuscript). Cortical and hippocampal engrams in the extended two-region model exhibited decoupled engram reactivations and failed to maintain recall in RDT. However, cortical, thalamic, and hippocampal engrams in the extended three-region model showed coupled engram reactivations and sustained stable recall in RDT throughout consolidation. Therefore, although cortical engrams can be recalled in both the two- and three-region networks, only the cortical engrams in the three-region model have reactivations coupled with those of engrams in the other regions and, hence, support stable recall in a shared readout region (see page 5, line 197 in the revised manuscript).

Reviewer Point P 2.9 — The authors could discuss how their model helps understand other competing theories (e.g. multiple trace theory).

Reply: We have included a discussion of how our model relates to previous theories of systems consolidation with a particular focus on multiple trace theory and trace transformation theory (see page 11, line 503 in the revised manuscript).

Reviewer 3

Major Points

Reviewer Point P 3.1 — The stimuli used for encoding episodic memory are overly simple and non-overlapping.

Reply: We have included new simulations where we trained our network model on overlapping stimuli and verified that they exhibit engram dynamics analogous to our original results with non-overlapping stimuli (see page 3, line 100 and Fig. S3 in the revised manuscript). Specifically, we trained our network on a set of random stimuli where each stimulus consisted of a random 25% of the neurons in the stimulus population resulting in an average overlap of 25% between two different stimuli (Fig. S3A in the revised manuscript). We then tested memory recall with partial cues consisting of a random 50% of the corresponding full stimulus. Our results showed that CTX engrams undergo maturation and HPC engrams are subject to de-maturation as evidenced by their respective true positive rate (t.p.r.) curves (Fig. S3D in the revised manuscript) in a manner analogous to the network trained on random non-overlapping stimuli (Fig. 1E in the revised manuscript). Notably, the false positive rate (f.p.r.) of CTX engrams in the case of overlapping stimuli does not settle at near-zero as it did with non-overlapping stimuli (compare Fig. S3E and Fig. 1F in the revised manuscript). This resulted in increased variability in CTX recall accuracy after maturation of its engrams in the case of overlapping stimuli (compare Fig. S3C and Fig. 1D in the revised manuscript). However, engram cells in CTX are initially silent and become active with consolidation in both cases (compare Fig. S3D and Fig. 1E in the revised manuscript). Therefore, CTX engrams exhibited maturation with both overlapping and non-overlapping training stimuli. Note that currently there are no experiments describing how multiple engrams encoding different tasks acquired close in time evolve over the course of systems consolidation — only experiments tracking engrams of a single task over weeks are available [1]. As a result, we used non-overlapping stimuli in our original simulations to minimize the overlap between engrams encoding different stimuli in the same region (see Fig. S19 in the revised manuscript) and, consequently, minimize their interactions in an attempt to capture how individual engrams evolve as reported in the literature. Interestingly, our new simulations with overlapping stimuli suggest that engrams are still subject to maturation and de-maturation in this configuration but memory recall accuracy becomes less reliable after systems consolidation — a prediction that can be tested in future experiments.

Reviewer Point P 3.2 — The feedforward anatomy of these 2 or 3 recurrent neural networks (RNNs) does not reflect the complex anatomical interactions between the hippocampus and cortex. For example, the 1994 Alvarez and Squire model of memory consolidation had multiple cortical areas with bidirectional connections between the cortex and the MTL. Bidirectional coupling, reentrant connectivity, and associating different regions of cortex with a memory are key to episodic memory.

Reply: We have expanded our model circuit to include multiple cortical regions and reentrant connectivity and showed that this configuration also exhibits engram dynamics consistent with previous experiments but with bidirectional synaptic coupling between hippocampal and cortical regions. Precisely, Fig. S14 in the revised manuscript shows that an expanded circuit with two cortical regions with reentrant connectivity, an indirect multisynaptic pathway from hippocampus to cortex, and direct monosynaptic projections from cortex to hippocampus exhibits engram dynamics (i.e., maturation of cortical engrams and de-maturation of hippocampal engrams) and coupled engram reactivations in a manner analogous to the original three-region model (Fig. 2 in the revised manuscript). Therefore, even though our original model circuit did not capture the full range of anatomical interactions between hippocampus and cortex, our expanded network brought our model closer to the brain circuitry involved in episodic memory by incorporating multiple cortical regions, reentrant connectivity, and bidirectional coupling between hippocampus and cortex. We also acknowledge that even our expanded model circuit does not yet capture the complete range of anatomical interactions between hippocampus and cortex,

but the impact of this limitation is lessened when we consider that the mechanism our model proposed to elucidate how brain regions can interact to promote systems consolidation (i.e., coupled engram reactivations) is present in both a minimal (Fig. 2 and Fig. S13 in the revised manuscript) and a more comprehensive (Fig. S14 in the revised manuscript) circuit model of systems consolidation. Specifically, our combined modeling results suggest that the coupling of engram reactivations across brain regions may be a generic mechanism to support the engagement of multiple regions in the encoding, consolidation, and recall of memories (page 5, line 192 in the revised manuscript). Hence, the level of complexity in our model circuits offers the advantage of rendering simulations computationally tractable while still preserving the ability of the model to offer insights into how multiple brain regions interact to support episodic memory.

Reviewer Point P 3.3 — The dynamics of the model is not analyzed to show that there is oscillatory coupling between regions. It is claimed, but there should be analyses and quantitative metrics showing coupling and synchrony between areas. The current thinking is that there is synchrony between sharp wave ripples in the hippocampus and spindles in the cortex. Does the present model show evidence for that?

Reply: We have quantified the degree of coupling between regions in the model and we have included a discussion of how engram dynamics in our model compare to oscillations associated with systems consolidation. First, we have measured the degree of coupling between pairs of regions in the model by plotting the cross-correlogram of their engram activity and extracting the lag between them (see Fig. S1, Fig. S13, Fig. S14, Fig. S15, Fig. S16, Fig. S17, Fig. S20, Fig. S22, and Fig. S24 in the revised manuscript). We have included a detailed description of these cross-correlograms and the computation of the associated lags (see page 17, line 701 in the revised manuscript). Second, we have discussed how our work does not explicitly model sharp-wave ripples, spindles, and slow oscillations but instead captures the fact that the coupling of these oscillations has been linked to coupled engram reactivations across brain regions (see page 11, line 470). Future work could build on our model to incorporate these stereotypical oscillations associated systems consolidation.

Reviewer Point P 3.4 — The results are not directly compared with existing experimental evidence.

Reply: We agree with the reviewer that our modeling results are not directly compared to available experimental data and we have included a discussion of this limitation (see page 11, line 463 in the revised manuscript). Specifically, systems consolidation takes place over very long timescales (i.e., from days to years) and simulating such long periods is impractical due to the very large scale of our network models and the resulting very long simulation times. Despite this limitation, our modeling results are consistent with the available experimental reports regarding engram dynamics underlying systems consolidation (see page 11, line 468 in the revised manuscript).

General Comments

Reviewer Point P 3.5 — In general, I found the results section difficult to read. The figures are extremely busy with multiple panels that are difficult to navigate. The captions are hard to digest with A/G, E/K and J/L referred to as being related. The wording of the results points to these panels and also make it hard to follow the theme or idea being made.

Reply: We have moved figures to the Supplementary Material, split figures, and modified figure captions to facilitate reading the Results section. Specifically, we have moved Figure 2 in the original manuscript to the Supplementary Material and split it into Figure S6 and Figure S7 in the revised manuscript. We have also merged Figure S1A-C in the original manuscript to Figure S6 in the revised manuscript. Consequently, Figure S1D-J in the original manuscript became a separate Figure S8 in the revised manuscript. Furthermore, Figure 4 in the original manuscript was moved to the Supplementary Material as Figure S18 in the revised manuscript and we have also merged into it Figure S6 of the original manuscript. Importantly, the above reorganization of figures (i.e., moving to Supplementary Material, splitting, and merging) was done to streamline the Results section and, hence, refocus the presentation of results on the key findings of our paper. Lastly, we have changed the caption of the following figures in the revised manuscript to ensure each panel is described separately: Figure 5, Figure S6, Figure S7, Figure S8, and Figure S21. These changes hopefully facilitate navigating each figure individually and, consequently, help keep the focus on main ideas when referencing individual figure panels in the Results section.

Reviewer Point P 3.6 — I am not sure what makes these areas the cortex, hippocampus, and thalamus. The connectivity of these 3 RNNs is not necessarily close to the anatomy from these regions. THL in this case is just an intermediary between the HPC RNN and the CTX RNN. The projection to the THL is static.

Reply: We agree with the reviewer that the RNNs in our model do not necessarily represent cortex, hippocampus, and thalamus and, hence, we have included a discussion of this feature in the revised manuscript. Specifically, we have acknowledged that we assigned the labels of CTX, HPC, and THL to the RNNs in our model with the goal of matching the engram dynamics in each RNN to previous experimental findings (see page 9, line 388 in the revised manuscript). However, we have also pointed out that the generic nature of the RNNs in our model lends our results agnostic to specific regions and, hence, offers computational insights into how diverse engram dynamics may coexist in multiple regions to support episodic memory. Furthermore, we have provided several arguments to support the claim that the RNN between HPC and CTX in our model matches THL (see page 9, line 366 in the revised manuscript). Nonetheless, we recognize that THL may not be the only candidate intermediary region recruited by HPC to support the maturation of CTX engrams (see page 9, line 375 in the revised manuscript). Lastly, the choice of static STIM→THL synapses was motivated by I) our modeling results showing that changes in feedforward synapses underlie engram cell state transitions (i.e., silent-to-active or active-to-silent), and II) previous experimental findings suggesting that THL engram cells are active in both recent and remote recall (see page 15, line 602). As a result, static STIM→THL are set to capture the active involvement of THL at recent and remote time points as reported in the experimental literature.

Reviewer Point P 3.7 — A strength of this model is the biologically plausible synaptic plasticity. How do the “Hebbian (i.e., triplet STDP) and non-Hebbian (i.e., heterosynaptic and transmitter-induced) forms of plasticity” all contribute to the present results. Are all of these necessary? Are some more important than others?

Reply: We have included new simulations showing that each Hebbian and non-Hebbian form of synaptic plasticity in our model is essential to reproduce engram dynamics observed in experiments (see Fig. S4 and page 4, line 108 in the revised manuscript). Specifically, we blocked the triplet STDP, heterosynaptic,

and transmitter-induced forms of plasticity one at a time and verified that this disrupts engram dynamics in the network in each case. First, blocking triplet STDP led to silent engram cells in both CTX and HPC at the end of training and after 12 hours of consolidation (Fig. S4A in the revised manuscript) because this prevented the formation of engram cell ensembles with strong recurrent excitatory weights within an ensemble encoding the same stimulus relative to the weights between engram cells encoding different stimuli (i.e., no significant distinction between on- and off-diagonal weights in Fig. S4B in the revised manuscript). As a result, although engram cells in CTX and HPC initially become active with consolidation, they eventually turn silent again due to indiscriminate potentiation of inhibitory synapses onto engram cells as evidenced by the declining true positive rate in these regions (Fig. S4A in the revised manuscript). These results were expected since in the absence of triplet STDP there is no Hebbian learning in the model and, hence, potentiation of excitatory synapses is non-specific (see Eq. 10a in the revised manuscript). Second, blocking heterosynaptic plasticity prevented accurate recall of engram cells in both CTX and HPC (Fig. S4C in the revised manuscript). In this case, engram cells are active in both regions at the end of training and throughout consolidation but presenting partial cues of one stimulus leads to reactivation of the engrams encoding all stimuli as evidenced by the 100% true positive rate and false positive rate in these regions. This is a result of large and indiscriminate potentiation of recurrent excitatory weights in both CTX and HPC (Fig. S4D in the revised manuscript). This was also expected since in the absence of heterosynaptic plasticity potentiation due to triplet STDP is left unchecked (see Eq. 10b in the revised manuscript). Third, blocking transmitter-induced plasticity prevented engram maturation in CTX because this disrupts engram reactivations in this region and, consequently, prevents the potentiation of STIM→CTX synapses that would have led to active CTX engram cells (Fig. S4E in the revised manuscript). This is consistent with previous results that showed that blocking transmitter-induced plasticity impairs engram reactivations [4]. Importantly, the previous work that proposed combining triplet STDP, heterosynaptic plasticity, and transmitter-induced plasticity to model memory in a single-region spiking recurrent network showed that blocking any of these forms of plasticity also disrupted stable memory formation and recall in their network [4]. Furthermore, a mean-field theory incorporating this combination of plasticity mechanisms also supports their essential role for stable memory [4]. Taken together, our new simulations described above demonstrate that each Hebbian and non-Hebbian form of plasticity in our model is crucial to reproduce experimentally-observed engram dynamics by impacting the evolution of engram cells in specific ways.

Reviewer Point P 3.8 — The figures show that different neurons or synapses are contributing to encoding the stimuli (engrams). There are a number of ablation simulations to show the various contributions. But there should be some explanatory dynamics. Especially for the excitatory and inhibitory contributions. Could you vary the E/I ratios? Or could you lesion the different types of synaptic plasticity rules to see which ones are contributing to the learning?

Reply: We have included new simulations that showed that the engram dynamics in our model are robust to the choice of E/I ratio due to inhibitory synaptic plasticity and that each form of Hebbian and non-Hebbian plasticity in our model is essential to reproduce experimentally-observed engram dynamics. Specifically, we have halved and doubled the E/I ratio of 4 used in our original simulations to probe the sensitivity of our results to this ratio and to gain insights into the role of inhibitory synaptic plasticity in the engram dynamics in our model (see Fig. S5 and page 4, line 109 in the revised manuscript). With the baseline E/I ratio of 4, the network exhibits engram maturation in CTX and engram de-maturation in HPC with inhibitory synaptic plasticity acting to control the activity of excitatory neurons in each region of the network while enabling engram reactivations in both CTX and HPC (Fig. S5A-B in the

revised manuscript). With an E/I ratio of 2 or 8, similar engram dynamics are observed in the network but the changes in inhibitory weights over the course of consolidation are more pronounced than in the baseline case (compare Fig. S5C-D/E-F to Fig. S5A-B in the revised manuscript). This means that inhibitory synaptic plasticity flexibly modifies inhibitory weights onto engram cells depending on the E/I ratio to ensure the network activity is kept under control (target firing rate $\gamma = 4$ Hz) while still enabling engram reactivations. Furthermore, we have blocked each Hebbian (i.e., triplet STDP) and non-Hebbian (i.e., heterosynaptic and transmitter-induced) form of plasticity in our model individually and showed that each is crucial for reproducing engram dynamics observed in experiments (see Fig. S4 and page 4, line 108 in the revised manuscript). For a detailed discussion of the impact of each form of plasticity on the evolution of engram cells in the network, please refer to the related previous point P 3.7.

Reviewer Point P 3.9 — The authors use “coupling”, “oscillations” and “coordinated communication” in their language. Line 418 states, “our model captures the fact that oscillations in HPC, THL, and CTX need to be coupled for effective systems consolidation by displaying coupled engram cell reactivations.” But this should be backed up with an analysis of the synchrony and oscillatory dynamics. I did not see such an analysis. The simulated ablations are not enough.

Reply: We have included an extensive analysis of the coupling of engram reactivations between regions in our model (see Fig. S1, Fig. S13, Fig. S14, Fig. S15, Fig. S16, Fig. S17, Fig. S20, Fig. S22, and Fig. S24 in the revised manuscript; see also related point P 3.3). We have also provided a detailed description of how we evaluated engram reactivation coupling (see page 17, line 701 in the revised manuscript).

Reviewer Point P 3.10 — The results are mostly qualitative. Quantitative statistics should be provided for all the results. For example, the weight matrices and memory recall panels that are in most figures need statistics to back up the claims being made.

Reply: We have added quantitative statistics to all plots of memory recall, mean weights, and weight distributions (see Fig. 1C-F, Fig. 2C-E, Fig. 3A-B, Fig. 4A-C, Fig. 5B-J, Fig. S3C-E, Fig. S4A-F, Fig. S5A-F, Fig. S6C/D/F/G-I, Fig. S7C/D/F, Fig. S8C-E/G, Fig. S11A-B, Fig. S12C, Fig. S14C, Fig. S15C/F, Fig. S16C/F, Fig. S18A-D, Fig. S23A-C, Fig. S24A/C/E, and Fig. S25B-G in the revised manuscript). We have also quantified the degree of coupling between engrams in different regions of our model (see Fig. S1, Fig. S13, Fig. S14, Fig. S15, Fig. S16, Fig. S17, Fig. S20, Fig. S22, and Fig. S24 in the revised manuscript; see also related point P 3.3). For a description of the statistical analysis we have performed, please see page 17, line 697 in the revised manuscript.

Specific Comments

Reviewer Point P 3.11 — Line 64. “One of four non-overlapping random stimuli is presented to the network at a time either for training or testing.” What would happen if there were overlapping stimuli? This seems to be a more rigorous and realistic test of memory consolidation.

Reply: We have performed additional simulations with overlapping stimuli and found that CTX engrams exhibit maturation and HPC engrams display de-maturation in a manner similar to our original

simulations with non-overlapping stimuli (see page 3, line 100 and Fig. S3 in the revised manuscript). For a discussion of the new simulations with overlapping stimuli, please see related point P 3.1.

Reviewer Point P 3.12 — Figure 1C and 2C. There is no overlap for the weights. Give more details on how they were chosen for the figure. Is it just the recurrent weights into the Engram cells? What about the input weights from the Stimuli?

Reply: We have modified the caption of Fig. 1C, Fig. S6C/G, Fig. S7C, and Fig. S8C-D in the revised manuscript to clarify that the mean weight plots only include weights between engram cells (i.e., pre- and post-synaptic neurons are engram cells). We have also modified the label of the x axis in the cumulative distribution plots of Fig. 1C in the revised manuscript to clarify that they refer to input weights from the stimulus population.

Reviewer Point P 3.13 — Section: “Subcortical engram cells are essential for CTX engram cell maturation.” The assumption is that only vHPC has direct connections to the CTX and this amounts to only 5% of the connections. But are you assuming that 95% are then coming from the THL? Is that a correct assumption? What about connections from CTX to HPC?

Reply: We have included a discussion acknowledging that while we examine the role of a specific circuit involving THL in systems consolidation of memory there are multiple alternative pathways from HPC to CTX that could contribute to this process (see page 9, line 365 and page 9, line 375 in the revised manuscript). In particular, we list three candidate regions that have a similar level of monosynaptic projections to CTX engram cells as THL and, hence, may also support communication between HPC and CTX: anterodorsal thalamus, medial entorhinal cortex layer Va, and basolateral amygdala (page 9, line 377 in the revised manuscript). We also discuss the possibility that a multisynaptic pathway along the dorsoventral axis of the hippocampus may be recruited for HPC-CTX communication (page 9, line 381 in the revised manuscript). Lastly, we emphasize that our model suggests that the HPC→THL→CTX pathway is a plausible *minimal* circuit for systems consolidation (page 9, line 387 in the revised manuscript). With regards to connections from CTX to HPC, we have performed new simulations with an extended circuit that includes CTX→HPC synapses and we verified that it also displays engram dynamics consistent with previous experiments: cortical engram maturation and hippocampal engram de-maturation (Fig. S14). For a discussion of the simulations with this extended circuit, please see related point P 3.2.

Reviewer Point P 3.14 — Figure 3. If I am understanding this correctly, the THL has perfect knowledge of the stimuli. The accuracy is 100%, the t.p.r is 100%, and the f.p.r is 0. Is this reasonable? Why would the HPC and CTX be necessary if there is perfect memory in the THL?

Reply: We have listed previous experiments that suggest that THL engram cells are active throughout consolidation and we have performed new simulations that suggest that THL acts in coordination with HPC and CTX to support stable recall in a shared downstream readout region. First, previous experiments have suggested that THL engram cells are active in both recent and remote recall (see page 15, line 607 in the revised manuscript) and, hence, the behavior of THL engrams in our model is consistent with these experiments (Fig. 2C in the revised manuscript). Second, to elucidate the potential computational benefits of including THL between HPC and CTX, we have added a shared downstream readout region (RDT) to the two- and three-region networks to probe their ability to support stable

recall in RDT (Fig. S15 and Fig. S16 in the revised manuscript). The extended two-region network displayed decoupled engram reactivations and failed to maintain stable recall in RDT. However, the extended three-region network exhibited coupled engram reactivations and sustained stable recall in RDT throughout consolidation. This suggests that HPC, CTX, and THL act in coordination to support recall in a shared readout region. Notably, RDT in our model has parallels with the basolateral amygdala in contextual fear conditioning: both receive projections from HPC and CTX as well as maintain active engram cells with systems consolidation (see page 5, line 197 in the revised manuscript). Lastly, we have also emphasized that in our original three-region model memory recall requires retrieval in either HPC or CTX in a manner consistent with previous experiments (see page 5, line 169 in the revised manuscript).

Reviewer Point P 3.15 — Line 216. “This results in decoupled oscillations in HPC, THL and CTX (Fig. 3D).” There needs to be an analysis of coupled oscillations to make this claim.

Reply: We have included an analysis of the degree of coupling between regions in the model for the intact (i.e., control) HPC→THL→CTX network (Fig. S13 in the revised manuscript) and the cases where we blocked either HPC or THL engram cells during consolidation (Fig. S17). These analyses showed that engram reactivations are coupled in the intact HPC→THL→CTX circuit but become decoupled when either HPC or THL engram cells are blocked during consolidation. See also related point P 3.3.

Reviewer Point P 3.16 — Section: “Thalamocortical coupling underlies retrograde amnesia profiles.” I don’t see the retrograde amnesia in the reported results. It seems that to show this, the HPC would need to be lesioned at different stages of consolidation to see a graded amnesia depending on consolidation. In fact, Figure 3C (control) is near identical to Figure 7B (graded amnesia). Am I missing something?

Reply: We have I) modified the discussion of the results in the section “Thalamocortical coupling underlies retrograde amnesia profiles” to clarify the interpretation of recall curves with ablated HPC, and II) performed new simulations ablating the HPC at two different points in the consolidation phase with results analogous to our original simulations with HPC ablation in the testing phase (see page 8, line 317; page 8, line 337; and Fig. S25 in the revised manuscript). First, our original simulations with HPC ablation in the testing phase (Fig. 5 in the revised manuscript) followed the same methodology previously employed in computational studies of systems consolidation that examined retrograde amnesia induced by HPC lesion [7]. In this approach, the goal is to compare recall in the intact and lesioned networks at recent and remote recall. In our simulations, successful memory recall requires retrieval in either HPC or CTX independent of retrieval in THL in a manner consistent with previous experiments [1] (see page 5, line 169 in the revised manuscript). Therefore, our intact (i.e., control) network relies on HPC for recall at recent time points but switches to CTX with consolidation (Fig. 2C in the revised manuscript). However, when HPC is ablated in the testing phase the network relies exclusively on CTX for recall and, hence, memories cannot be recalled at recent time points but can be recalled at remote time points (Fig. 5B in the revised manuscript) — thus characterizing graded retrograde amnesia. Similar analysis reveals a flat retrograde amnesia with weak THL→CTX coupling (Fig. 2E-F) and an absent retrograde amnesia with strong THL→CTX coupling (Fig. 2I-J). Furthermore, we have included additional simulations where we ablated the HPC either at the very beginning of the consolidation phase (i.e., immediately after training) or only after 12 hours of consolidation and we observed the same

relationship between the degree of THL→CTX coupling and the pattern of retrograde amnesia induced by HPC lesion (see page 8, line 337 and Fig. S25 in the revised manuscript). Precisely, ablating the HPC at the start of the consolidation phase in the network with baseline THL→CTX coupling prevented high recall rates in both recent and remote time points as evidenced by the resulting CTX recall curve (Fig. S25B in the revised manuscript). However, ablating the HPC in the same network only after 12 hours of consolidation did not disrupt neither recent nor remote recall since recall switched from HPC to CTX before the HPC ablation and CTX was able to sustain high recall rates after the HPC ablation (Fig. S25C in the revised manuscript). Therefore, this network exhibits a graded retrograde amnesia: HPC ablation leads to amnesia only when the lesion occurs shortly after memory encoding (compare Fig. S25B and C in the revised manuscript). Similar analysis of the network with weak THL→CTX coupling reveals that it displays a flat retrograde amnesia (Fig. S25D-E in the revised manuscript) while the network with strong THL→CTX coupling shows an absent retrograde amnesia (Fig. S25F-G in the revised manuscript). Taken together, our original simulations with HPC ablation in the testing phase and our new simulations with HPC ablation in the consolidation phase both suggest that the extent of THL→CTX coupling at the end of memory encoding is a major predictor of the retrograde amnesia pattern induced by HPC damage.

Reviewer Point P 3.17 — Methods section. There are numerous open parameters. How were those values chosen? What is their justification?

Reply: We have included an indication of the source of each parameter value in Table 1 in the revised manuscript. Specifically, we have labelled each parameter value with one of the following labels: (a) for values taken from [4]; (b) for values chosen somewhat arbitrarily without targeted optimization; and (c) for values optimized over several preliminary simulations. Overall, our strategy was to use parameter values available in the literature whenever possible to minimize the number of parameters to be optimized in our work. As a result, we have focused on optimizing synaptic plasticity rates, connection probabilities, and initial weights in the network to reproduce engram dynamics observed in previous experiments [1].

Reviewer Point P 3.18 — Equation 12. The synaptic consolidation time constant is important for the present results. Give more background on how it works? How is it justified by neurobiology? What assumptions were made?

Reply: We have I) expanded the discussion of how Equation 12 models synaptic consolidation, II) provided references to synaptic tagging experiments that motivate this model of synaptic consolidation, and III) listed major assumptions made by this model (see page 13, line 562 to 571).

Reviewer Point P 3.19 — Lines 503-506. Why are there different distributions for the different versions of the model (i.e., uniform for the 2 RNN model and circular RFs for the 3 RNN model)? Couldn't these variations make a difference in the outcome?

Reply: We have used a different configuration for STIM feedforward synapses in the three-region network to facilitate learning in THL when STIM→THL synapses are static. Specifically, we have set STIM→THL synapses static to support active THL engram cells during recall at recent and remote time points in a manner consistent with previous experiments (see page 15, line 602 in the revised manuscript). Given the static nature of STIM→THL projections, we changed both the stimuli and

the initialization of STIM→THL synapses in the three-region network to facilitate learning in THL: we used training and testing stimuli whose non-random spatial structure (i.e., horizontal bars and their central 50% as in Fig. 2A in the revised manuscript) could be explored by circular receptive fields in STIM→THL projections to expedite the learning of accurate engrams in THL (see page 15, line 612 in the revised manuscript). Importantly, even though in theory the same engram dynamics exhibited by our current model could be reproduced using random stimuli and randomly-connected static STIM→THL synapses in the three-region network, the separation of plasticity timescales between subcortical and cortical regions in the model would have to be larger to enable the emergence of accurate THL engrams at the end of training while CTX engram cells are still silent. We have included a discussion of the observations above on page 11, line 476 in the revised manuscript.

Reviewer Point P 3.20 — Lines 544-546. Why are different stimuli used for the different models? The 2 RNN model is trained with random non-overlapping stimuli. The 3 RNN model is trained with four horizontal bars (as opposed to random stimuli) and is tested with partial cues consisting of the central 50% of the full bars (full bars and cues depicted in Fig. 3A). Couldn't these variations make a difference in the outcome?

Reply: We have used a different set of stimuli in the three-region model to facilitate learning in THL when STIM→THL synapses are static. Specifically, we have set static STIM→THL synapses to support active THL engrams in recent and remote recall in a manner consistent with previous experiments (see page 15, line 602 in the revised manuscript). As a result, we changed both the stimuli and the configuration of STIM→THL synapses in the three-region network to facilitate learning in THL. In particular, we used non-random stimuli (i.e., horizontal bars and their central 50% as in Fig. 2A in the revised manuscript) whose spatial structure could be explored by circular receptive fields in STIM→THL projections to accelerate the learning of accurate stimulus representations (i.e., engrams) in THL (see page 15, line 612 in the revised manuscript). Notably, the engram dynamics exhibited by our current model could theoretically be reproduced with a model that uses randomly-connected static STIM→THL synapses to learn representations of random stimuli but this would require a larger gap in the plasticity timescales of subcortical vs. cortical regions in the network to allow the formation of accurate THL engrams at the end of training while CTX engram cells remain silent up to that point. We have included a discussion of the above remarks on page 11, line 476 in the revised manuscript.

References

1. Takashi Kitamura, Sachie K Ogawa, Dheeraj S Roy, Teruhiro Okuyama, Mark D Morrissey, Lillian M Smith, Roger L Redondo, and Susumu Tonegawa. Engrams and circuits crucial for systems consolidation of a memory. *Science*, 356(6333):73–78, 2017.
2. Til Ole Bergmann and Bernhard P Staresina. Neuronal oscillations and reactivation subserving memory consolidation. In *Cognitive neuroscience of memory consolidation*, pages 185–207. Springer, 2017.
3. Friedemann Zenke and Wulfram Gerstner. Limits to high-speed simulations of spiking neural networks using general-purpose computers. *Frontiers in Neuroinformatics*, 8:76, 2014.

4. Friedemann Zenke, Everton J Agnes, and Wulfram Gerstner. Diverse synaptic plasticity mechanisms orchestrated to form and retrieve memories in spiking neural networks. *Nature Communications*, 6(1):1–13, 2015.
5. Dheeraj S Roy, Young-Gyun Park, Sachie K Ogawa, Jae H Cho, Heejin Choi, Lee Kamensky, Jared Martin, Kwanghun Chung, and Susumu Tonegawa. Brain-wide mapping of contextual fear memory engram ensembles supports the dispersed engram complex hypothesis. *bioRxiv*, page 668483, 2019.
6. Joëlle Lopez, Karine Gamache, Carmelo Milo, and Karim Nader. Differential role of the anterior and intralaminar/lateral thalamic nuclei in systems consolidation and reconsolidation. *Brain Structure and Function*, 223(1):63–76, 2018.
7. Pablo Alvarez and Larry R Squire. Memory consolidation and the medial temporal lobe: a simple network model. *Proceedings of the National Academy of Sciences*, 91(15):7041–7045, 1994.

REVIEWER COMMENTS

Reviewer #1 (Remarks to the Author):

The authors addressed my comments.

Reviewer #2 (Remarks to the Author):

I appreciate you taking the time to address all of my comments. I was particularly pleased with the fact that you went the extra mile and conducted new simulations to substantiate your answers; it is a commendable effort. I have communicated this evaluation to the editors and recommended your current manuscript for publication. I will be looking forward to seeing it online.

Reviewer #3 (Remarks to the Author):

I have worked through the revision of "Coordinated hippocampal-thalamic-cortical communication crucial for engram dynamics underneath systems consolidation". I still have major concerns. One concern is regarding readability. The complex figures remain in the body of the manuscript and now there are 25 supplemental figures! Many terms in these figures are not explained. Another concern is the lack of analysis at the network level. I asked for analysis of neural dynamics, synaptic plasticity and the oscillations referred to in the text. But this was not done adequately. Therefore, whatever the contribution to understanding memory consolidation are lost. Lastly, I do not think the authors took my comments seriously as I will outline below.

In general, I recommend the authors think about what message they want this paper to convey, and then completely reorganize the paper to show this clearly and succinctly, while backing it up with mechanistic analyses and comparisons to empirical data.

Reviewer Point P 3.1 — The stimuli used for encoding episodic memory are overly simple and non-overlapping.

This was responded to on line 100: "Notably, our network model exhibits analogous engram dynamics when trained with random overlapping stimuli (Fig. S3)." I still think this is an important issue for a memory model that should be analyzed more than with one line. The supplemental figure is referred to without any explanation of the methods or discussion of results.

Reviewer Point P3.3 — The dynamics of the model is not analyzed to show that there is oscillatory coupling between regions.

This was responded by "First, we have measured the degree of coupling between pairs of regions in the model by plotting the cross-correlogram of their engram activity and extracting the lag between them (see Fig. S1, Fig. S13, Fig. S14, Fig. S15, Fig. S16, Fig. S17, Fig. S20, Fig. S22, and Fig. S24 in the revised manuscript). We have included a detailed description of these cross-correlograms and the computation of the associated lags (see page 17, line 701 in the revised manuscript)." Line 701 is the methods paragraph on "Statistical Analysis". This gets back to my point about the contribution. A model should propose a mechanism of how some phenomena such as memory consolidation. By burying the analysis and only discussing "our work does not explicitly model sharp-wave ripples, spindles, and slow oscillations but instead captures the fact that the coupling of these oscillations has been linked to coupled engram reactivations" with no further explanation leads one scratching their head. SPW, spindles and slow oscillations are thought to lead to reactivation and consolidation of memories. There should be a thorough analysis of how the dynamics of their model relates to HPC and CTX.

Reviewer Point P3.4 — The results are not directly compared with existing experimental evidence.

The response was, "We agree with the reviewer that our modeling results are not directly compared to available experimental data and we have included a discussion of this limitation".

Without a proposed mechanism and no comparison to empirical data, the contribution of this model is unclear.

Reviewer Point P 3.5 — In general, I found the results section difficult to read. The figures are extremely busy with multiple panels that are difficult to navigate.

I still find this to be true. Take Figure 1 for example. There are 6 panels. Each one is complex. The terms are never explained in the text or the caption. A and B. What are the black and white dots for the stimulus? What do the colors mean? C. How were those weights calculated and compared. What do the asterisks mean on each panel? The panel on the right of C shows a bunch of colors with barely an explanation. The caption read "population activity of engram cells in the consolidation phase". How was this calculated? Tested? Why isn't it discussed in the text? D-F. The false positive and true positive rates are undefined. There is a brief description in the methods. But since this is a key metric, it needs to be described in detail in the results section.

It should be noted that nearly all the figures (5 in the main text, and 25 supplemental) follow this general layout. Therefore, it is critically important to spend some time initially guiding the reader through these complex figures.

Reviewer Point P 3.7 — A strength of this model is the biologically plausible synaptic plasticity. How do the "Hebbian (i.e., triplet STDP) and non-Hebbian (i.e., heterosynaptic and transmitter-induced) forms of plasticity" all contribute to the present results.

The response on line 104, "Critically, each form of Hebbian and Non-Hebbian plasticity in our is essential for ... engram dynamics... (Fig S4)". Figure S4 is another complex figure that follows the layout of Figure 1. No other explanation is provided of how this key mechanism might be important for understanding memory consolidation.

Reviewer Point P 3.9 — The authors use "coupling", "oscillations" and "coordinated communication" in their language. See my comment to Point P3.3 above.

Response to the reviewers

Douglas Feitosa Tomé, Sadra Sadeh, Claudia Clopath

We thank the reviewers for their thorough evaluation of our work. Here, we address each of their concerns point-by-point.

Reviewer 1

The authors addressed my comments.

Reply: We thank the reviewer for all their previous comments.

Reviewer 2

I appreciate you taking the time to address all of my comments. I was particularly pleased with the fact that you went the extra mile and conducted new simulations to substantiate your answers; it is a commendable effort. I have communicated this evaluation to the editors and recommended your current manuscript for publication. I will be looking forward to seeing it online.

Reply: We thank the reviewer for all their previous comments.

Reviewer 3

Major Points

Reviewer Point P 3.1 — The stimuli used for encoding episodic memory are overly simple and non-overlapping.

This was responded to on line 100: “Notably, our network model exhibits analogous engram dynamics when trained with random overlapping stimuli (Fig. S3).” I still think this is an important issue for a memory model that should be analyzed more than with one line. The supplemental figure is referred to without any explanation of the methods or discussion of results.

Reply: The reviewer raises an important point and we have included a thorough discussion of the methods and results of our simulations with overlapping training stimuli (see page 4, line 125 to 139 in the revised manuscript).

Reviewer Point P 3.3 — The dynamics of the model is not analyzed to show that there is oscillatory coupling between regions.

This was responded by “First, we have measured the degree of coupling between pairs of regions in the model by plotting the cross-correlogram of their engram activity and extracting the lag between them (see Fig. S1, Fig. S13, Fig. S14, Fig. S15, Fig. S16, Fig. S17, Fig. S20, Fig. S22, and Fig. S24 in the revised manuscript). We have included a detailed description of these cross-correlograms and the computation of the associated lags (see page 17, line 701 in the revised manuscript).” Line 701 is the methods paragraph on “Statistical Analysis”. This gets back to my point about the contribution. A model should propose a mechanism of how some phenomena such as memory consolidation. By burying the analysis and only discussing “our work does not explicitly model sharp-wave ripples, spindles, and slow oscillations but instead captures the fact that the coupling of these oscillations has been linked to coupled engram reactivations” with no further explanation leads one scratching their head. SPW, spindles and slow oscillations are thought to lead to reactivation and consolidation of memories. There should be a thorough analysis of how the dynamics of their model relates to HPC and CTX.

Reply: We agree with the reviewer on this crucial point and we have included a detailed discussion of: I) how the engram dynamics in our model relate to experimentally-observed neural dynamics in HPC, CTX, and THL; and II) **our model’s proposed neural mechanisms through which coupled engram reactivations mediate systems consolidation of memory** (see page 6, line 233 to 256; page 8, line 349 to 353; and page 11, line 509 to 516 in the revised manuscript). In addition, we elaborate on the proposed functional roles of coupled engram reactivations in systems consolidation when discussing extensions of our three-region network (see page 6, line 258 to 270).

Reviewer Point P 3.4 — The results are not directly compared with existing experimental evidence.

The response was, “We agree with the reviewer that our modeling results are not directly compared to available experimental data and we have included a discussion of this limitation”. Without a proposed mechanism and no comparison to empirical data, the contribution of this model is unclear.

Reply: We agree with the reviewer on this critical point and we have included: I) a new figure to **directly compare neural activation rates in previous experiments [1] and in our model**, II) a description of the associated statistical tests, and III) a discussion of how our simulation results compare to the reported experimental data [1] (see Fig. S13; page 17, line 736 to 740; page 6, line 207 to 231; and page 11, line 502 to 509 in the revised manuscript). For a discussion of our model’s proposed mechanisms underlying systems consolidation, please see related point P 3.3.

Reviewer Point P 3.5 — In general, I found the results section difficult to read. The figures are extremely busy with multiple panels that are difficult to navigate.

I still find this to be true. Take Figure 1 for example. There are 6 panels. Each one is complex. The terms are never explained in the text or the caption. A and B. What are the black and white dots for the stimulus? What do the colors mean? C. How were those weights calculated

and compared. What do the asterisks mean on each panel? The panel on the right of C shows a bunch of colors with barely an explanation. The caption read “population activity of engram cells in the consolidation phase”. How was this calculated? Tested? Why isn’t it discussed in the text? D-F. The false positive and true positive rates are undefined. There is a brief description in the methods. But since this is a key metric, it needs to be described in detail in the results section. It should be noted that nearly all the figures (5 in the main text, and 25 supplemental) follow this general layout. Therefore, it is critically important to spend some time initially guiding the reader through these complex figures.

Reply: The reviewer raises a valid point and we have made a series of changes in the main text and in figure captions to facilitate reading the Results section and navigating figures. Specifically, we have performed the following:

- We have provided a description of the computation of mean weight matrices in the Results section (see page 3, line 74 in the revised manuscript).
- We have provided a description of the computation of the population activity of engram cells in the Results section (see page 3, line 85 in the revised manuscript).
- We have included the definition of lag_{max} in the Results section (see page 3, line 86 in the revised manuscript).
- We have included the definition of memory recall true positive rate, false positive rate, and accuracy in the Results section (see page 3, line 99 in the revised manuscript).
- We have included specific lag_{max} values in the Results section to substantiate claims regarding the degree of coupling between engram reactivations in different regions of our model (see page 3, line 90 and page 6, line 235 in the revised manuscript).
- We have added a description of the meaning of the various colors used in the schematics of the stimulus population, training stimuli, and testing stimuli (see the captions of Fig. 1A, Fig. 2A, Fig. S3A, Fig. S6A, Fig. S7A, Fig. S8A, Fig. S12A, Fig. S15A, Fig. S16A, and Fig. S17A in the revised manuscript).
- We have included a description of the statistical tests denoted by asterisks on figure panels for every figure that includes any statistical test (see the captions of Fig. 1, Fig. 2, Fig. 3, Fig. 4, Fig. 5, Fig. S3, Fig. S4, Fig. S5, Fig. S6, Fig. S7, Fig. S8, Fig. S11, Fig. S12, Fig. S15, Fig. S16, Fig. S17, Fig. S19, Fig. S24, Fig. S25, and Fig. S26 in the revised manuscript).
- We have added a description of what different colors denote in population activity plots and firing rate histograms (see the captions of Fig. 1C, Fig. 2C, Fig. 3C, Fig. 4A, Fig. S2A-B, Fig. S4A, Fig. S5A, Fig. S9A, Fig. S10A, Fig. S16D, and Fig. S17D in the revised manuscript).
- We have added a description of what consolidation times black and gray indicate in cumulative weight distribution plots (see the captions of Fig. S5B and Fig. S25A in the revised manuscript).
- We have included a description of the specific synapses whose weights are tested using the Kolmogorov-Smirnov test in Fig. 5C-D/G-H and Fig. S26B-G (see the captions of Fig. 5 and Fig. S26 in the revised manuscript).

Reviewer Point P 3.7 — A strength of this model is the biologically plausible synaptic plasticity. How do the “Hebbian (i.e., triplet STDP) and non-Hebbian (i.e., heterosynaptic and transmitter-induced) forms of plasticity” all contribute to the present results. Are all of these necessary? Are some more important than others?

The response on line 104, “Critically, each form of Hebbian and Non-Hebbian plasticity in our is essential for . . . engram dynamics. . . (Fig S4)”. Figure S4 is another complex figure that follows the layout of Figure 1. No other explanation is provided of how this key mechanism might be important for understanding memory consolidation.

Reply: The reviewer raises an important point and we have included a detailed discussion of our simulations with ablation of Hebbian and non-Hebbian forms of plasticity to elucidate how each is essential to reproduce experimentally-observed engram dynamics associated with systems consolidation in our network model (see page 4, line 141 to 166 in the revised manuscript).

Reviewer Point P 3.9 — The authors use “coupling”, “oscillations” and “coordinated communication” in their language. See my comment to Point P3.3 above.

Reply: We have included I) a description of how we measure the degree of coupling of engram reactivations in our model before discussing the obtained results (see page 3, line 86 in the revised manuscript), and II) a discussion of how the coupled engram reactivations in our model relate to experimentally-observed neural dynamics associated with systems consolidation (see page 6, line 233 to 256 in the revised manuscript; see also related point P 3.3).

References

1. Takashi Kitamura, Sachie K Ogawa, Dheeraj S Roy, Teruhiro Okuyama, Mark D Morrissey, Lillian M Smith, Roger L Redondo, and Susumu Tonegawa. Engrams and circuits crucial for systems consolidation of a memory. *Science*, 356(6333):73–78, 2017.

REVIEWERS' COMMENTS

Reviewer #3 (Remarks to the Author):

Thank you for addressing many of my concerns. I think these explanations will help readers understand the paper's results. I have no further comments or suggestions.

Reviewer #3 (Remarks to the Author):

Thank you for addressing many of my concerns. I think these explanations will help readers understand the paper's results. I have no further comments or suggestions.

Response to the reviewers

Douglas Feitosa Tomé, Sadra Sadeh, Claudia Clopath

We thank the reviewers for their thorough evaluation of our work.

Reviewer 3

Thank you for addressing many of my concerns. I think these explanations will help readers understand the paper's results. I have no further comments or suggestions.

Reply: We thank the reviewer for all their previous comments.